# Retrieval of Water Vapor using Ground-based Observations from a Prototype ATOMMS Active cm- and mm-Wavelength Occultation Instrument

Dale M. Ward[1], E. Robert Kursinski[2], Angel C. Otarola[1,3], Michael Stovern[4], Josh McGhee[2], Abe Young[5], Jared Hainsworth[6], Jeff Hagen[7], William Sisk[8] and Heather Reed[9]

[1]Department of Atmospheric Sciences, University of Arizona, Tucson, AZ 85721, USA
[2]Space Sciences and Engineering, Boulder, CO, 80301, USA
[3]TMT International Observatory, LLC., Pasadena, CA 91105, USA
[4]Environmental Protection Agency, Denver, CO 80202-1129, USA
[5]Department of Physics, University of Arizona, Tucson, AZ 85721, USA
[6]Hill Air Force Base, A-10 Mechanical Systems, Ogden, UT 84056, USA
[7]Lithe Technology, Tucson, AZ 85721, USA
[8]Department of Astronomy, University of Arizona, Tucson, AZ 85721, USA
[9]LASP, University of Colorado, Boulder, CO 80303, USA

*Correspondence to*: Dale M. Ward (dward@email.arizona.edu)

**Abstract.** A fundamental goal of satellite weather and climate observations is profiling the atmosphere with in situ-like precision and resolution with absolute accuracy and unbiased, all-weather, global coverage. While GPS radio occultation (RO) has come perhaps closest in terms of profiling the gas state from orbit, it does not provide sufficient information to simultaneously profile water vapor and temperature. We have been developing the Active Temperature, Ozone and Moisture Microwave Spectrometer (ATOMMS) RO system that probes the 22 and 183 GHz water vapor absorption lines to simultaneously profile temperature and water vapor from the lower troposphere to the mesopause. Using an ATOMMS instrument prototype between two mountaintops, we have demonstrated its ability to penetrate through water vapor, clouds and rain up to optical depths of 17 (7 orders of magnitude reduction in signal power) and still isolate the vapor absorption line spectrum to retrieve water vapor with a random uncertainty of less than 1%. This demonstration represents a key step toward an orbiting ATOMMS system for weather, climate and constraining processes. ATOMMS' water vapor retrievals from orbit will not be biased by climatological or first guess constraints, and will be capable of capturing nearly the full range of variability through the atmosphere and around the globe, in both clear and cloudy conditions, and will therefore greatly improve our understanding and analysis of water vapor. This information can be used to improve weather and climate models through constraints on and refinement of processes affecting and affected by water vapor.

## 1. Introduction/Motivation

Water vapor is an important constituent in Earth's atmosphere and its distribution in space and time must be known to understand and predict weather and climate. Water vapor is fundamental to the radiative balance of the Earth, both as the most important greenhouse gas and indirectly through clouds. Through its latent heat, water vapor is crucial to formation and evolution of severe weather, transport of energy both upward and poleward in the troposphere and transfer of energy between the surface and atmosphere. Furthermore, water vapor dominates tropospheric radiative cooling which drives convection (Sherwood et al., 2010). Uncertainty in modeled cloud feedback results in the factor of 3 spread in predictions in the surface temperature response to a doubling of atmospheric $CO_2$ concentrations and the cloud feedback depends critically on the strength of the water vapor feedback (Held and Soden, 2000). Predicted amplification of extreme precipitation with warmer temperatures is tied directly to predicted increases in extreme water vapor concentrations and future extreme precipitation may be underestimated (e.g., Allan and Soden, 2008).

Water vapor is challenging to measure because of the wide range of concentrations and scales across which it varies. Water vapor observations must be unbiased and capture the full range of variability in clear and cloudy conditions across the globe in order to improve the understanding and analysis of water vapor, This information is used to initialize weather prediction systems, to monitor trends and variations and to improve weather and climate models through constraints on and refinement of processes affecting and affected by water vapor (e.g., Bony et al., 2015).

Satellite observations are required to gain a global perspective for weather prediction and climate monitoring and for constraining the critical processes at work in different regions across the globe. Unfortunately, present satellite observations provide limited constraints on the water vapor field, particularly when clouds are present, which in turn limits the skill of weather forecasts and our detailed knowledge of water vapor across the globe. For example, GOES observations provide high time and horizontal resolution but very limited vertical information. While hyperspectral IR on polar orbiting satellites provide more information, their temporal sampling is limited and their water vapor estimates are quite noisy with fractional, root mean-square (RMS) differences ranging from 25% in the lower troposphere to 70% around 400 hPa and a tendency toward dry biases up to 30%, depending on cloud type (Wong et al., 2015). While downward looking microwave radiance measurements are particularly useful for determining the column water over the ocean (e.g., Wang et al., 2016), they provide significantly less vertical information than IR and are inherently ambiguous over land, snow and ice due to surface emissivity variations. The point is that present state-of-the-art, radiance-based satellite water vapor remote sensing systems have serious limitations in terms of performance and sampling biases associated with clouds and surface conditions, accuracy, vertical resolution and the ambiguity inherent in the conversion of radiances to the atmospheric state (Rodgers 2000).

Because of these satellite limitations, balloon-borne sondes and dropsondes continue to be the measurement of choice for field campaigns focused on answering key questions about the atmosphere. In fact, the globe would be covered with sondes if the cost to do so were not so completely prohibitive. Operational global weather observing systems therefore rely primarily on more affordable but vertically coarse satellite radiance measurements and the inherent ambiguities in the information they

provide. Unfortunately, this limits how much understanding we can gain from these observations about important atmospheric processes like those associated with clouds, convection and surface exchange.

In this context, GPS radio occultation (RO) has provided a welcome advance in satellite remote sensing through its ability to profile the atmosphere with ~200 m vertical resolution, approaching that of sondes, in all-weather conditions, with very small random and absolute uncertainties. As such, GPS RO has become an important data source for numerical weather prediction (NWP), despite its relatively sparse coverage to date (e.g., Cardinali and Healy, 2014). Its high impact comes from its unique combination of ~200 m vertical resolution, all weather sampling and very low random and absolute uncertainties via its direct connection to atomic frequency standards and relatively simple and direct retrieval method. GPS RO profiles atmospheric refractivity. Two limitations of GPS RO are (1) its inability to separate the dry air and water vapor contributions to refractivity and (2) its insensitivity to water vapor in the colder regions of the troposphere and above (e.g., Kursinski et al., 1997; Kursinski and Gebhardt, 2014). The insensitivity occurs when there is so little water vapor that the majority of the refractivity is dominated by the dry air component.

In recognition of the strengths and weaknesses of GPS RO and radiance measurements as well as the need for better information about water vapor, in 1997 research groups at the University of Arizona and the NASA Jet Propulsion Laboratory (Herman et al., 1997 and Hajj et al., 1997) identified and began developing an RO system that is now called the Active Temperature, Ozone and Moisture Microwave Spectrometer (ATOMMS), which is designed to overcome these GPS limitations by transmitting and receiving signals between satellites in low Earth orbit (LEO) near the 22 and 183 GHz water vapor absorption lines as well as nearby ozone absorption lines. Profiling both the speed of light like GPS RO as well as the absorption of light, which GPS RO does not measure, enables ATOMMS to profile temperature, pressure and water vapor simultaneously from near the surface to the mesopause with little random or systematic uncertainty (Kursinski et al., 2002). It will also profile ozone from the upper troposphere into the mesosphere, scintillations produced by turbulence, slant path cloud liquid water and detect larger cloud ice particles, with approximately 100 m vertical resolution and corresponding 70 km horizontal resolution (Eq. 13, Kursinski et al., 1997). Kursinski et al. (2002) found that such a system could provide water vapor retrievals with a random uncertainty of $1 - 3\%$ from near the surface to well into the mesosphere. Kursinski et al. (2009) estimated the degradation in clouds would be less than a factor of 2. A summary of LEO to LEO occultation measurement concept studies and demonstrations to date at microwave and IR wavelengths is given in Liu et al. (2017).

Regarding the sampling densities that can be achieved with ATOMMS, Kursinski et al. (2016b) noted that a constellation of 60 very small satellites, carrying both ATOMMS and GNSS RO sensors, would produce approximately 26,000 ATOMMS and 170,000 GNSS occultations profiles each day, for a fraction of the cost of a single, operational, polar orbiting weather satellite. These numbers of profiles are approximately 10 and 100 times present GPS RO and radiosonde sampling densities. Such an orbiting ATOMMS constellation providing dense, very high vertical resolution, precision and accuracy water and temperature profiling via radio occultation will complement existing observations of clouds, precipitation and energy fluxes and tie the entire weather and climate system together. This combination will also dramatically improve the realism and utility of global analyses for climate as well as forecasting (increasingly extreme) weather (Kursinski et al., 2016b).

With regard to constraining processes, we briefly discuss three important and representative application areas: moist convection, weather fronts and polar weather and climate.

**Moist convection** is ubiquitous across the globe but inadequately understood which leads to inaccurate representation in models. Environmental variables critical for understanding and predicting moist convection and associated severe weather include temperature, water vapor, stability, and conditional instability in particular, the level of free convection, convective available potential energy (CAPE), convective inhibition (CIN), winds and divergence. Unfortunately, coarse vertical resolution and ambiguities inherent in converting radiance spectra to the atmospheric state limit the ability of satellite radiances to provide detailed constraints on convection related processes. GPS RO provides much needed vertical information across the globe and is particularly useful for determining temperatures and stability in the upper troposphere where conditions are very dry. However, the ambiguity of the wet and dry gas contributions to refractivity under the warmer, moister conditions deeper in the troposphere limit the utility of GPS RO refractivity profiles there.

In contrast, ATOMMS will be the first orbiting remote sensing system to simultaneously profile temperature and water vapor with very high ~100 m vertical resolution and very small uncertainties needed to tightly constrain these environmental quantities relevant to convection, in clear and cloudy conditions, through the troposphere, across the entire globe. While ATOMMS profiles will not resolve detailed horizontal structure at scales much below 70 km, they are sensitive to these scales via the phase and amplitude scintillations that small scale turbulence produces on the ATOMMS signals (Kursinski et al., 2016b). Furthermore, 100 km, which is approximately the horizontal resolution of ATOMMS, is the scale most important for forecasting severe convection in the form of thunderstorms (Durran and Weyn, 2016).

**Weather fronts** are another fundamental class of severe weather poorly constrained by satellite radiance measurements. Unlike radiances, RO measurements can profile fronts from orbit because RO profiles readily penetrate through clouds and the vertical and horizontal resolutions of RO are well matched to the vertical and horizontal scales of weather fronts. While GPS RO can profile fronts in the upper troposphere (e.g. Kuo et al., 1998), the lack of refractivity contrast between the warm-wet and the cold-dry sides of fronts deeper in the troposphere limits GPS RO profiling of fronts there (Hardy et al., 1994). ATOMMS high precision temperature, pressure and water vapor profiles in clear and cloudy conditions will readily distinguish between the warm and cold sides of fronts down through the lower troposphere and precisely determine the location of any frontal surface that crosses an ATOMMS profile (Kursinski et al., 2002).

This unprecedented capability to measure fronts globally will also enable detailed characterization of the dynamics and moisture fluxes of atmospheric rivers out over remote ocean regions to better predict and prepare for the torrential rainfall and flooding they produce following landfall. These observations will also guide refinements in model representations of atmospheric rivers to increase and extend the accuracy of weather forecasts and the climatologically important mid-latitude water vapor transport in reanalyses and climate models (e.g., Guan and Waliser, 2016).

**Profiling in Polar regions,** particularly the near-surface environment, is critical to understanding the causes of ongoing and future climatic changes there. Reducing uncertainty due to our limited knowledge about the critical processes at work there requires quantitative, process-resolving observations that span the entire range of environmental conditions and

behavior across these remote regions. Present understanding comes largely from operational sondes and a small number of field campaigns (e.g., Esau and Sorokina, 2010). While satellites radiance measurements already provide dense sampling of these remote, high latitude regions, they have yielded relatively little insight due to intrinsic ambiguities associated with poor vertical resolution, frequent clouds, near-surface inversions and variations in surface emissivity. As a result, many "global"

satellite products do not extend to the poles (e.g. Chen et al., 2008). While GPS RO has much needed very high vertical resolution, cloud penetration and insensitivity to surface conditions, its impact is also limited, because of the unknown contributions of water vapor and the bulk dry gas to the measured refractivity profiles.

In this context, ATOMMS' precise and very high vertical resolution profiling of temperature, stability, water vapor, pressure gradients, clouds and turbulence, down to the surface, over all types of surfaces, in clear and cloudy conditions, across

the diurnal and seasonal cycles, will bring unprecedented information about the high latitudes and, in particular, the lowermost troposphere, to constrain and reduce presently large uncertainties in surface fluxes and the surface energy budget there.

ATOMMS will simultaneously probe through clouds to determine the gas state as well as the cloud properties themselves, including their phase (liquid, ice and mixed) which are critical in the surface energy budget (e. g., Klingebiel et al., 2015) and fundamental to calculating upward and downward short and long wave radiative fluxes through the atmosphere.

ATOMMS will profile the frequent polar boundary layer clouds too close to the surface to be characterized by CloudSat (Kay and Gettleman, 2009).

ATOMMS will constrain winds via horizontal pressure gradients to further constrain wind shear and moisture fluxes. This wind and cloud information together with ATOMMS' simultaneous profiling of stability and turbulent scintillations will provide a new set of observational constraints over the entire high latitude region to expose flaws in and guide improvements

to presently inaccurate and poorly constrained model parameterizations of sensible and latent heat fluxes. The ability to estimate turbulence and radiative cooling at cloud top are also critical to determining cloud lifetimes and the radiative budget because turbulent entrainment rates influence droplet size and therefore albedo (Esau and Sorokina, 2010). ATOMMS global perspective would provide critical information for understanding why the two poles are evolving so differently.

The preceding examples reveal inadequacies in our present observing system that limit our understanding, and the

substantial increase that ATOMMS promises in our observationally based knowledge and understanding. The performance of ATOMMS profiles approach that of sondes and, when implemented as a constellation such as in Kursinski et al. (2016b), would provide far denser coverage across the globe. For example, the vast Amazon rainforest which is presently profiled twice a day by only 8 sondes (Itterly et al., 2016), would be sampled by approximately 300 ATOMMS profiles and 1,800 GNSS RO profiles each day via the ATOMMS satellite constellation noted above. Thus, an ATOMMS constellation would

create a continuous, dense, global data set, with performance approaching that of sondes, that researchers could divide up as they like into smaller domains (creating essentially their own regional (field) campaigns) to better understand and model key processes and reduce weather and climate prediction uncertainty across the globe.

Our work here is focused on a mountaintop demonstration of ATOMMS' ability to measure water vapor through rain and clouds. Using ground-based ATOMMS prototype instrumentation, we demonstrate the ability of ATOMMS to retrieve

changes in the path-averaged water vapor between the instruments operating between two mountaintops in Southern Arizona to within 1%, during weather conditions that ranged from clear to cloudy to thunderstorms with heavy rain. The ATOMMS mountaintop retrievals worked up to optical depths of 17. The smaller than 1% discrepancies between the measured ATOMMS spectra and the forward modeled water vapor spectra (described in Section 4), in clear, cloudy and rainy condition are unprecedented and more than one order of magnitude smaller than the 25% to 70% uncertainties in AIRS retrievals reported in Wong et al. (2015). At still higher cloud and rain opacities, such as the conditions encountered during our ATOMMS mountaintop experiment, IR and microwave emission-based water vapor retrievals simply do not work. ATOMMS performance in cloud and rain is achieved via a differential transmission approach using a calibration signal, in contrast to passive IR and microwave sensors systems that work via emission. In addition, the vertical resolution attainable via active occultation observing systems is at least an order of magnitude better than that of passive sensors.

The structure of the paper is as follows. Section 2 summarizes the ATOMMS concept for satellites operating in low Earth orbit (LEO) and Section 3 describes this mountaintop experiment. In Section 4, we discuss the water vapor retrievals from the measured mountaintop data. Sources of uncertainty are covered in Section 5, while Section 6 examines validation of the water vapor retrievals with available in-situ measurements. Finally, in Section 7, the encouraging results from the ATOMMS ground-based system lead us to a discussion of the unique capabilities of a future ATOMMS satellite occultation system for improving numerical weather forecasts, monitoring climate changes, and improving our understanding and model representation of processes related to water vapor.

## 2. ATOMMS Concept

ATOMMS is a natural extension of the GPS RO concept. It extends the capabilities and overcomes several limitations of GPS RO by simultaneously measuring atmospheric bending and absorption at several essentially monochromatic signal frequencies in two frequency bands centered on the 22 GHz and 183 GHz water absorption lines, referred to as Low-Band and High-Band respectively. The High-Band includes several ozone absorption lines used to profile ozone. During ATOMMS satellite to satellite occultations, signals transmitted from one satellite are received by the other which yields measurements of the signal phase and amplitude during the occultation. From these, occultation profiles of bending angle and absorption are derived and then used to derive radial profiles of refractivity and the extinction coefficient using Abel Transforms (*Kursinski et al.,* 2002). These are then combined with knowledge of spectroscopy together with the equations of refractivity and hydrostatic equilibrium to derive profiles of air temperature, pressure, water vapor, ozone, and some properties of condensed water.

ATOMMS functions as a precise, active spectrometer over the propagation path between the transmitter and receiver. Retrievals of water vapor from radiance measurements are inherently ambiguous because both the unknown signal source emission and attenuation along the path are unknown and must be solved for, creating an ill-posed problem (e.g., Rodgers, 2000). In comparison to radiance retrievals, ATOMMS has the advantage that the transmitted signal strength is well known

and the observed quantity is simply the attenuation along the path, which makes the retrievals much more direct and less ambiguous. The active approach also enables retrievals retrievals with small random and systematic uncertainty under conditions of large path optical depths, which is not possible for passive retrievals.

Because ATOMMS uses phase coherent signals to measure Doppler shift and bending angle like GPS RO, we write the signal attenuation in terms of amplitude rather than intensity as follows,

$$A(f) = A_0(f)e^{-\tau/2} \tag{1}$$

where $A$ is the measured signal amplitude after the absorption, $A_0$ is the amplitude of the signal that would be measured in the absence of atmospheric attenuation and $\tau$ is the optical depth at the signal frequency, $f$. The factor of ½ multiplying the optical depth comes about because intensity is proportional to amplitude squared. The total optical depth is due to the gas phase optical depth plus the attenuation due to hydrometeors. The gas phase optical depth includes water vapor and dry air absorption, which depend on temperature and pressure. The hydrometeor attenuation also depends on temperature (Kursinski et al., 2009).

**Differential Absorption**

A key to ATOMMS performance is its double differential absorption approach (Kursinski et al., 2002). First, the amplitude observable is the *change* in signal amplitude over an occultation relative to the amplitude measured at time, $t_0$, when the signal path between the two spacecraft is entirely above the atmosphere. Second, the amplitudes of two (or more) signals are measured simultaneously during each occultation. The frequency, $f$, of one signal is placed on the absorption line of interest while the frequency of the second signal, $f_{CAL}$, is farther from line center so that signal can function as an amplitude calibration signal.

The quantity used in the ATOMMS retrievals is the ratio of two amplitude ratios,

$$R(f, f_{CAL}, t, t_0) = \frac{A(f,t)}{A(f_{CAL},t)} \Big/ \frac{A(f,t_0)}{A(f_{CAL},t_0)} \tag{2}$$

The amplitude ratio in the denominator represents the ratio of the amplitude of the tuned signal to the amplitude of the calibration signal at reference time, $t_0$, when the signal is nominally above the atmosphere. The amplitude ratio in the numerator represents the ratio of the amplitude of the tuned signal to the amplitude of the calibration signal at measurement time, $t$, during the occultation. Taking the natural logarithm of $R$ and multiplying by two yields the change in the difference between the optical depths at frequencies $f$ and $f_{CAL}$, from the reference time, $t_0$, to time, $t$.

$$2\log(R) = \tau(f,t) - \tau(f_{CAL},t) - [\tau(f,t_0) - \tau(f_{CAL},t_0)] \tag{3}$$

If the signal path is entirely above the atmosphere at reference time, $t_0$, as will be the case in a LEO-LEO occultation geometry, then the optical depths at time $t_0$ are zero and Eq. (3) simplifies to

$$2 \log(R) = \tau(f, t) - \tau(f_{CAL}, t) \tag{4}$$

The frequency separation between $f$ and $f_{CAL}$ is chosen such that $R$ retains most of the absorption signature while cancelling unwanted common sources of error such as gain variations due to pointing errors, scintillations due to atmospheric turbulence and attenuation due to scattering by hydrometeors. This ratio of ratios approach enables precise measurement of water vapor in the presence of clouds and rain with very small random and systematic uncertainty as we demonstrate below.

### 3. Overview of the ATOMMS Mountaintop Experiment

We designed and built a ground-based, prototype ATOMMS instrument and then used it to demonstrate some key aspects of ATOMMS capabilities and performance in several fixed geometries in southern Arizona with path lengths ranging from 800 m to 84 km. The prototype ATOMMS High-Band system transmits and receives two simultaneous continuous wave (CW) signals tunable from 181 to 206 GHz. The prototype Low-Band system consists of eight CW transmitters and receivers at fixed frequencies from 18.5 to 25.5 GHz spaced approximately one GHz apart, centered approximately on the 22 GHz water vapor absorption line. Below we summarize the content of previous published work based on field experiments with the ATOMMS ground-based prototype.

In terms of ATOMMS water vapor retrievals, Kursinski et al. (2012) demonstrated agreement at the 2% level between water vapor measurements derived along an 820 m path using the ATOMMS High-Band instrument and a nearby, capacitive-type hygrometer. High-Band mountaintop measurements yielded the first detection by ATOMMS of $H_2^{18}O$ via its 203 GHz absorption line (Kursinski et al., 2016b). Such measurements in the upper troposphere will determine isotopic ratios to constrain the hydrological cycle (Kursinski et al., 2004).

Accurate knowledge of spectroscopy is key to interpreting the ATOMMS measurements. ATOMMS itself is perhaps the best 183 GHz spectrometer ever implemented. Its measurements of the line shape near the 183 GHz line center match that of the HITRAN model to within 0.3% (Kursinski et al. 2012) which agrees 8 times better than the best prior estimates of Payne et al. (2008). These same measurements revealed that the line shape of the popular Liebe et al., (1993) model is incorrect (Kursinski et al., 2012). Farther from line center, 5 to 25 GHz above line center, ATOMMS measurements revealed significant discrepancies with the HITRAN line shape (Kursinski et al., 2016b). These discrepancies may help explain inconsistencies between 183 GHz derived water vapor estimates discussed in Brogniez et al. (2016) that may be associated with atmospheric turbulence (Calbet et al., 2018).

In terms of sensing hydrometeors, Kursinski et al. (2012) derived cloud liquid water content (LWC) by combining ATOMMS High-Band measurements with precipitation radar measurements along the ATOMMS signal path. Kursinski et al. (2016b) further demonstrated the ability to derive both cloud LWC and rainfall rates by combining the ATOMMS Low-Band and High-Band measurements. ATOMMS also acts as a scintillometer to sense atmospheric turbulence. Kursinski et al. (2016b) derived the strength of atmospheric turbulence from scintillations of the ATOMMS signal amplitudes and further demonstrated how

these turbulent amplitude variations can be reduced via amplitude ratioing, as needed to derive accurate water vapor estimates in turbulent conditions.

On August 18, 2011, we collected approximately four hours of data with the instruments located on Mt. Lemmon Ridge (2752 m altitude) and Mt. Bigelow (2515 m altitude), separated by approximately 5.4 km. The observing geometry is shown in Fig. 1. The Mt. Lemmon instrument contained the 183 GHz transmitter and 22 GHz receiver and the Mt. Bigelow instrument contained the 22 GHz transmitter and 183 GHz receiver. The water vapor pressure derived from these ATOMMS measurements represents an average over the 5.4 km path which runs above a valley between the mountaintops on which the instruments sit.

**Differences between mountaintop and LEO measurements**

The mountaintop-to-mountaintop geometry differs from the satellite-to-satellite geometry in several important aspects. In the satellite-to-satellite occultation geometry, the ATOMMS differential absorption measurements yield *absolute* water vapor concentrations because the reference signal strength is measured above the atmosphere where there is no absorption. Since we cannot evacuate the path between the two mountaintops, mountaintop-to-mountaintop observations are limited to measuring *changes* in water vapor relative to a selected reference period as defined in Eq. (3). In the satellite geometry, a profile of water vapor is retrieved as a function of altitude via an Abel Transform (Kursinski et al., 2002). In the mountaintop experiment, the signal path is fixed and the retrieved quantity is the change in the average water vapor along the fixed path as a function of time.

In the satellite to satellite occultation geometry, the majority of the signal attenuation occurs along the lowest altitude portion of the signal path centered at the ray tangent point which is 100 to 500 km in length. The attenuation contributed at higher altitudes along the ray path is comparatively much smaller than the contribution near the ray path tangent altitude due to both the limb sounding geometry and the exponential decay in water vapor concentrations with altitude. We note that the Abel transform isolates the contribution from the lowest altitude portion of the signal path. For a vertical resolution of 100 m, the horizontal length of the path through the lowest layer is approximately 70 km (Eq. 13, Kursinski et al., 2002). Because the large water vapor concentrations in the lower and middle troposphere produce impenetrably high opacities near the 183 GHz line when integrated over such long signal paths, this portion of the troposphere must be profiled using the weak 22 GHz absorption line and the ATOMMS Low-Band system from space. This is also the altitude region where liquid water clouds are most common. To achieve our goal of an all-weather observing system, the observations must provide enough information for the inversion routine to be able to separate the signal attenuation due to liquid water absorption from that due to water vapor absorption. Kursinski et al. (2009) showed that the spectral shape of the cloud liquid water absorption at the Low-Band frequencies depends primarily on the cloud liquid water path and cloud temperature. Simultaneously measuring the amplitudes of four Low-Band signals, with at least one of the signal frequencies on the high side of the 22 GHz line, in addition to refractivity plus application of a hydrostatic constraint, enables water vapor, cloud liquid water path and effective cloud temperature to be estimated simultaneously. Thus, with absorption information from at least four Low-Band frequencies, we

can isolate liquid water clouds from water vapor and unwanted variations due to instrumental noise and turbulence. Simulations in Kursinski et al. (2009) showed the uncertainty in cloudy conditions should increase by no more than a factor of 2 relative to clear sky conditions. We also note that Kursinski et al. (2009) recommended using at least 5 signal frequencies in order to expose spectral modeling errors and provide the quantitative information needed to refine the modeling of both the

water vapor and liquid water spectra.

In this mountaintop demonstration, the atmospheric path from transmitter to receiver took place over a narrow altitude range from 2752 m to 2515 m above sea level and was only 5.4 km in length. Over this short path, the water vapor attenuation due to absorption by the weak 22 GHz line was too small to measure accurately. Therefore, in this experiment, we used the ATOMMS High-Band signals to probe near the stronger 183 GHz water line to retrieve changes in water vapor along the path.

Below we show that the liquid attenuation has a relatively flat spectral response across the High-Band frequencies utilized for the mountaintop retrieval of water vapor and essentially ratios out. In the satellite case, at altitudes where liquid clouds commonly occur, the combined attenuation from liquid water and water vapor will make the atmosphere too opaque to probe with the High-Band frequencies and ATOMMS will therefore profile these conditions with the Low-Band signals near the 22 GHz line as noted above.

Another difference is that in the LEO-LEO geometry, profiles of atmospheric refractivity and temperature are derived from a Doppler shift proportional to atmospheric bending (e.g., Kursinski et al., 1997). In a fixed geometry, there is no equivalent Doppler shift and we therefore had to determine the air temperature via another method which is described in Section 4.

A final point relates to instrument stability. The duration of a typical LEO-LEO occultation is approximately 100 s,

which allows little time for instrument drift, while mountaintop measurements can continue for hours or days. Therefore, to maintain instrument stability over the four hour mountaintop observation period, we used water chillers to minimize temperature variations of critical portions of the transmitters and receivers.

In spite of the differences noted above, this ground-based experiment clearly demonstrates the ability of an ATOMMS-type system to probe through and accurately retrieve changes in water vapor under conditions of large total optical

depths with liquid water present along the path.

**Observed Optical Depths**

The measured changes in optical depth at 198.5 GHz (blue line, raw) and 24.4 GHz (red line, raw) are shown in Fig. 2. 198.5 GHz was the frequency of the High-Band calibration signal during this experiment. Also shown are the derived

changes in liquid optical depth at 198.5 GHz (black line), which was computed by subtracting the optical depth changes due to variations in the retrieved vapor pressure and temperature from the total observed optical depth change. The change in optical depth relative to reference period 1 will always be positive for liquid (rain and clouds), because there was no rain or clouds during the reference period. However, the change in optical depth due to changes in vapor pressure and temperature

can be negative, which means that the overall change in optical depth relative to the reference period can be less than the optical depth change due to liquid alone.

The instruments were housed in tents to protect them from weather conditions that spanned from clear to cloudy to thunderstorms with heavy rain, as indicated by the annotations in Fig. 2. This wide range of conditions and associated optical depths provided an excellent field test to evaluate and demonstrate several key ATOMMS capabilities. In-situ measurements of temperature, pressure and water vapor were made at each tent. Web cameras in each tent pointed at the opposite ATOMMS instrument site, providing periodic images of weather conditions and visible opacity.

Fig. 2 indicates that when the ATOMMS observations began, a light rain was falling. The rain ended prior to the First Reference period. A brief rain shower was observed from about 14:43 to 15:02 PM. The sharp peak in the 198.5 GHz liquid optical depth just before 15:00 and absence of a peak in the 24.4 GHz liquid optical depth likely indicates an increase in the number of smaller raindrops. This was followed by a brief clear period before the next rain shower began at 15:10. This rain was initially light, but became a heavy thunderstorm at 15:30. From 15:30 to 16:00 the 198.5 GHz tone was too attenuated to be observed at the receiver. During the heavy rain, the 24.4 GHz liquid optical depth reached a peak value of 10. The 198.5 GHz signal was detected again at 16:00 as the rain lightened. By 16:30, the rain was considerably lighter. The radar data from the Tucson WSR-88D radar (Crum and Alberty, 1993) and field observations indicated that rain was still falling over portions of the path between the two instruments. Note that the liquid optical depths did not return to zero before the next heavier rain shower began around 17:15.

Between 16:28 and 16:31, a cloud advected through the observation path. Field notes and images taken every 30 s show a cloud moving into and through the field of view. Initially the cloud extended only part way across the observation path. It then apparently spanned the entire path for a brief period of less than 2 minutes before gradually clearing out of the observation path. The presence of smaller cloud droplets caused the 198.5 GHz liquid optical depth to increase around 16:30, while little if any change was apparent in the 24.4 GHz liquid optical depth. The fact that the 24.4 GHz optical depth did not drop to 0 indicates some light rain was present as well. The decrease in 198.5 GHz liquid optical depth after the peak at 16:30 likely indicates that cloud droplets or drizzle obscured only part of the observation path.

**Signal Tuning and Detection**

The High-Band portion of the ATOMMS ground-based prototype instrument simultaneously transmits and receives two continuous wave signals that are tunable from 181 to 206 GHz. For this mountaintop experiment, the frequency of the signal generated by one transmitter was swept through a tuning sequence that spanned the instrument's tunable frequency range. This signal was received by a narrowband heterodyne receiver whose second local oscillator was simultaneously swept through its matching tuning sequence. The frequency of the other signal was fixed at 198.5 GHz in order to function as the amplitude calibration signal for measuring differential absorption. There were 122 tuning frequencies in the sweep, separated by 0.25 GHz, except for a gap between 191.5 and 193.5 GHz. This gap is due to the limited receiver response for Intermediate Frequencies (IF) less than one GHz and the first stage local oscillator (LO) being set to 192.5 GHz.

When executing the tuning sequence, the tuned transmitter tone dwelled at a particular frequency in the tuning sequence for 100 ms before moving to the next frequency in the sequence. The timing of the transmitter-receiver tuning was synchronized using GPS receivers. Each received ATOMMS signal was filtered, down converted in frequency, digitized and recorded. The signal frequency in the final receiver stage ranged from 8 to 35 kHz for each of the 122 tuned frequencies. The frequency and power of the down-converted signals were determined using a Fast Fourier Transform (FFT), calculated over a 50 ms integration time. The reason that only half of the 100 ms tuning dwell time was used was to allow time for each synthesizer tune to settle. Each FFT-derived signal power estimate was then converted to an amplitude by taking the square root. The calibration signal amplitudes were computed using the same method.

One sweep through the frequency tuning sequence took 12.2 s. The instrument cycled through the four combinations of the two transmitters and two receivers before repeating the tuning cycle in order to help isolate any transmitter or receiver issues. Thus, a full tuning cycle was completed every 48.8 s. The observations from the four combinations of transmitter-receiver pairs were then averaged together such that new estimates for the ATOMMS signal amplitude ratios at all of the 122 tuning frequencies were generated every 48.8 s (Eq. (2)). As a result, the integration time used to estimate the signal amplitude and frequency for each of the 122 frequencies in the tuning sequence was four times 50 ms, or 200 ms.

## 4. Interpretation of Measurements

ATOMMS observations of $R$, defined in Eq. (2), are sensitive to *changes* in the integrated water vapor along the path between the instruments. The retrieval algorithm discussed below determines changes in water vapor pressure relative to a reference period. We selected two reference periods that are identified in Fig. 2. The first period spanned 2:23 to 2:31 PM, shortly after data acquisition began, and the second spanned 4:51 to 4:56 PM, approximately 2.5 hours later. These are periods of relatively constant amplitude spectra due to relatively constant vapor pressure and temperature and relatively low optical depth, which maximizes the number of usable frequencies nearest line center. Comparing solutions derived using the two different reference periods provides some assessment of instrumental drift.

The retrieval algorithm determines the change in vapor pressure relative to the reference period by finding the best forward-calculated fit to each observed ATOMMS amplitude ratio spectrum (Eq. 2) using a least squares method. To forward model the clear sky atmospheric attenuation, we used an atmospheric propagation tool known as the Atmospheric Model (am), version 7.2 (Paine, 2011), which we will refer to as *am7.2*. This model was shown to fit the ATOMMS measurements to the 0.3% level in previous work with the ground-based ATOMMS prototype system (Kursinski et al., 2012). In operation, the ATOMMS ratio, $R$ in Eq. (2), is determined from measurements at times, $t$ and $t_0$, for a range of frequencies, $f$, which produces a frequency spectrum of the ratio. In forward calculations of Eq. (2), we assume that the vapor pressure, air temperature, and air pressure are known at the reference time, $t_0$, and the air pressure and temperature are known at time, $t$. The solution is determined by finding the change in vapor pressure from the reference value that provides the best least squares fit between

the forward-calculated and observed ATOMMS ratio spectra. During this experiment, we were able to accurately determine signal amplitudes up to total optical depths due to gas plus liquid water of 17.

For the purposes of determining the average water vapor along the path, we used 15 tuning frequencies spanning 187.861 GHz to 191.361 GHz to make the water vapor retrievals. Since the greatest sensitivity to changes in vapor pressure occurs at line center, it is desirable to utilize frequencies as close to line center as possible. For this field test, tuning tones with frequencies lower than 187.861 GHz were too attenuated to be measured accurately even during clear skies. During periods of lighter rain and clouds, the additional attenuation by liquid water caused the retrieval frequencies nearest line center to become too opaque to measure accurately, reducing the number of frequencies available for the fit. The liquid optical depth in Fig. 2 is the liquid optical depth measured by the calibration signal, $f_{CAL}$ = 198.5 GHz. The liquid optical depth was computed by subtracting the forward-calculated change in gaseous extinction relative to the reference period from the observed change in optical depth relative to the reference period, which includes changes in both liquid and gaseous extinction. During the heaviest rain period, none of the High-Band signals could be measured due to strong liquid attenuation.

The retrieved path-averaged vapor pressure between the instruments is shown in Fig. 3A. The figure shows 12 different solutions that were used to estimate the random uncertainty in the retrieval of vapor pressure. The methodology used to compute the 12 solutions is described in Section 5. The half range of the 12 solutions shown in Fig. 3B is generally less than 0.1 hPa. Most of the fractional uncertainties are well below 1% of the vapor pressure, indicating that the solution is highly constrained by the observations. The path-averaged vapor pressure varied from 10.2 to 16.5 hPa over the nearly four hour observation period. The measured vapor pressure peaked in association with the rainy period before 15:00. Following that rain shower, there was a brief intrusion of drier air centered near 15:15 before the vapor pressure rapidly increased prior to the thunderstorm at 15:30. Immediately following the heavy rain after reacquisition of the High-Band signals, the vapor pressure dropped to its lowest value. In Section 6, we note that similar advection of dry air following summertime thunderstorms in this region have been observed in previously published work (Kursinski et al., 2008) and show that our estimation of the minimum vapor pressure was consistent with the nearby radiosonde observations from Tucson. During the brief cloud passage at 16:30, there was a sharp increase and peak in the vapor pressure that brought the relative humidity up to approximately 100%. The vapor pressure fell sharply following the passage of the cloud. There was one more peak in vapor pressure at 17:00 before the sharp rise associated with the rain that began at 17:30.

**Determining temperature**

Retrieving changes in water vapor versus time from the measured absorption spectra requires knowledge of atmospheric temperature and pressure. In the eventual LEO-LEO occultation measurements, ATOMMS will profile both the atmospheric Doppler shift and attenuation of the occulted signals, from which profiles of temperature, pressure and water vapor will be derived (Kursinski et al., 2002). In the static mountaintop-to-mountaintop geometry, there is no Doppler shift and only the attenuation portion of the ATOMMS measurements is available. Pressure was determined using barometers on each mountaintop. Determining the atmospheric temperature along the signal path was more challenging.

During this experiment, three nearby thermometers measured the surface air temperature. An Arduino weather station was located next to each ATOMMS instrument and an automated weather station was located in the town of Summerhaven, about 300 m below Mt. Lemmon and 700 m to the north. Unfortunately, these surface temperature observations were not entirely representative of the air temperature aloft along the ATOMMS signal path because of their close proximity to the surface and a high bias in the Arduino temperatures due to heat generated by the ATOMMS instrumentation inside the protective tents.

To better estimate the temperature along the signal path, we derived the average air temperature along the path from the pressure scale height using the hypsometric equation and time-varying barometric pressure measured at the two ATOMMS instruments

$$\overline{T_V} = \frac{g\Delta Z}{R_d} \left[ \ln\left(\frac{P_{Big}}{P_{Lem}}\right) \right]^{-1} \tag{5}$$

where $g$ is gravitational acceleration, $\Delta Z$ is the altitude difference between Mt. Lemmon and Mt. Bigelow, $R_d$ is the gas constant for dry air, $P_{Big}$ and $P_{Lem}$ are the measured air pressures on Mt. Bigelow and Mt. Lemmon respectively, and $\overline{T_V}$ is the layer mean virtual temperature. The air temperature is obtained from the virtual temperature, e.g., Wallace and Hobbs (1977).

While Eq. (5) ideally provides the desired layer mean temperature needed for spectral calculations of $R$, there are issues with this approach. The sensitivity of Eq. (5) to small dynamic pressure variations made short term temperature estimates noisy. The horizontal separation between Mt. Lemmon and Mt. Bigelow caused the estimated temperature to be sensitive to propagating pressure perturbations. Finally, the assumption of hydrostatic balance in Eq. (5) is not true during thunderstorm activity. To alleviate these issues, we used a one hour running mean of the air pressure.

Temperatures derived in this manner are biased by small biases in barometric pressure. To minimize this bias, we shifted the entire temperature time series by 2.15 K so that the relative humidity was 100% at 16:30, when the cloud was present. Figure 4 shows the derived air temperature between the instruments that was used in the retrievals in black, as well as the nearby, in-situ thermometer observations, which are shown in red, green, and blue. The uncertainty associated with this temperature estimation is discussed in Section 5.

**Water vapor spectra**

Figure 5 shows four examples of fitted ATOMMS ratio spectra. The outstanding agreement between the measured and modeled spectra is immediately evident in that most of the individual ATOMMS amplitude ratio spectra fall within ±0.15 hPa (which is ±1%) of the calculated spectra. This is true for most of the individual retrievals.

Figure 5A shows a retrieval made during the clear period around 15:08, following the first rain period. All 15 frequencies spanning 187.861 to 191.361 GHz were available and closely fit the forward-calculated ATOMMS ratio. Figure 5B shows a retrieval made during the first rain period at 14:51. While the two frequencies nearest line center were lost due to

the increase in optical depth caused by rain, the remaining 13 ATOMMS frequencies yielded accurate vapor pressure retrievals during the rain.

Panels C and D of Fig. 5 show retrievals made at 16:29, during the cloudy period. The solution in Panel C uses the first reference period while the solution in Panel D uses the second reference period, which is closer to the time of the cloudy period. The difference between the shape of the ATOMMS ratio spectrum in Fig. 5C and 5D is due to the use of the two different reference periods, which changes the amplitude ratio in the denominator of Eq. (2). The increased liquid optical depth due to the cloud eliminated the three frequencies nearest line center. Although scatter about the best fit forward calculation line is larger than that in Panels A and B, the fitted forward calculations constrain the water vapor solution quite well, despite the presence of the cloud and some light rain. The better fit that results when using the second reference period indicates that there was some subtle instrumental drift over the 2.5 hours between reference periods. Near the cloud peak, the Reference 1 water vapor solutions are greater than the Reference 2 solutions by only 0.03 hPa (0.2%), indicating the level of robustness of these vapor pressure retrievals.

## 5. Sources of Uncertainty and Validation of Results

There are a number of sources of uncertainty in the ATOMMS mountaintop water vapor retrievals that include

(1)  Measurement errors including signal-to-noise-ratio (SNR) and instrument drift,

(2)  Undesired environmental effects such as scintillations due to turbulence,

(3)  Errors in modeling including gaseous spectroscopy and particulate scattering,

(4)  Biases due to errors in the reference period air temperature and water vapor estimates, and

(5)  Errors in the estimated time varying, path-averaged, air temperature

(6)  Uncertainty in spectral fitting

In terms of measurement errors (Category 1), the high SNR that enabled penetration and water vapor retrievals up to optical depths of 17 is not a significant source of error, except, of course, when optical depths exceeded 17 and became impenetrable. As noted, we did see signs of subtle instrument drift over approximately 2.5 hours, which is 9,000 s, that shifted the retrieved water vapor amount by 0.2%. However, because the duration of a LEO occultation is only about 100 s, errors due to instrument drift in LEO should be very small.

Turbulence-induced amplitude scintillations (Category 2) were quite significant during the periods of strong convection. These were reduced by almost an order of magnitude via amplitude ratioing with the calibration signal (Kursinski et al., 2016b). The strong peaks near 14.6 hours in Fig. 3B are caused by momentary noise in the calibration signal, which influences the frequency ratioing. Outside of this peak the largest fractional uncertainty is about 1.8% of the vapor pressure (green line). We attribute most of this to turbulent-induced scintillations that remain after the frequency ratioing. Thus, for the

conditions of this field experiment, the upper bound for the random error in the vapor pressure retrieval due to turbulence is about 1.8% of the vapor pressure.

In terms of spectroscopic errors (Category 3), we again note that ATOMMS is itself a very high spectral resolution spectrometer such that the ATOMMS data can be used to refine the spectroscopic models and make them as accurate as the ATOMMS observations.  Along these lines, we also note that in order to diagnose and reduce spectroscopic errors, Kursinski et al. (2009) recommended increasing the required number of Low-Band signals from 4 to 5 to make the solutions systematically over-determined in order to identify systematic errors in spectroscopic models and then refine those models.

Errors in the reference period temperature and water vapor estimates (Category 4) create unknown biases in our mountaintop estimates. These biases are not relevant to the eventual LEO system because, in the LEO-LEO occultation geometry, the reference period occurs when the signal path is above the detectable atmosphere where the atmospheric density is essentially zero.

The primary cause of temperature-related uncertainty is in the *change* in temperature between the reference period and the observation time (category 5).  Errors in the absolute temperature are relatively insignificant, i.e., temperature biases are not a significant source of uncertainty in the water vapor retrievals in comparison to errors in estimating the change in temperature relative to the reference periods. For the conditions of this particular experiment, based on forward calculations made with *am7.2* for the range of temperature and vapor pressure conditions observed during the experiment, the sensitivity of the change in derived water vapor due to a temperature change relative to the reference period temperature was approximately -0.17 hPa/°C.  Examples of the sensitivity of the ATOMMS ratio, Eq. (2), to changes in vapor pressure, temperature, and air pressure relative to the reference conditions for this experiment are shown in Fig. 6. The figure plots the forward-computed ATOMMS ratio spectrum for four different changes relative to the reference conditions. For the conditions of the field experiment, we were able to measure amplitudes for signal frequencies of 187.861 GHz and higher.  Lower frequencies closer to line center were too attenuated to track. The figure shows the change in the ATOMMS ratio spectrum resulting from a change in air pressure of 10 hPa, which is much larger than the $\pm 2$ hPa changes in air pressure that were observed during the experiment.   Therefore, the sensitivity of the ATOMMS ratio to changes in air pressure is quite small relative to changes in vapor pressure. As the figure shows, for frequencies greater than 187.861 GHz, a one hPa decrease in vapor pressure produced approximately the same ATOMMS amplitude ratio spectrum as a 5.9° C increase in air temperature. Larger changes in vapor pressure, such as the -3 hPa line in the figure, are easily distinguished from changes in air temperature. Based on Fig. 4, the uncertainty in the change in temperature relative to the reference period temperature during this experiment was less than 3°C, which places an upper bound of a 0.5 hPa water vapor uncertainty due to the temperature uncertainty.

The misfit between the measured ATOMMS amplitude spectral ratios and the forward calculation of those spectral ratios (category 6) are sensitive to all of the error types noted above. To understand and characterize the robustness in the spectral fits, we varied the number of frequencies used in the fits. The baseline retrieval utilized the amplitudes of the 15 signals whose frequencies range from 187.861 to 191.361 GHz. Five additional retrievals were implemented using different subsets of these 15 frequencies. Specifically these subsets were the 10 lowest frequencies, the 10 highest frequencies, the 5

lowest frequencies, the 5 middle frequencies, and the 5 highest frequencies within the 187.861 to 191.361 GHz frequency range. We also ran the same 6 cases using the second reference period. The same temperature versus time was used for all 12 cases.

Figure 3A shows the resulting 12 solutions. The blue line in Fig. 3B shows the spread across the 12 retrievals, defined as the maximum minus the minimum vapor pressure divided by two. This half range represents a conservative estimate of the random uncertainty of the retrieved vapor pressure changes that includes both measurement and *am7.2* modeling errors. The average half range is 0.077 hPa which corresponds to a fractional uncertainty of approximately 0.6%. This small spread across the 12 cases indicates that instrument drift over the four hour observational period was quite small and that the ATOMMS spectral observations tightly constrained the vapor pressure with little ambiguity over a wide range of clear, cloudy and rainy conditions in optical depths up to 17.

The amplitude ratio in Eq. (2) reduces common mode sources of error and uncertainty. Ratioing of the amplitudes of two signals, as was done here, eliminates the effects of liquid particle extinction to the extent that the liquid extinction is spectrally flat over the ATOMMS tuning range and calibration frequencies. For raindrop-sized spheres of water, Mie theory predicts that the mm wavelength spectrum of extinction is nearly flat. For smaller cloud droplets, Mie theory combined with the dielectric model of liquid water indicate that the mm (and cm) wavelength extinction increases approximately linearly with frequency due to absorption by liquid water. Near 16:30, the passage of a cloud between the mountaintops coincided with an increase in the 198.5 GHz extinction but no increase in the 24.4 GHz extinction, indicating the presence of very small particles along the path. We adjusted the retrieval algorithm to account for this expected cloud droplet spectral dependence over the High-Band frequency range which caused the retrieved vapor pressure to increase by 0.8%. The increase was necessary to compensate for the slight spectral variation in liquid water attenuation that resulted from using the Mie cloud model (Bohren and Huffman, 1983). Surprisingly, the spectral misfit to the ATOMMS observations increased slightly. The reason is not clear.

This small 0.8% change in the retrieved vapor pressure provides some indication of how effective the calibration signal ratioing is in minimizing the sensitivity of the ATOMMS water vapor retrievals to hydrometeors. In the future, the High-Band system will have 4 rather than its present 2 signals in order to place calibration signals on both the low and high frequency sides of the 183 GHz water vapor line to reveal and compensate for any overall spectral tilt caused by particle extinction as well as other effects. This should greatly reduce cloud ambiguity in the 183 GHz based water vapor retrievals.

## 6. Validation against in-situ measurements

In previously published work, we demonstrated the ability of the ATOMMS prototype system to accurately retrieve changes in water vapor along a relatively short 820 m path across the University of Arizona campus in clear conditions. In that experiment, the atmosphere was well mixed and nearly homogeneous along the observation path such that the retrieved changes in water vapor from ATOMMS matched those observed with an in situ sensor near one end of the path to 1-2%

(Kursinski et al., 2012).   Based on these results, our intent had been to validate these ATOMMS moisture retrievals in the presence of clouds and rain via comparison with independent, in-situ moisture measurements analogous to the ~1% validation of clear sky ATOMMS retrievals along a shorter path demonstrated by Kursinski et al. (2012).  However, we came to realize that quantitative validation of the ATOMMS water vapor retrievals for this mountaintop experiment was limited by the substantial spatial inhomogeneity of the moisture field itself associated with a longer path, over mountainous terrain, during thunderstorm activity. The large variations of water vapor produced by the turbulent, moist, convective activity limited the level of agreement between the several in-situ sensors.

The spatial inhomogeneity of the water vapor field is evident in Fig. 7, which shows that ATOMMS water vapor retrieval and observations from three nearby in-situ sensors as well as the measurement from the Tucson radiosonde at the altitude of the ATOMMS experiment.  The differences between the in-situ sensors are indicative of the magnitude of moisture variations along the 5.4 km path.  The observation geometry in Fig. 1, shows that the ATOMMS-derived vapor pressure is an average over the 5.4 km path that runs above a valley between the mountaintops on which the instruments sit. The High-Band transmitter was located at the position marked and labeled as "Physics/Atmos bldg Radio Ridge" at an altitude of 2752 m and the High-Band receiver was located at the position marked and labeled as "Catalina Station Steward Observatory" at an altitude of 2515 m. In-situ sensors were located on the ground at the two instrument sites, with another at the location marked and labeled as "Summerhaven," which is about 830 m from the observation path in a valley at an elevation of 2439 m.

The spatial variability of the water vapor during this experiment was large. A measure of the water vapor variability over the 5.4 km observation path is provided by computing the root mean square (RMS) differences for the three available in-situ sensors during the experiment, namely the two sensors at each end of the observation path and data from a sensor in the town of Summerhaven in the valley below the observation path. The RMS of the differences between the three in-situ sensors and the ATOMMS derived water vapor was approximately 8% during the period from 14:00 to 15:30, which preceded the first heavy rain period. Water vapor variations during the most active convective periods were likely larger. In the appendix, we discuss the difficulty and very high (prohibitive?) cost of designing and employing an in-situ observational network capable of verifying the ATOMMS retrievals for the conditions encountered during this experiment.

**Cross correlations**

Despite the inherent differences in the horizontal averaging of ATOMMS and the in-situ instruments, there is substantial cross-correlation between these water vapor measurements. We show this by examining the correlation between the ATOMMS-retrieved path-average water vapor and the in-situ water vapor sensor located on Mt. Bigelow.  Figure 8A shows the ATOMMS retrieval for the path averaged vapor pressure in blue and the measured vapor pressure from the in-situ sensor on Mt. Bigelow in red. Substantial cross correlation is clearly evident between the two data sets. The other colored lines in Fig. 8A show time-shifted segments of the in-situ observations, as described below, that make the correlation between the datasets more visually apparent. In order to demonstrate and quantify the cross correlation between the ATOMMS-derived vapor pressure and the in-situ observations, we separated the datasets into several different time segments because the time lag

between the two observations of water vapor varies as the wind conditions change. We discuss four particular time segments defined as follows

(1) 14.06 to 14.79 hours, which is approximately the first 45 minutes of data collection;

(2) 14.99 to 15.49 hours, which is the period leading up to the first heavy rain period when the ATOMMS High-Band signals became too attenuated to track;

(3) 16.00 to 16.42 hours, which is the period when the High-Band signals reappeared following the heavy rain; and

(4) 16.75 to 17.39 hours, which is the period immediately following the cloudy period.

Figure 8B shows the correlation coefficients as a function of sample time lag. Consecutive ATOMMS samples are separated by 48.8 s. The peak cross correlation coefficients range from 0.78 to 0.97, which indicate strong correlation between the ATOMMS-derived water vapor pressure and the in-situ observations of water vapor pressure on Mt Bigelow. Positive lags indicate periods when ATOMMS observed water vapor variations occurred earlier than those variations in the in-situ observations on Mt Bigelow. Although the winds were occasionally gusty, with variable direction due to shower and thunderstorm activity, there were two systematic shifts in the prevailing wind direction observed in the field: a shift from W to NNW around 15:48 and a shift from NNW to ENE around 16:55. These wind shifts were observed both from the motion of clouds in sequences of web camera images taken from Mt Bigelow and by the Tucson WSR-88D radar (Crum and Alberty, 1993). The ATOMMS instruments were oriented along a NE to SW direction, with Mt. Bigelow on the SW end (Fig. 1). Figure 8B indicates that the first three time segments had positive lags, while the last time segment had a negative lag. This is consistent with our wind observations, in which the wind direction had a component from the observation path toward Mt. Bigelow for the first three time periods, and from Mt. Bigelow to the observation path for the fourth time period.

**Moist bias in *in situ* sensor sampling**

Another issue in validating the ATOMMS water vapor retrievals against the in-situ sensor results is a moist bias in the ground measurements relative to the overlying air after the period of heavy rain. The bias is due to evaporation from the wet surface moistening the near-surface air, which is the air whose properties are measured by the in-situ sensors. As a result, with the exception of the cloud around 16:30, the retrieved ATOMMS water vapor amounts over the 80 minutes following the heavy rain were systematically lower than the surface measurements. This continued until approximately 17:20 when the steady increase in water vapor and rain began and continued through the end of the experiment. The largest differences occurred shortly after the most intense rain, when ATOMMS measured a vapor pressure of 10.2 hPa, the smallest of the entire experiment. This value is approximately 25% lower than water vapor measured at the surface stations. Such behavior where moisture at the surface varies little while air aloft becomes significantly drier following summertime thunderstorms is common in this region (e.g., Fig. 4 in Kursinski et al. (2008)). It is also common in the Amazon (e.g., Fig. 7 in Schiro et al., 2016) and may be associated with mid-level inflow of drier air into the precipitating region that results in evaporative cooling and descent of this air (e.g., Leary, 1980 and Houze, 2004).

For the period of relatively dry air following the cloud, the 00 UTC Tucson radiosonde profile provides perhaps the best validation of the ATOMMS results. The sonde launched between 16:30 and 16:45 from a location about 28 km southwest of the experiment and ascended through the Mt. Bigelow to Mt. Lemmon altitude interval between 16:35 and 16:50 at a location approximately 20 km south of the observation path. According to the sonde, the average vapor pressure in the layer

between Mt. Bigelow and Mt. Lemmon was about 12.3 hPa which is within a few percent of the ATOMMS water vapor retrievals following the cloud's passage. We also note that moisture concentrations measured on Mt. Lemmon decreased steadily through this period reaching a minimum of 12.7 hPa at 17:25, a value essentially identical to the ATOMMS moisture retrieval at this time (Fig. 7). This decrease, despite the evaporative moistening from the wet surface, suggests that dry air was indeed advecting over Mt. Lemmon. Thus, the combination of the sonde profile, the ATOMMS measurements and Mt Lemmon

surface measurements all indicate passage of a relatively dry, horizontally extended, air layer following the heavy rain.

Further examination of the operational sonde profiles launched in Tucson that morning around 4:30 AM and particularly that afternoon, around 4:30 PM, provide additional clues as to what happened that afternoon. Figure 9 shows the specific humidity and potential temperature calculated from the Tucson August 19, 00 UTC sonde for the lowest 3000 m above Tucson. The green hatched region shows the altitude interval across the ATOMMS observation path. In the afternoon sonde

profile, the potential temperature, $\theta$, and specific humidity, $q$, are nearly constant between the surface and 2300 m above sea level (msl), indicating that the boundary layer (BL) near 16:30 local time extended to about 2300 msl. In contrast, cloud base at 3150 msl where the dew point equals the temperature in the sonde profile, and the 500 m near-adiabatic layer immediately below it, further indicate that earlier in the afternoon, the well-mixed, dry adiabatic, sub-cloud BL very probably extended up to 3150 msl. Between 2300 and 2750 msl is a thermal inversion layer that is noticeably drier than the air immediately above

and below it. The ATOMMS measurements were made within this altitude interval. The relatively low moisture concentrations in this layer measured by both ATOMMS and the afternoon sonde combined with the fact that the $\theta$ of this inversion layer is lower than the $\theta$ of the peak afternoon BL indicates this air was likely cooled diabatically by evaporation of precipitation falling through it during the turbulent period of heavy rain. The net effect of this process was to increase the $q$ and reduce the $\theta$ of this air, causing it to descend from a higher altitude to where it was measured by ATOMMS. Similarly, the fact that the

$\theta$ of the late afternoon boundary layer below the ATOMMS layer, is 2.5 K lower than that of the peak afternoon BL also indicates that that air has also been evaporative cooled and descended as a result. Such evaporative cooling and descent and moistening of dry air layers is a well-known feature of squall lines (e.g., Houze, 2004) and cause microbursts which are well known in Arizona (e.g. Willingham et al., 2010). Further understanding of the details of what happened that afternoon will require detailed modeling with a convection resolving model, which is beyond the scope of the present research.

## 7. Discussion

The results of this ATOMMS field test demonstrate that the differential absorption concept using an active microwave spectrometer works very well, yielding performance consistent with theoretical expectations that is well beyond the capabilities and performance of passive radiometers. Using a prototype ATOMMS instrument we developed, we measured differential absorption spectra and then forward modeled those spectra, achieving better than 1% agreement, through clear air, clouds and rain to determine the changes in the path-averaged water vapor pressure between the ATOMMS instruments. We demonstrated water vapor retrievals made during cloudy and rainy periods that were only slightly noisier than those made during clear sky periods. Accurate retrievals of water vapor pressure were made through optical depths up to 17, thus demonstrating the exceptionally wide dynamic range achievable via the differential absorption approach. The fact that this performance was achieved under turbulent conditions associated with intense, local thunderstorms also indicates the effectiveness of the differential approach in reducing the impact of turbulence.

While the variable, turbulent conditions associated with convective activity together with passing clouds and rain provided an excellent test of the ATOMMS system's ability to function and perform in very challenging conditions, it also limited the level of validation that could be achieved against in-situ surface sensors. The disagreement amongst the three nearby in-situ sensors revealed the substantial inhomogeneity in the water vapor field in the vicinity of the 5.4 km observation path. Prior to the first heavy rain period, the RMS of the differences between the in-situ sensors was approximately 8%, which set an upper bound to which the ATOMMS retrieved changes in water vapor pressure could be validated by the in-situ sensors. It is also important to note that ATOMMS measured the change in the *path-averaged* vapor pressure which will differ somewhat from point measurements along the path with a magnitude that depends on the inhomogeneity of the water vapor along the path.

During the period following the heavy rain, the ATOMMS measurements revealed systematically drier conditions than the nearby in-situ sensors. These differences were likely due to the fact that the in situ sensors were located at the surface while the path between the ATOMMS instruments was aloft. As a result, the in-situ sensors measured the humidity of air moistened by evaporation from the rain soaked surface, while ATOMMS measured the humidity of air aloft above the valley between the two instruments. The nearby Tucson radiosonde indeed indicated that, following the thunderstorm, a layer of drier air passed through the area. Thus, direct validation the ATOMMS retrievals against the in-situ sensors was limited to about 8%. In the appendix we discuss why it would have been extremely difficult to validate our retrievals at the 1% level with in-situ observations for the conditions encountered during this field experiment.

The better than 1% agreement achieved between the measured ATOMMS spectra and a forward microwave propagation model was substantially better than the comparisons with in situ sensors and indicates the very small level of uncertainty associated with the changes in water vapor that ATOMMS measured. Despite our varying both the combinations of signal frequencies used in the retrievals and the reference times, the agreement remained better than 1%, indicating that

there is simply very little ambiguity in the retrievals of changes in the path-averaged vapor pressure. This essentially brings laboratory-quality measurements out into the field, a very desirable and sought-after property of any measurement system.

In terms of the number of signal frequencies required to accurately determine the water vapor, we used between 5 and 15 tuned signal frequencies plus a calibration signal at a fixed frequency for the water vapor spectral fits. The agreement and consistency of these results indicate that the amplitudes from just a few tuned frequencies and a fixed frequency amplitude calibration signal are needed to produce water vapor retrievals with very small random and absolute uncertainties. We also note that the spectral sweeps used in the mountaintop experiment were intentionally finely spaced in frequency, and therefore slow as well as redundant in order to assess instrument performance, the absorption and scattering spectra and the performance of the retrievals. Faster spectral sampling, as required for LEO-LEO occultations, is readily achievable using a combination of faster switching synthesizers and a smaller number of frequencies to sample the spectrum.

These field measurements of attenuation made near the 183 GHz water vapor absorption line in the presence of rain and liquid clouds enabled us to assess the attenuation due to liquid hydrometeors and the ambiguities associated with them. In terms of *raindrop-sized* liquid hydrometeors, Mie theory predicts that their attenuation across the 183 GHz band has little dependence on signal frequency. As a result, the attenuation due to rain largely ratioed out when we applied the differential absorption technique to determine the changes in water vapor. According to Mie theory, the attenuation of *cloud droplet-sized* liquid hydrometeors in the 183 GHz band has a spectral dependence that increases approximately linearly with frequency. However, when we accounted for this anticipated dependence, the fit between the observations and forward calculations from a microwave propagation model became slightly worse. The reasons for this are as yet unclear.

In the eventual LEO configuration, the ATOMMS signals will encounter a wider range of hydrometeors and spectral dependencies across both the High and Low-Band frequency bands. For example, the 183 GHz band will profile water vapor at high altitudes through ice clouds that will attenuate the signals via Rayleigh scattering which depends approximately on the fourth power of the signal frequency. The LEO version of ATOMMS will provide the information necessary to observe and account for such non-vapor effects using at least three simultaneous signal frequencies to place amplitude calibration signals on both the low and high sides of the absorption line and the third frequency on the line. At altitudes where most *liquid* hydrometeors are encountered, observations in the 22 GHz band will be used to make water vapor retrievals. The liquid water absorption spectrum across the Low-Band frequencies is generally more complex than the ice particle scattering across the High-Band frequencies. Thus, in order to separate the water vapor absorption from the cloud liquid water absorption, we must observe the amplitudes from at least four Low-Band frequencies, with at least one of the signal frequencies on the high frequency side of the 22 GHz absorption line, since the liquid water absorption increases with frequency across the entire low frequency band, while the water vapor absorption is greatest at line center and will have the opposite frequency dependence on the high frequency side of the line. Under clear sky conditions, measurements of three to four simultaneously frequencies will allow evaluation and possibly refinement of the spectroscopy of the 22 and 183 GHz water lines. At least one additional frequency would be required to evaluate and improve spectroscopy when clouds are present.

The ability of ATOMMS signals to penetrate though optical depths up to 17 demonstrated here (which would have reached 19 with more stable synthesizers) and retrieve water vapor to 1% under a wide range of atmospheric conditions ranging from clear to cloudy to rain is well beyond the capability of radiometric systems whose penetration is typically limited to optical depths around unity. This large dynamic range allows ATOMMS to retrieve water vapor from the mesosphere into the lower troposphere as its concentration varies by many orders of magnitude. It is also necessary to be able to retrieve water vapor when there is increased attenuation from clouds. The stronger 183 GHz line is used at higher altitudes and the weaker 22 GHz line is used at lower altitudes. A design goal for ATOMMS is to have sufficient dynamic range to achieve a large vertical overlap of the High and Low-Band measurements and retrieved profiles. A vertical overlap will provide a valuable crosscheck since the errors in the Low-Band and High-Band systems will be largely independent. The two bands will have different dependencies and sensitivities to turbulence and spectroscopic uncertainty. In the vertical overlap region the observable High-Band frequencies will be far from line center, while the information from the Low-Band signals will be from frequencies closer to line center.

A fundamental goal for weather and climate monitoring, prediction and understanding is all-weather unbiased global sampling. IR systems have substantial biases in their coverage due to the limited ability of IR photons to penetrate through clouds (e.g., Hearty et al., 2013) and its ~2 km vertical resolution is poor in comparison to the verticals scales at which water varies in the atmosphere. While downward-viewing passive microwave systems penetrate through clouds, their vertical resolution is very coarse and their retrievals over land are significantly less accurate than over oceans. GPS RO does provide unbiased global coverage, but is limited by the inability to separate the wet and dry gas contributions to the index of refraction.

ATOMMS is much closer to an all-weather global remote sensing system that will minimize sampling biases. ATOMMS combines the self-calibration and vertical resolution advantages of occultation systems with relatively easy to interpret observations of signal attenuation through the atmosphere that can be inverted to produce accurate, high vertical resolution profiles of water vapor without a priori constraints. In contrast, passive IR and microwave systems require technically challenging measurements of absolute radiance in orbit, which are fundamentally more difficult to interpret and retrievals of water vapor are more uncertain, vertically coarse, and require a priori constraints. An orbiting ATOMMS system achieves near-absolute, long term stability for climate monitoring simply by measuring *changes* in amplitude over the 100 s duration of LEO-LEO occultations.

Given this present situation, ATOMMS' precise, all-weather retrieval capability, as demonstrated here, would achieve a major advance in remote sensing of the atmosphere. These results support the prediction that an ATOMMS system in LEO would be a major advance toward achieving the fundamental satellite observing system goals of very high vertical resolution, all-weather temperature and water vapor sounding with very small random and absolute uncertainties, across the entire globe in support of weather prediction, climate monitoring and the quantitative constraints on process needed to improve models. A mission design concept using a constellation of very small ATOMMS satellites using cubesat technology is given in Kursinski et al., 2016b. ATOMMS has the potential to provide global observations from space that approach, and in some ways exceed, the performance of sondes.

**Appendix A: In-Situ Observational Network Required for Validation of ATOMMS Retrievals**

We now discuss the question regarding the quality, quantity and spacing of in-situ observations that would be required to validate the ATOMMS retrievals of changes in vapor pressure with time, which we believe are accurate to within 1%. Chilled mirror hygrometers can reach accuracies of 1%, at least in the laboratory. However, when we discussed validating ATOMMS instruments to 1% with a chilled mirror hygrometer expert at NCAR, we were told that no in-situ measurements can reliably achieve 1% accuracy out in the field (Holger Vömel, personal communication). Chilled mirrors are also expensive. We purchased one for $9000 and even the less accurate miniature ones used on balloons are more than $1000 apiece. Therefore, while a series of chilled mirrors could be placed along the path, their accuracy might not be as good as required to achieve 1%. They would likely be the closest to 1% that is available.

The next consideration is how to satisfy the constraints imposed by the ATOMMS measurements which include (1) a raised observational path between the instruments sufficiently high above the ground surface to avoid surface reflections and (2) a sufficiently long path length to produce enough absorption to enable precise and accurate water vapor retrievals. To avoid contamination of the water vapor observations by the ground surface, the in-situ sensors must be located well above the surface (~50 m) and close to the signal path, but not so close that they interfere with the ATOMMS signal transmission.

Given the variability of the water vapor along the path, the next question is how closely must the in-situ instruments be spaced along the signal path to achieve a specified level of accuracy. We estimated the water vapor variability over the 5.4 km observation path by computing the root mean square (RMS) differences for the three available in-situ sensors during the experiment, namely the two sensors at each end of the observation path and data from a sensor in the town of Summerhaven in the valley below the observation path. The RMS of the differences between the three in-situ sensors and the ATOMMS measurements was approximately 8% during the period from 14:00 to 15:30, which preceded the first period of heavy rain.

To determine how many in-situ sensors would be required to achieve 1% agreement, we turn to the results of Otarola et al. (2011) who used aircraft measurements to determine how the ratio of the standard deviation of humidity point measurements divided by the path averaged humidity varies with the path length over which the point measurements are averaged. The Otarola et al. (2011) findings are shown in Fig. A1. The straight line segments in the figure represent power law type behavior. The power law exponent of the lines of $std(q)/mean(q)$ in Figure 9 that pass near the point of stdev/mean = 8% for a path of 5 km is approximately 0.35. Given this power-law exponent and the requirement to keep uncertainties smaller than 1%, the path length required to achieve $std(q)/mean(q) = 1\%$ is approximately 10 m. This result is shown graphically in Fig. A1 by the dashed blue line that passes through the ATOMMS conditions of stdev/mean = 8% for a path of 5 km.

Thus, in situ sensors, accurate to 1% each, would need to be placed every 10 m along a 5.4 km path to achieve an in situ-based path average consistent with the ATOMMS measurements to the 1% level. This would require approximately 400

total in-situ instruments, a very large number of laboratory quality sensors.  It would be difficult, if not impossible, to locate these sensors close enough to the signal propagation path without interfering with the signal itself.  Furthermore, if the water vapor variations during the heavy rainfall were still larger than the 8% variations preceding the heavy rainfall, then still denser in-situ sampling would be required.

This immediately raises the question of whether one could actually develop, deploy, operate, maintain and protect such a large number of instruments along an elevated path during the kind of severe weather that was required to achieve the high opacities that were observed.  We considered using one or more unmanned aerial vehicles (UAV) carrying precise humidity, temperature and pressure sensors making measurements along the path during the ATOMMS measurements.  This solution has advantages of flexibility and relatively low cost, but it is not clear that any existing UAV humidity instrumentation

can meet our performance needs.  Furthermore, the biggest problem with an UAV approach is simply that the UAVs may not survive the intense convective activity that produced the high optical depths observed during our experiment.

        We also considered deploying a series of tethered balloons along the 5.4 km path.  However, the problem again is that during intense convective activity, with heavy rain, lighting, severe winds and downdrafts, the balloons would have been dangerous, potentially starting fires when struck by lightning, with at least a subset being destroyed, and the likelihood that

the measurement accuracy required to validate ATOMMS would have been low.  Given that sonde humidity sensors are notorious for getting wet during rain which yields positively biased humidity during and following rain, just the rainfall itself would likely have degraded the balloons' measurement accuracy.

        We discussed using instrumented towers with experts at NCAR, with experience deploying in situ sensors for field experiments.  Towers appear to offer the approach most likely capable of successful, accurate measurements aloft during such

extreme weather conditions.  However, issues of safety for both the instruments and personnel and environment remain as the towers would certainly act as lightning rods, with the potential to start fires.  Furthermore, purchasing and deploying the hundreds of towers of sufficient height required to achieve confirmation at 1% would be quite expensive.

        Assuming an approximate cost of $2,500 per chilled mirror hygrometer, 400 such instruments would cost one million dollars.  Each would require a data collection system and should be monitored somehow during data collection.  The

25     instruments would then need to be placed at the altitude of the ATOMMS signal path where they would have to be protected from heavy rain, winds and lightning.  It is also not clear how many personnel would be required to implement, maintain and operate such an array.

        The point of the preceding discussion is that verification by in situ measurements at the level of 1% uncertainty achieved by the ATOMMS measurements and retrievals out in the field is very difficult (if even possible).  As noted, we have

30     not yet identified any practical, cost-effective way to make a sufficient number of *in-situ* observations along the beam path that could have been used to evaluate the ATOMMS retrievals at their level of 1% precision during periods of intense convection.

**Acknowledgments**

We want to thank Jeff Kingsley for his support in making critical resources available at the University of Arizona's Steward Observatory needed to complete the ATOMMS instrumentation, and Chris Walker for sharing the Steward Observatory Radio Astronomy Laboratory (SORAL) facilities with us during development of the prototype ATOMMS instrument. We also want to thank Jim Grantham for providing access and modifications to the Mt Bigelow and Mt Lemmon facilities to support these observations. We thank David Adams and two anonymous reviewers whose constructive criticism improved the presentation of this paper considerably. This work was supported by the National Science Foundation Major Research Instrumentation (MRI) Program grant 0723239 and the National Science Foundation, Division of Atmospheric and Geospace Sciences (GEO/AGS) grants 0946411 and 1313563. In particular, we want to thank Jay Fein, program scientist and manager at NSF, who passed away in 2016. Without Jay's insight and relentless effort and support, this research would never have been funded and taken place.

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

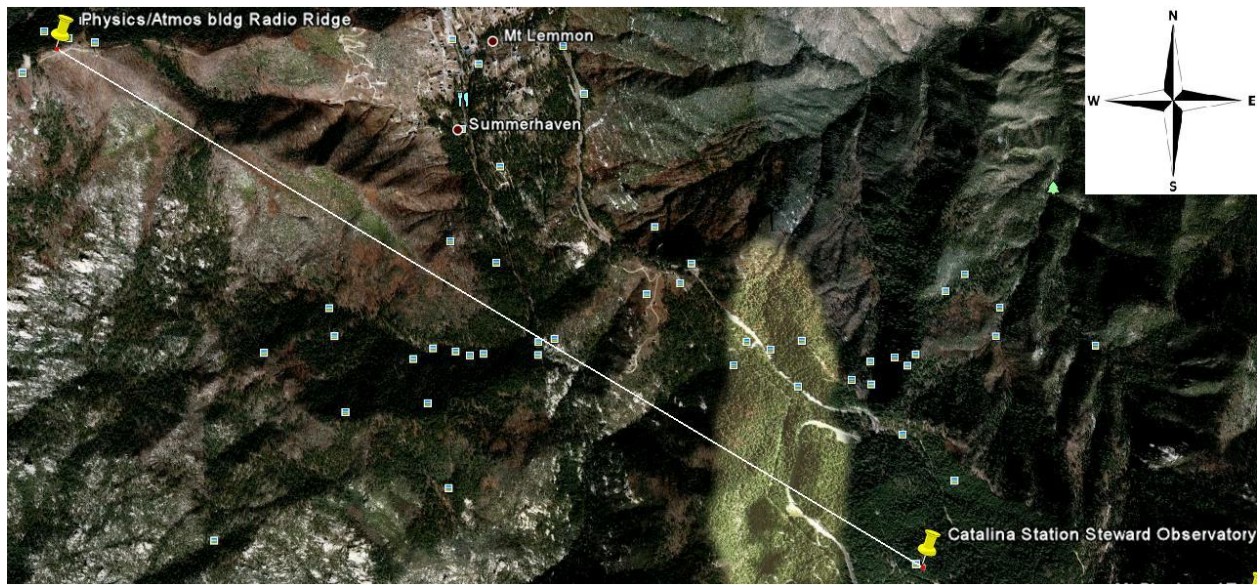

**Figure 1: Geometry for the ATOMMS ground-based prototype instrument tests. The High-Band transmitter was located on Radio Ridge near Mt. Lemmon at an altitude of 2752 m, and the High-Band receiver was located 5.4 km away at the Catalina Station Observatory near Mt. Bigelow at an altitude of 2515 m. The signal propagation path lies along a northwest to southeast line.**

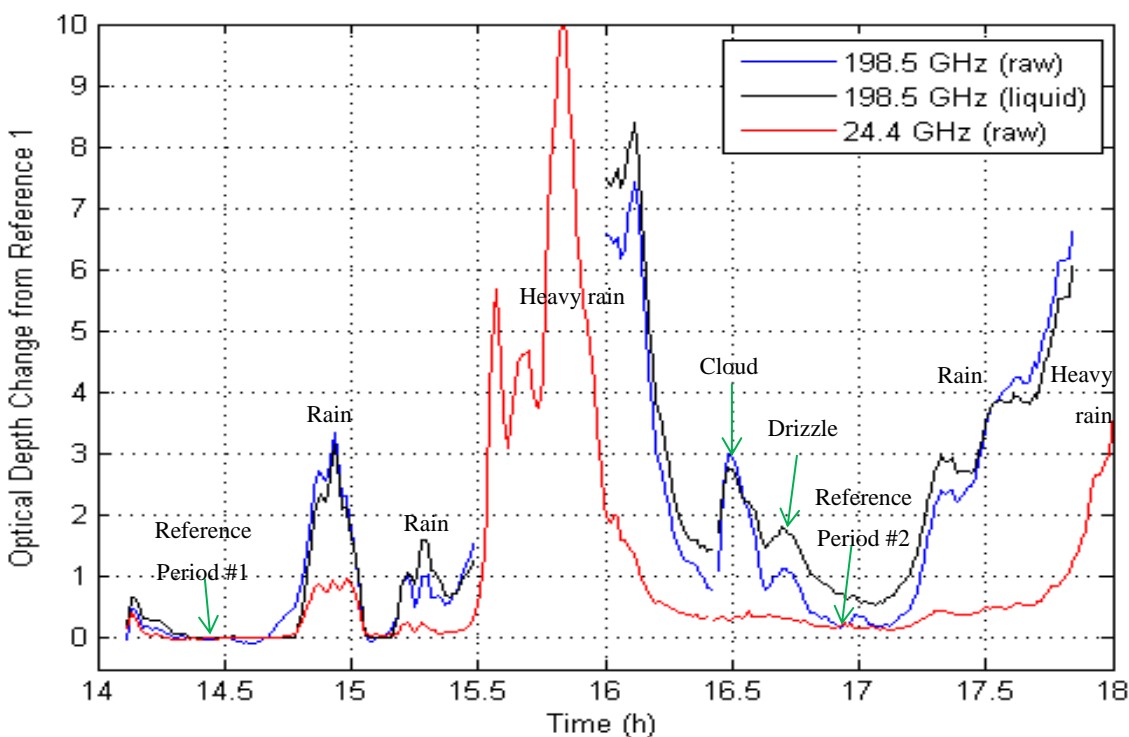

**Figure 2: Blue and red lines show observed changes in optical depth at 198.5 GHz and 24.4 GHz relative to reference period 1. The black line shows changes in optical depth at 198.5 GHz due to changes in liquid water after removing the contribution from changes in vapor pressure and temperature.**

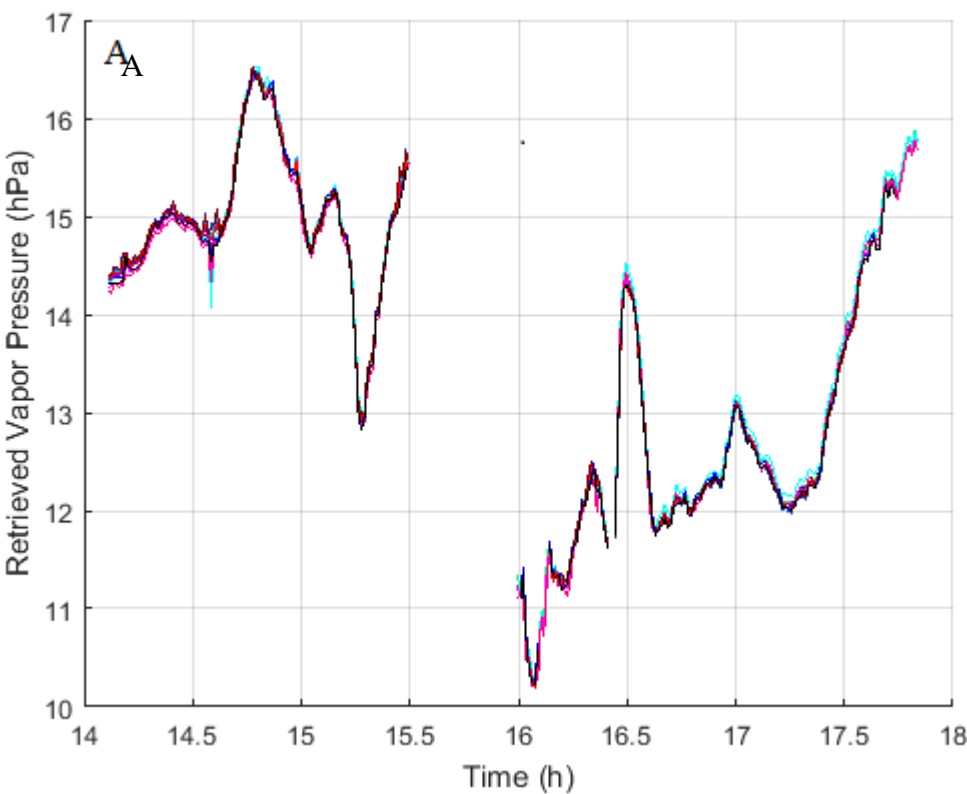

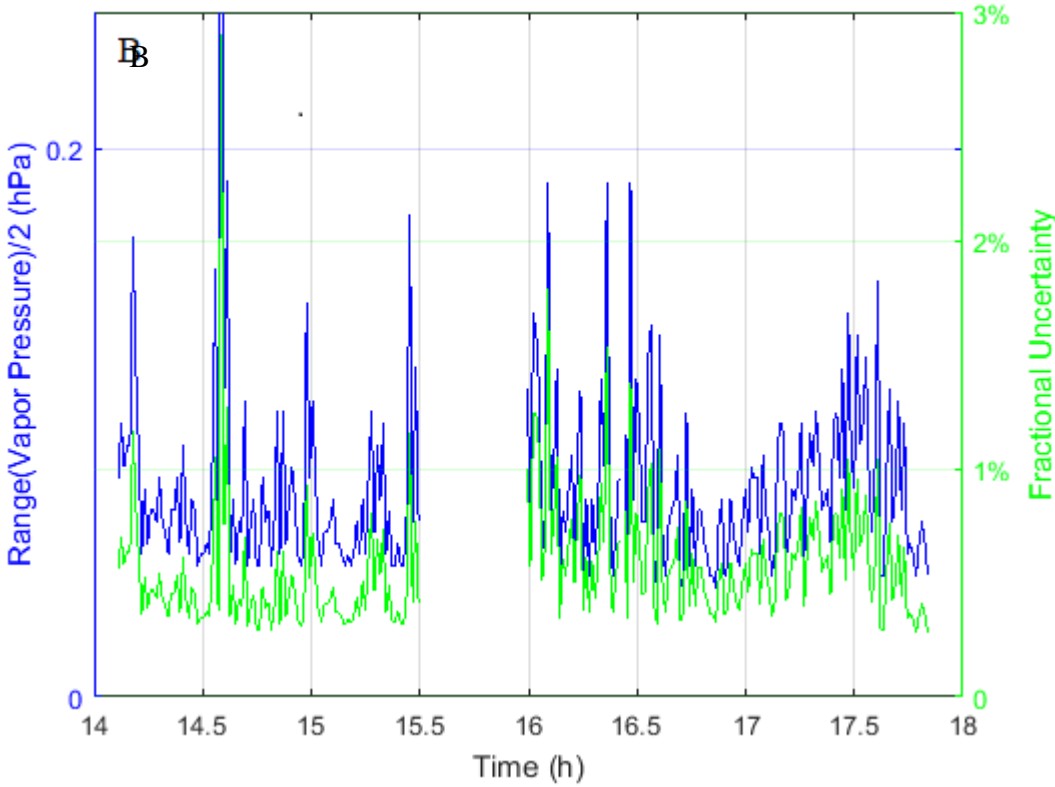

**Figure 3: A. Retrieved vapor pressure for the 12 retrieval test cases described in the text. Each line is a different color. B. Blue line and left axis indicate the half range, which is one half of the maximum minus minimum vapor pressure from the 12 retrieval cases; green line and right axis is the half range divided by the absolute vapor pressure at each retrieval point expressed in percent. The strong peaks near 14.6 hours are due to momentary noise in the calibration signal.**

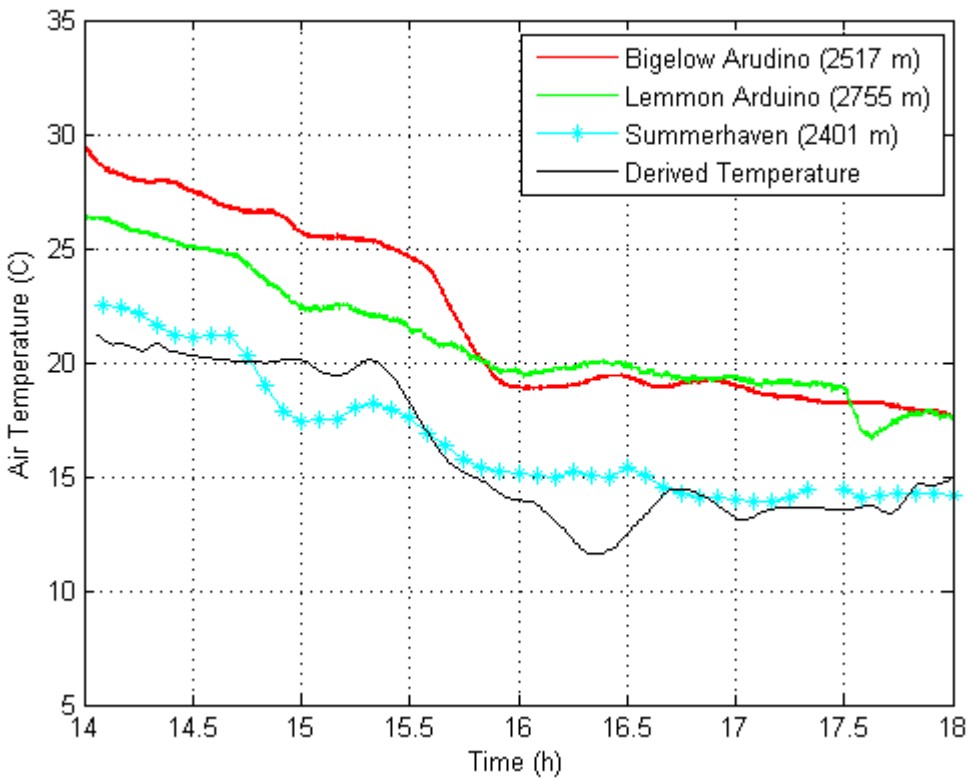

**Figure 4: Observed and derived air temperatures during the ATOMMS ground-based experiment.**

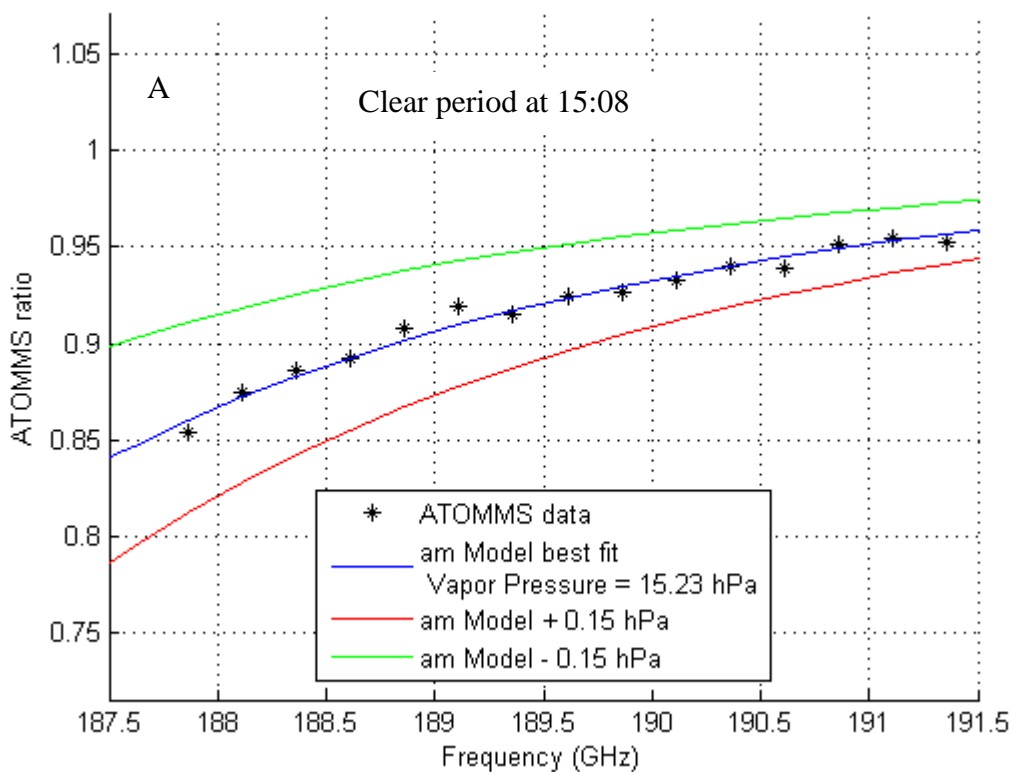

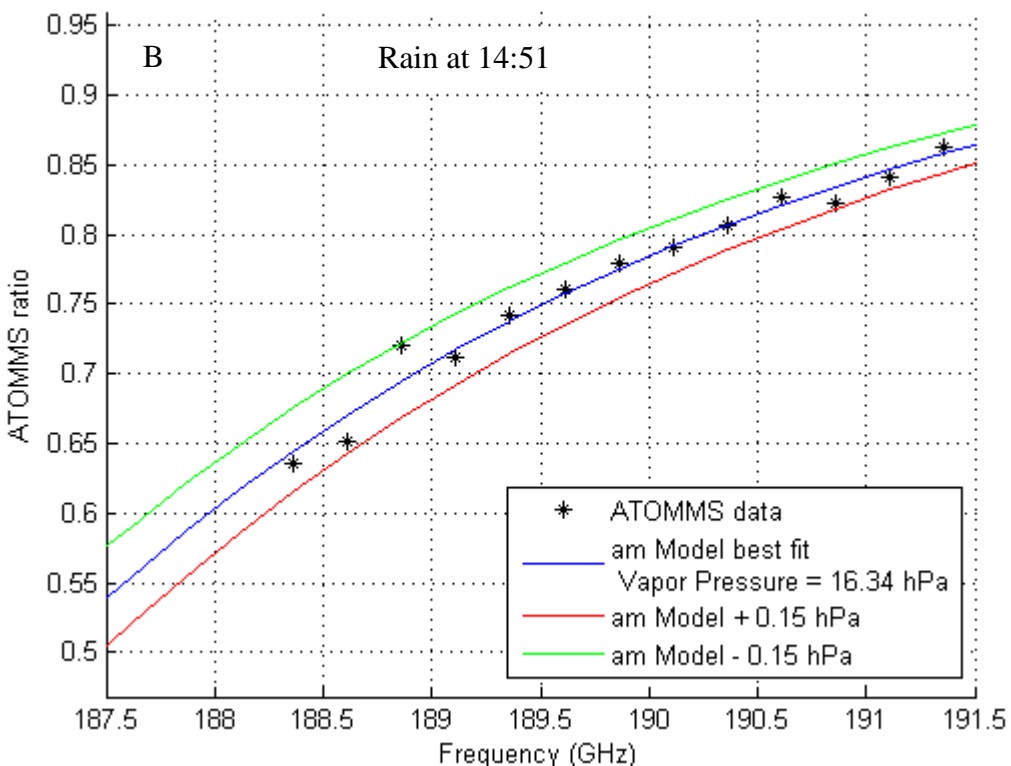

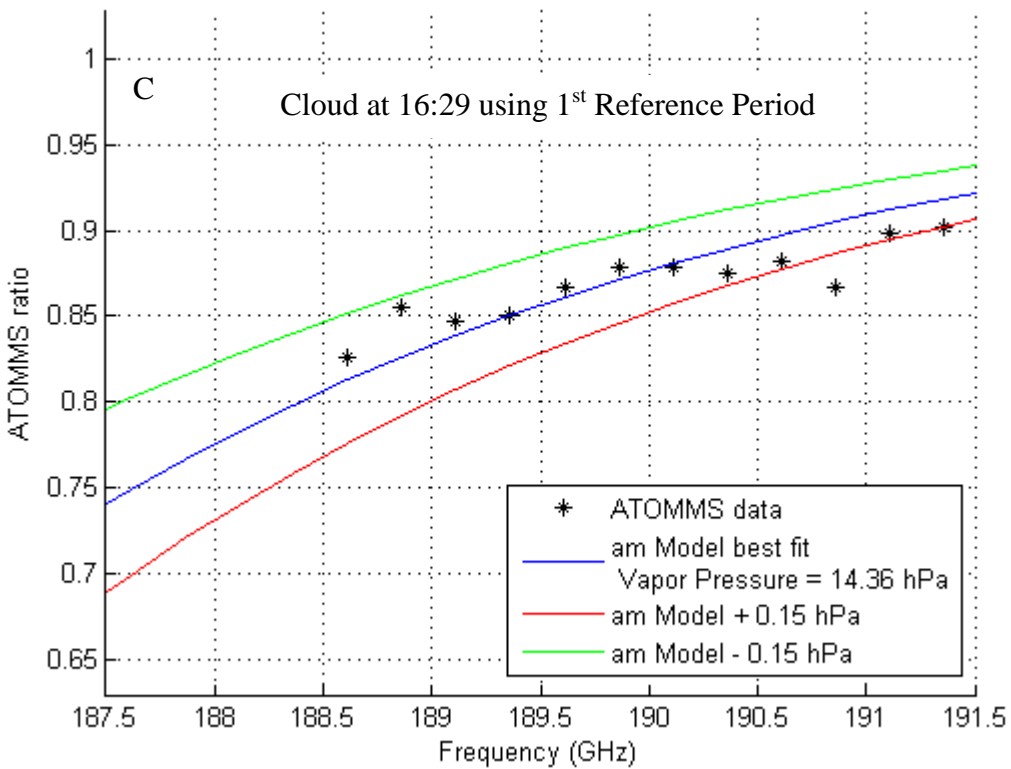

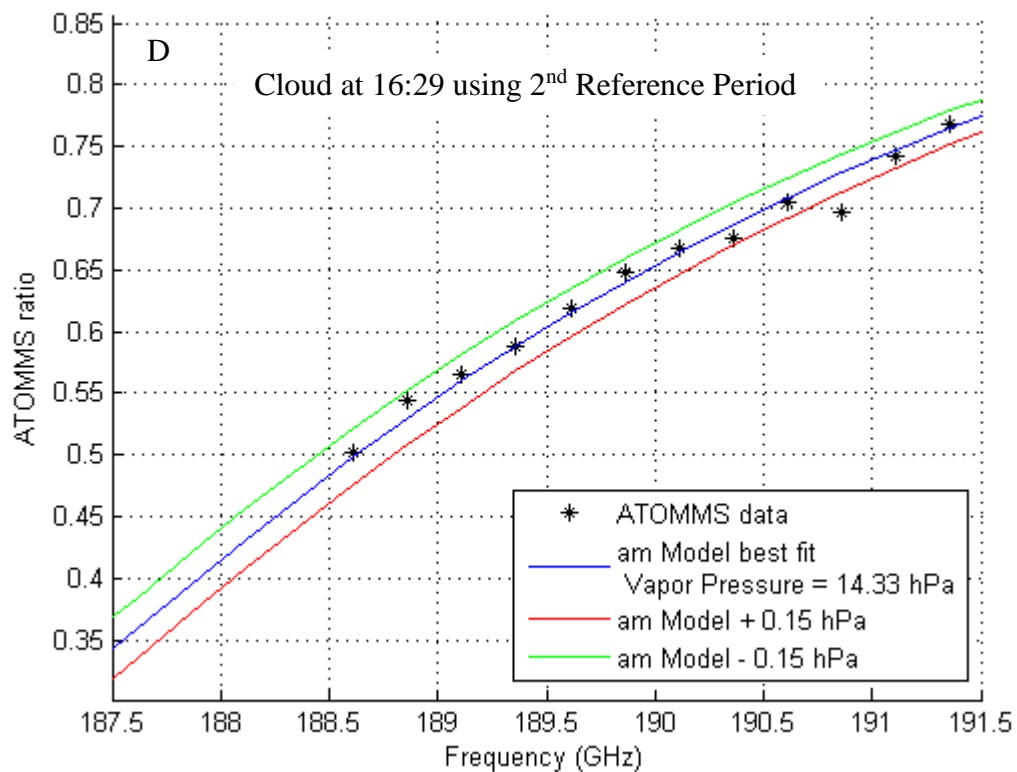

**Figure 5: Examples of fitting the observed ATOMMS amplitude ratio, Eq. (2) (black asterisks), to the forward calculated ATOMMS ratio using using *am7.2*. Blue line is the best fit line for the indicated vapor pressure. Red line is am forward calculation for a vapor pressure 0.15 hPa greater than the best fit vapor pressure. Green line is forward calculation for a vapor pressure 0.15 hPa less than the best fit vapor pressure. The solutions shown in panels A, B, and C used reference period 1, while the solution in panel D used reference period 2.**

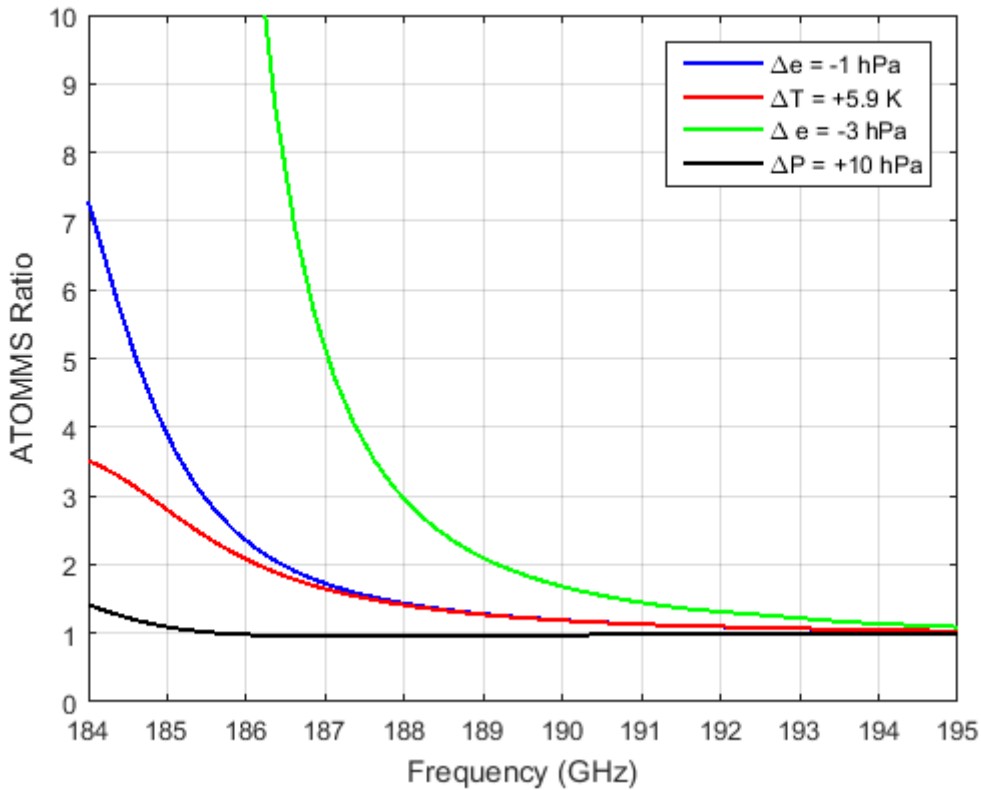

**Figure 6: ATOMMS ratio for four changes in the atmospheric conditions along the 5.4 km observation path relative to reference conditions: vapor pressure decreased by 1 hPa (blue), temperature increased by 5.9 K (red), vapor pressure decreased by 3 hPa (green), and air pressure increased by 10 hPa (black). The reference conditions were air pressure = 743 hPa, air temperature = 20° C, and vapor pressure = 15 hPa.**

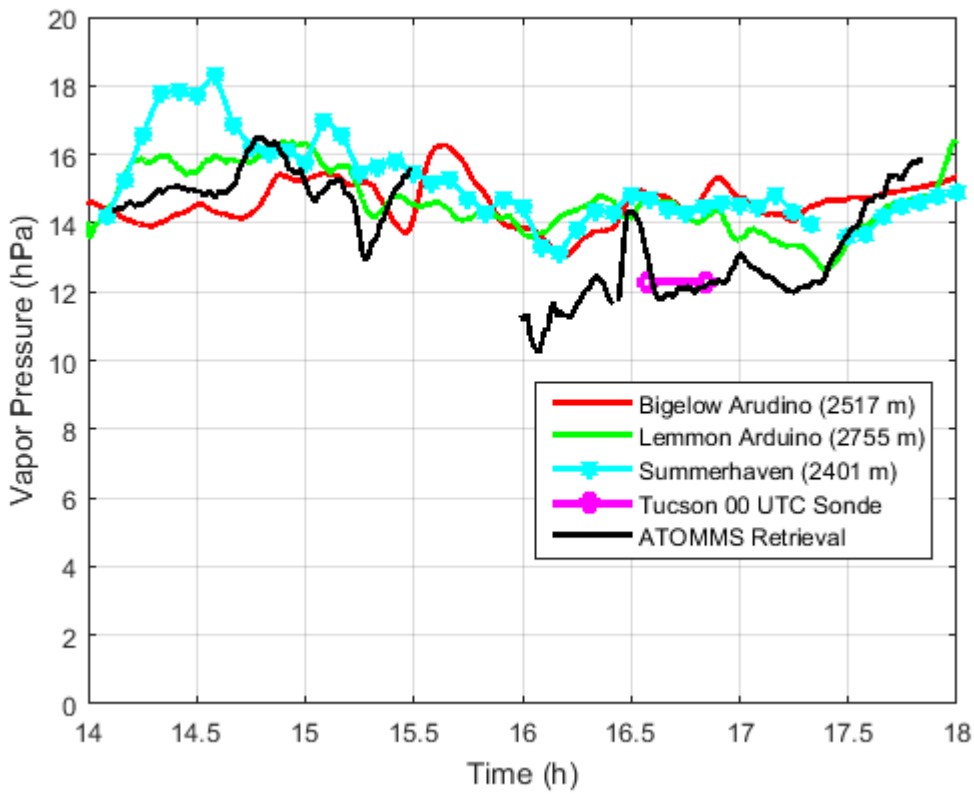

**Figure 7: Observed and retrieved vapor pressures. The sonde line indicates the average vapor pressure over the altitude range of the ATOMMS instruments as reported in the 00 UTC Tucson sonde for August 19.**

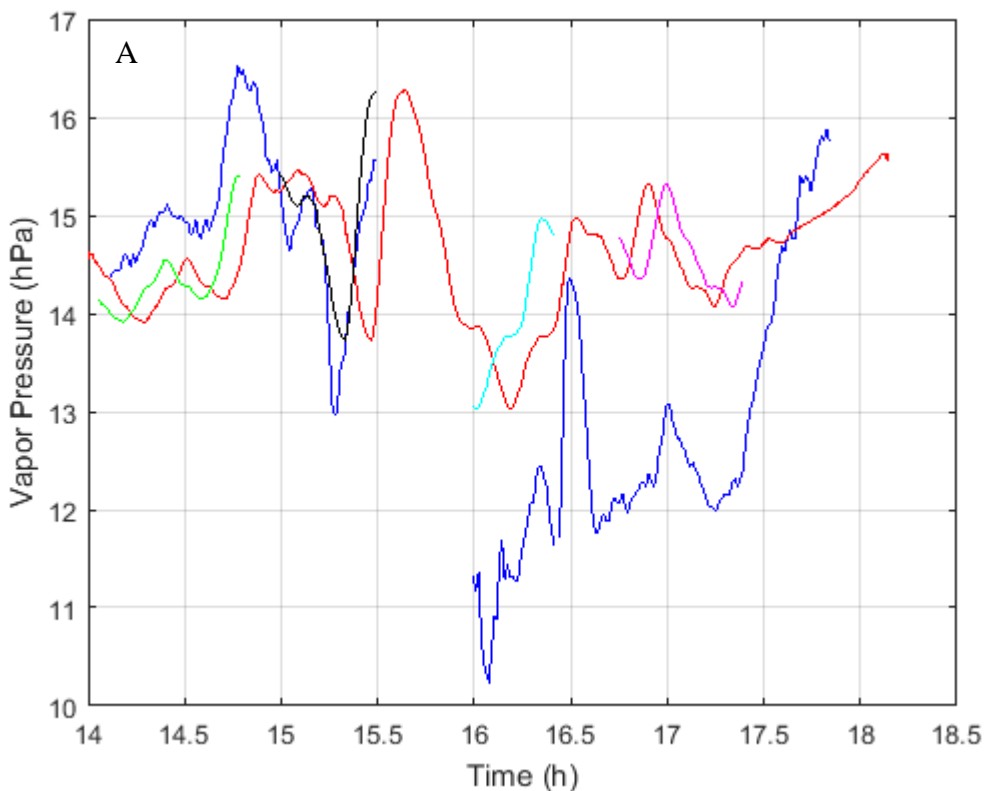

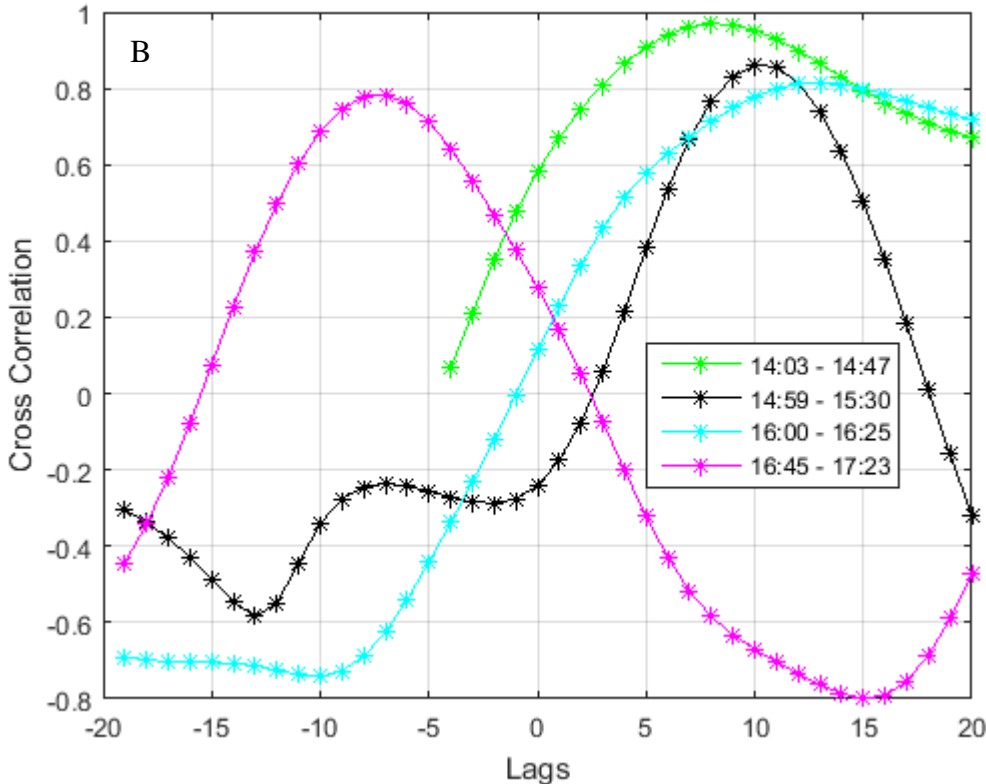

**Figure 8: A. Vapor pressure derived from ATOMMS observations (blue) and measured with an in-situ sensor on Mt. Bigelow (red). Also shown in other colors are four time segments of the in-situ observations shifted in time (as described in text) to highlight correlation between the two vapor pressure data sets. The time shift for each colored line is indicated in panel (B). B. Cross-correlation coefficients as a function of sample lags between the ATOMMS-derived vapor pressure and in-situ measurements of water vapor taken on Mt. Bigelow. The four lines correspond with the four time segments described in the text: green (14:03 – 14.47 hours), black (14:59 to 15:30 hours), cyan (16:00 to 16:25 hours), and magenta (16:45 to 17:23 hours).**

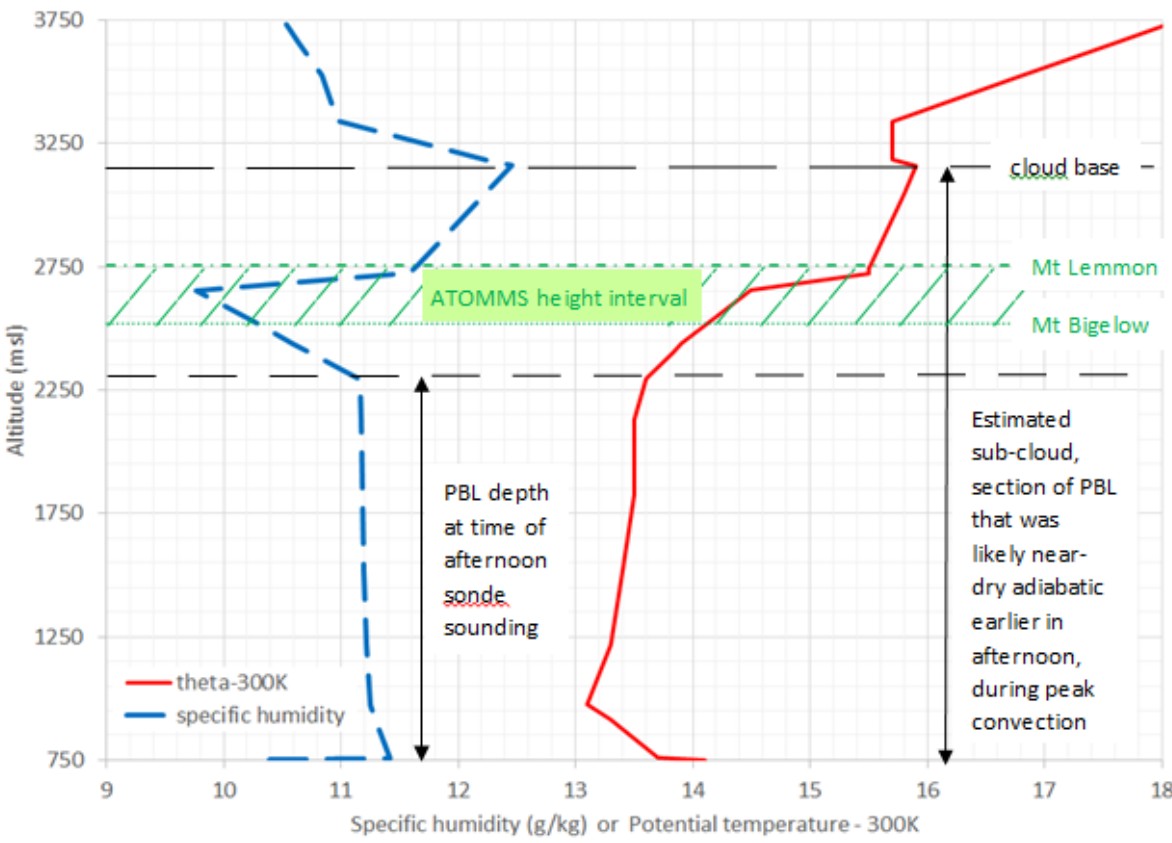

**Figure 9: Vertical profiles of specific humidity and potential temperature minus 300 K calculated from the 00 UTC Tucson sonde. The local time of the sonde launch was approximately 16:30 on August 18. Theta label for the red line stands for potential temperature and PBL stands for planetary boundary layer.**

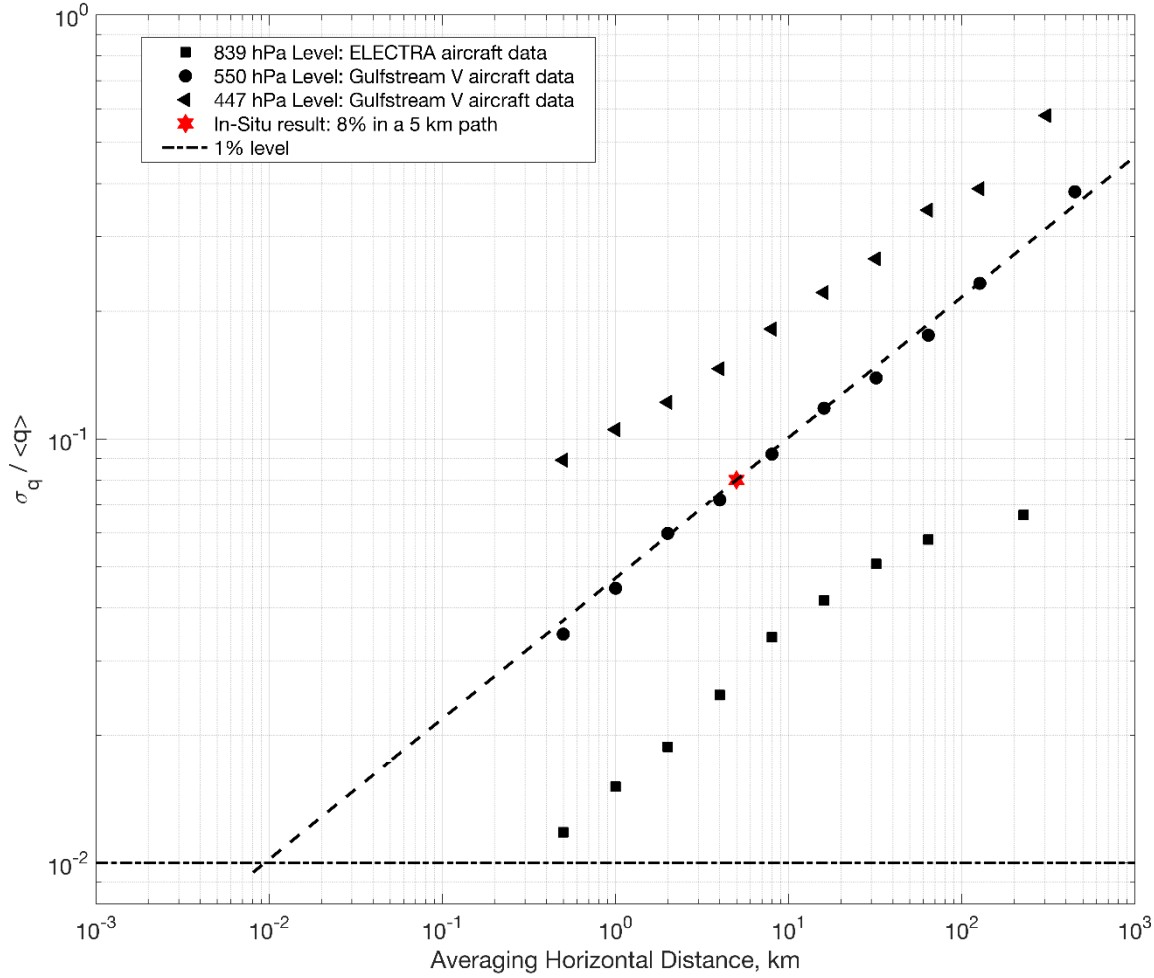

**Figure A1: Ratio of the standard deviation of absolute humidity to the mean absolute humidity based on aircraft data taken at different altitudes, which is indicated by the air pressure along different flight paths. The red star on the dashed line constructed for the 550 hPa altitude observations corresponds with the value calculated from the three in-situ sensors operating during the ATOMMS mountaintop experiment (ratio of 8% for a 5 km path). The slope of the dashed line corresponds with a power law exponent of 0.35 for the dependence of std(q)/mean(q) with the length of the path, which is consistent with Kolomogorov turbulence. Extrapolation of this line to a std(q)/mean(q) value equal to 1% indicates that in-situ observations are required every 10 m in order to validate the 1% accuracy of the ATOMMS retrievals. Adapted from Otarola et al. (2011).**