# Peer review of "Retrieval of Water Vapor using Ground-based Observations from a Prototype ATOMMS Active cm- and mm-Wavelength Occultation Instrument"

_Atmospheric Measurement Techniques, 2017_

## Referee Comment (RC1) · Anonymous Referee #1 · 12 Sep 2017

Comments on:

"Retrieval of water vapor using ground-based observations from a prototype ATOMMS cm- and mm- wavelength occultation instrument ", by D.M.Ward et al., version dated 26 July 2017, submitted to Atmos. Meas. Tech. Discussions.

General comments

1. This paper extends studies in preparation for a new satellite radio occultation technique to profile atmospheric humidity. In order to explore some aspects of the deploy-

ment of this technique between two LEO satellites, the paper describes implementation of the same technique with transmitter and receiver on two mountain tops.

2. It is a well-conceived experiment and the ideas and results are interesting. The paper is generally well written. The results are presented concisely but adequately. The discussions of the results and of the future potential of this technology are careful and persuasive.

3. There is some problem with Figure 2 – the annotation referred to in the text is missing.

4. I recommend publication subject to minor revision to address the points below.

Specific comments

5. p.1, l.24: "precision" and "absolute accuracy". The words "precision" and "precise" are used here and in other places, and it is not clear whether the usage is technical or just general. Also, nowadays, "uncertainty" is usually preferred for "accuracy". So in this case, does "precision" mean "random uncertainty" and does "absolute accuracy" mean "systematic uncertainty"? If so, I suggest to re-write this sentence to clarify this, and to review all other occurrences of "precise" "precision" and "accuracy" throughout.

6. p.1, l.25: "constraining processes". This is too brief to be clear. It is well explained on p.2, l.1.

7. p.1, l.28: "precise". See comment 5 above.

8. p.2, l.32: "precision". See comment 5 above.

9. p.4, l.1: "precise". See comment 5 above.

10. p.4, l.3-4: ". . . attenuation . . . distributed along the path . . . In contrast, . . .". Although this problem is more acute for passive radiometry, it is also present for RO – the attenuation is also distributed along the path and it is not possible to say where exactly along the path it takes place, even in the full RO retrieval context. So "In contrast"

is too strong.

11. p.4, l.8: "in terms of amplitude rather than intensity". Does this explain the factor of $\frac{1}{2}$ in eq.(1)? It may be worth pointing this out. Otherwise it looks like a rather unconventional definition of optical depth.

12. p.6, l.12-26. It would be helpful to say something here about another geometric difference: in this experiment, all the attenuation takes places at $\sim$2600 m in height whereas, even in a single LEO-LEO measurement, it takes place over a range of altitudes characteristic of limb-viewing geometry.

13. p.7, l.2-3: "as indicated by the annotations in Fig.2". The intended annotations in Fig.2 appear to be missing. See comment 3 above. Similarly, "the First Reference period" (p.2, l.8) is not clear and presumably is intended to be an annotation on Fig.2.

14. p.7, l.21 and l.23: 4:30 –> 16:30?

15. p.7, l.29: "as the calibration tone". Do you mean fCAL, which you call the "calibration signal" on p.4, l.20? It would be helpful to standardise references to fCAL, calibration signal, calibration frequency and calibration tone throughout.

16. p.8, l.1-5. The description of the processing is very compressed and it is not possible for the reader, from this alone, to understand how the processing is done. Can you give a reference to a more complete description?

17. p.8, l.15: "are identified in Fig.2". See comment 13 above.

18. p.8, l.22 and elsewhere: "the am Atmospheric Model". This is not clear; is the name of this forward model the "Atmospheric Model", abbreviated as "am". Later it is referred to as "AM" sometimes with a version number. Please make all these references consistent.

19. p.9, l.10-11: "12 different solutions". At this point the 12 solutions have not been mentioned or explained. Please refer forwards to section 5 for this description.

19. p.10, l.4-20. This is a very ingenious solution to the problem – nice piece of work! The only question remaining at this point in the reader's mind concerns the inherent uncertainties in this method. You discuss this later in section 5, and so I suggest here you refer forward to this discussion.

20. p.12, l.6 and l.26: "AM7.2" and "am". See comment 18 above.

21. p.18, l.7-13. In addition to these points, it is worth mentioning that GPS RO loses sensitivity to humidity, typically from the mid-troposphere upwards (at a height dependent on absolute humidity and hence temperature) because of the relatively low absorption coefficient of water vapour at GPS frequencies.

22. p.18, l.1 and l.23: "achieves" –> "would achieve"? "offers" –>"would offer"?

23. p.19, l.4-6. It would be more conventional to write this sentence as a simple statement with reference "(Holger Vömel, personal communication)".

24. p.19-20. In addition to the arguments presented in this Appendix, I think there is an additional one. If the typical scales that need to be measured are ∼13m, then any in situ observing system would need to be so close to the axis of the ground-based remote sensing system that it would interfere with the measurement. Otherwise it is not measuring the same path.

25. Fig.2. See comments 3 and 13 above.

Editorial points

26. p.1, l.2. Title: "mm- wavelength" –> "mm-wavelength"?

27. p.3, l.16. "Uncertainty" –> lower case?

28. p.11, l.20. "," after "seconds"?

29. p.16, l.30: "sought after" –> "sought-after"?

---

## Author Comment (AC1) · 24 Oct 2017

Responses to Comments from Anonymous Referee #1

First, we wish to thank this reviewer for the helpful review. Our responses are indicated below and numbered as they were by the reviewer.

General Comments:

3. It looks like the wrong figure was uploaded as it is missing the annotations. Below is the figure 2 that should have been included with the paper. Figure 2 in the original document has been changed to the figure below.

**Figure 2**

[Figure]

**Figure 2. Blue and red lines show observed changes in optical depth at 198.5 GHz and 24.4 GHz relative to reference period 1. The black line shows changes in optical depth at 198.5 GHz due to changes in liquid water after removing the contribution from changes in vapor pressure and temperature.**

Specific Comments:

5. In the manuscript we use "precision" to mean "random uncertainty" and "accuracy" to mean "systematic uncertainty." We have made the following change.

Page 1, line 24. The word "precision" will be changed to "random uncertainty"

6. The following sentences were added at the end of the abstract:

ATOMMS' water vapor retrievals from orbit will not be biased by climatological or first guess constraints, will be capable of capturing nearly the full range of variability through the atmosphere and around the globe in both clear and cloudy conditions, and will therefore greatly improving our understanding and analysis of water vapor. This information can be used to improve weather and climate models through constraints on and refinement of processes affecting and affected by water vapor.

7. Change the wording of the sentence beginning on page 1, line 28 to

"Despite its importance, our observations of its distribution in the atmosphere, and its trend with time, as well as our understanding of the factors controlling these are limited [Sherwood et al., 2013]."

This eliminates use of the word "precise."

8. Change the wording of two sentences in the last paragraph on page 2. The sentence beginning on page 2, line 26 contained the word "precisely" in line 27. Eliminate "precisely" by changing that sentence to

"Profiling both the speed of light like GPS RO as well as the absorption of light, which GPS RO does not measure, enables ATOMMS to profile temperature, pressure and water vapor simultaneously from near the surface to the mesopause with little random or systematic uncertainties (Kursinski et al., 2002)."

The sentence beginning on page 2, line 31 contained the word "precision." Eliminate the word precision by changing that sentence to

"Kursinski et al. (2002) found that such a system could provide water vapor retrievals with a random uncertainty of 1 – 3% from near the surface well into the mesosphere."

9. We believe the word "precise" is appropriate as it is used on page 4, line 1. However, we do believe a wording change is warranted for "precise" on page 5, line 3.

The sentence beginning on page 5, line 3 contained the word "precise." Eliminate use of "precise" by changing that sentence to

"This ratio of ratios approach enables measurement of water vapor in the presence of clouds and rain with very small random and systematic uncertainty as we demonstrate below.

10. In order to address this concern, several wording changes are proposed within the paragraph that begins on page 4, line 1. The new wording:

"ATOMMS functions as a precise, active spectrometer over the propagation path between the transmitter and receiver.  Retrievals of water vapor from radiance measurements are inherently ambiguous because both the signal source emission and attenuation along the path are unknown and must be solved for, creating an ill-posed problem (e.g., Rodgers, 2000).  In comparison to radiance retrievals, ATOMMS has the advantage that the transmitted signal strength is well known and the observed quantity is simply the attenuation along the path, which makes the retrievals much more direct and less ambiguous.  The active approach also enables retrievals with small random and systematic uncertainty under conditions of large path optical depths, which is not possible for passive retrievals."

11. Add the following sentence on line 11 of page 4. "The factor of ½ multiplying the optical depth comes about because intensity is proportional to amplitude squared."

12. Four wording changes have been made to address this comment.

(a) Add the following to the sentence after the sentence ending on page 5, line 27.

"The attenuation contributed at higher altitudes along the ray path due to both the limb sounding geometry and the exponential decay in water vapor concentrations with altitude."

(b) Change the sentence that begins on page 5, line 27.

Original Sentence: "We note that the Abel transform isolates the contributions of these layers."

New Sentence: "We note that the Abel transform isolates the contribution from the lowest altitude portion of the signal path."

(c) Change the sentence that begins on page 5, line 28.

Original Sentence: "For a vertical resolution of 100 m, the horizontal length of the lowest layer is approximately 70 km"

New Sentence: For a vertical resolution of 100 m, the horizontal length of the path through the lowest layer is approximately 70 km"

(d) Change the wording of the sentence that begins on page 6, line 12.

Original Sentence: "In this mountaintop demonstration, the atmospheric path from transmitter to receiver was only 5.4 km, such that the water vapor attenuation due to absorption by the weak 22 GHz line was too small to measure accurately."

New Wording (2 sentences): "In this mountaintop demonstration, the atmospheric path from transmitter to receiver took place over a narrow altitude range from 2752 m to 2515 m above sea level and was only 5.4 km in length.  Over this short path the water vapor attenuation due to absorption by the weak 22 GHz line was too small to measure accurately.

13. Replacing figure 2 eliminates this issue.

14. Our mistake. The time should be specified as 16:30 in both instances. Changes made.

15. To avoid confusion and for consistency, we have decided to use the term "calibration signal" throughout. The calibration signal amplitude at a particular selected frequency (typically in the wing of the absorption line) is used to remove or reduce unwanted amplitude variations before using the on line signal frequencies to estimate atmospheric absorption.

Here are all the instances of $f_{CAL}$, calibration frequency, and calibration tone that have been changed. The changes are shown below. If just a single word is changed, the change is indicated in blue.

page 3, line 11. "ATOMMS performance in cloud and rain is achieved via a differential transmission approach using a calibration tone (signal), in contrast to passive IR and microwave sensors systems that work via emission."

page 4, line 20.

Original sentence: "The frequency, $f$, of one signal is placed on the absorption line of interest while the frequency of the second signal, $f_{CAL}$, is farther from line center to function as a calibration signal."

Modified sentence. "The frequency, $f$, of one signal is placed on the absorption line of interest while the frequency of the second signal, $f_{CAL}$, is farther from line center to function as an amplitude calibration signal."

page 6, line 31. "198.5 GHz was the frequency of the High Band calibration tone (signal) during this experiment."

page 7, line 27. Several changes were made for clarification. This also partially addresses specific comment #16.

Original wording. "For this experiment, one transmitter swept through the tunable frequency range generating a tuned tone that was received by a receiver sweeping through the same tuning sequence. The other tone was fixed at 198.5 GHz in order to function as the calibration tone."

Revised wording. "For this mountaintop experiment, the frequency of the signal generated by one transmitter was swept through a tuning sequence that spanned the instrument's tunable frequency range. This signal was received by a narrowband heterodyne receiver whose second local oscillator was simultaneously swept through its matching tuning sequence. The frequency of the other signal was fixed at 198.5 GHz in order to function as the amplitude calibration signal for measuring differential absorption."

page 8, line 4.

Original sentence. "Calibration tone amplitudes were computed using the same method."

Modified sentence. "The calibration signal amplitudes were computed using the same method."

page 9, line 5.

Original sentence. "The liquid optical depth in Fig. 2 is the liquid optical depth at the calibration tone, $f_{CAL}$ = 198.5 GHz."

Modified sentence. "The liquid optical depth in Fig. 2 is the liquid optical depth measured by the calibration signal, $f_{CAL}$ = 198.5 GHz."

page 11, line 24. "These were reduced by almost an order of magnitude via amplitude ratioing with the calibration tone (signal) [Kursinski et al., 2016]."

page 13, line 3. "Ratioing of the amplitudes of two signals, as was done here, eliminates the effects of liquid particle extinction to the extent that the liquid extinction is spectrally flat over the ATOMMS tuning range and calibration frequencies." No change needed here.

page 13, starting on line 12. "This small 0.8% change in the retrieved vapor pressure provides some indication of how effective the calibration tone (signal) ratioing is in minimizing the sensitivity of the ATOMMS water vapor retrievals to hydrometeors.  In the future, the High Band system will have 4 rather than its present 2 tones (signals) in order to place calibration tones (signals) on both the low and high frequency sides of the 183 GHz water vapor line to reveal and compensate for any overall spectral tilt caused by particle extinction as well as other effects."

page 16, lines 32 and 33.

Original wording. "In terms of the number of signal frequencies required to accurately determine the water vapor, we used from 5 to 15 tuned signal frequencies plus a calibration frequency for the water vapor spectral fits.  The agreement and consistency of these results indicate that the amplitudes from just a few tuned frequencies and a calibration frequency are needed to produce water vapor retrievals with very small random and absolute uncertainties."

Revised wording. "In terms of the number of signal frequencies required to accurately determine the water vapor, we used from 5 to 15 tuned signal frequencies plus a calibration signal at a fixed frequency for the water vapor spectral fits.  The agreement and consistency of these results indicate that the amplitudes from just a few tuned frequencies and a fixed frequency amplitude calibration signal are needed to produce water vapor retrievals with very small random and absolute uncertainties."

page 17, line 18.

Original sentence. "The LEO version of ATOMMS will provide the information necessary to observe and account for such non-vapor effects using at least three simultaneous signal frequencies to place calibration tones (frequencies)  on both the low and high sides of the absorption line and the third frequency on the line."

Revised sentence. "The LEO version of ATOMMS will provide the information necessary to observe and account for such non-vapor effects using at least three simultaneous signal frequencies to place amplitude calibration signals on both the low and high sides of the absorption line and the third frequency on the line."

16. We propose to significantly change the wording under the heading "Signal Tuning and Detection," which begins on page 7, line 25. Major changes are shown in blue.

Original text. "The High Band portion of the ATOMMS ground-based prototype instrument simultaneously transmits and receives two continuous wave signals that are tunable from 181 to 206 GHz. For this experiment, one transmitter swept through the tunable frequency range generating a tuned tone that was received by a receiver sweeping through the same tuning sequence. The other tone was fixed at 198.5 GHz in order to function as the calibration tone. There were 122 tuning frequencies in the sweep, separated by 0.25 GHz, except for a gap between 191.5 and 193.5 GHz. This gap is due to the limited receiver response for Intermediate Frequencies (IF) less than one GHz and the first stage local oscillator (LO) being set to 192.5 GHz. This is likely the finest spectral resolution sampling of the 183 GHz line ever achieved in the field.

[revised manuscript text omitted]

Caption of figure 5, current text. "forward calculated ATOMMS ratio using the am Model, version 7.2."

Revised text. "forward calculated ATOMMS ratio using *am7.2*."

19. {first comment #19} We have referred to Section 5 already. The current text is "The retrieved path-averaged vapor pressure between the instruments is shown in Fig. 3A. The figure shows 12 different solutions that were used to estimate the random uncertainty in the retrieval of vapor pressure as described in Section 5."

The following change has been made for clarity: "The figure shows 12 different solutions that were used to estimate the random uncertainty in the retrieval of vapor pressure. The methodology used to compute the 12 solutions is described in Section 5."

19. {second comment #19}. This is already done on page 10, line 10, where it is stated "The uncertainty associated with this temperature estimation is discussed in Section 5."

To be more clear that sentence will be moved to the end of the last paragraph of the section "Determining Temperature," which is page 10, line 20 in the original document.

20. Changes specified in response to comment #18 above.

21. We respectfully disagree with this comment. GPS has very little sensitivity to water vapor via absorption because its frequencies are so low in comparison to the 22 GHz water absorption line. The sensitivity that GPS does have to water vapor is via the propagation delay due to the refractivity of water vapor which is significant basically at temperatures above 240 K (because there is enough water vapor present to see this effect in individual profiles). However, one cannot isolate that water vapor signature directly from the GPS observations alone. One must add additional constraints on the dry part of the refractivity to isolate the wet part of the refractivity buried in the GPS refractivity estimates.

22. We like use of the word "achieve" on page 18, line 2. No change here.

On page 18, line 23, though, we like the reviewer's suggestion to change "offers" to "would achieve." Change made in document.

23. Thank you.

Original wording. "However, when we discussed validating ATOMMS instruments to 1% with Holger Vömel, a chilled mirror hygrometer expert at NCAR, he indicated that no in-situ measurements can reliably achieve 1% accuracy out in the field."

Revised wording. "However, when we discussed validating ATOMMS instruments to 1% with a chilled mirror hygrometer expert at NCAR, we were told that no in-situ measurements can reliably achieve 1% accuracy out in the field (Holger Vömel, personal communication)."

24. Thank you. This is a point worth mentioning. The following sentence has been added after the second sentence in the last paragraph that begins of page 19. "It would be difficult, if not impossible, to locate these sensors close enough to the signal propagation path without interfering with signal itself."

25. Figure 2 will be replaced with the annotated figure.

Editorial Points:

26. We agree. There should probably not be a space between mm- and wavelength. We will have to check with the editorial staff. Change was made to original document.

27. The word "Uncertainty" will be changed to lower case "uncertainty."

28. Comma will be added after the word "seconds" on page 11, line 20.

29. Thanks. A hyphen should be used. "sought after" will be replaced by "sought-after" Change made.

---

## Referee Comment (RC2) · Anonymous Referee #2 · 3 May 2018

Review comment of Anonymous Referee # 2 on:

Retrieval of Water Vapor using Ground-based Observations from a Prototype ATOMMS Active cm- and mm- Wavelength Occultation Instrument
Dale M. Ward et al.

General Comments:

The paper is a very interesting and indeed necessary contribution to investigate the atmosphere based on occultation measurement technique by exploiting the microwave capacity. It describes in detail the ground based experiment by detecting water vapour with microwave signals under clear too very turbulent conditions as pre-study to a microwave occultation satellite mission. Besides a detailed results description an error and validation discussion was done. The discussion in the appendix about the difficulty or even impossability to in-situ validate such an experiment rounds up the paper. I recommend the paper for publication with minor revisions.

In this paper the low band signal is only used to determine whether there are small water particles present or not but it apparently is not used for the retrieval of water vapor, which is done only for the high band signal (Fig. 2). It is though claimed that to derive the water vapor content under stormy conditions when the high band becomes opaque, the low band is needed but it is not done for this experiment.
Did you get results on the 8 fixed frequencies at the 22 GHz region to investigate the heavy rain situation at 15:30, since for such heavy weather situation sufficant attenuation occured?

Specific Comments:

p. 2, line 16—20: Please give a reference to the RO technique including the specifications/limits of this remote technique. Please give references for other studies on microwave occultation technique pre-studies too.

p. 3, line 710: Please give a reference to this statement.

p. 5, line 1—4: Did you do investigations in terms of distance of frequency to the calibration frequency and the remaining error if unwanted sources (scintillation, ..) of errors show a frequency dependency?

p. 5, line 7: If the demonstration of these key aspects are published please give a reference.

p. 5, line12: Please determine here already on wich of these locations the transmitter and receivers are located.

p. 7, line 8: What is the reference period exactly. For a faster understanding it would helpt too if the reference period is marked in Fig. 2.

p. 8, line 15: Please mark the second reference period in the Fig.2 too.

p. 8, line 22: Please give a clear acronym for your Atmospheric Model, version 7.2 e.g. AM7.2 and make the notation consistent in the entire document.

p. 9, line 5: Please give the color code in the text too you are using in the figures for a faster understanding. It is not explained in the text what „raw" in the figure label means? Why is the blue line not always higher then the black line? To my understanding the blue lines contains all atmospheric effects and the black line only the liquid optical depth part.

p. 9, line 19: Please give a reference to this publications.

p. 10, line 3: Why where the tempearature sensors so close to the ATOMMS instruments positioned? Why not in a seperate small tent, if protection due to heavy rainfall was needed, to avoide temperature bias due to lost heat by the ATOMMS instrument?

p. 10, line 19: Please use the color code of the graph when explaining the figures to gurantee no confusion with the graphs.

p. 10, line 19: How do the aire pressure graphs look like including the one hour running mean of the air pressure?

p. 11, line 25: Did you estimate the residual error due to turbulences?

p. 12, line 8: How did you get the value of -0.17 hPa/C? Could you give a short explanation to this?

p. 13, line 11: Please give a reference to the Mie cloud model you where applying?

p. 14, line 22: Do you have information/graphs on the wind speeds correlating with the shift in time too e.g. by using the radar data?

p. 15, line 2: Do you have a reference on the Tucson WSR-88 RADAR?

p. 15, line 20: What was the horizontal distance of the radiosond to the actual microwave path, when passing through the mountain height levels of Mt. Bigelow and Mt. Lemmon? Was the radiosond actually passing the microwave path or was it very close or even far away from the microwave path?

p. 19, line 25. Please explaine shortly the other colored lines according to the altitude levels. It is not clear why e.g. the black and red lines show a less high ratio than the green and pink lines and they show even different linear dependencies within the length of the integration path.

Fig. 1: Is it possible to mark the position of the radiosond when passing through the altitude level of the two mountains?

Typo Comments:

p. 2, line 9--10: I would recommend to be consistence with the entire paper to write everything in hPa than in mb.

p. 4, line 30: Please change hydrometers to hydrometeors.

p. 7, line 21: Please make the notation for the time consistence in the entire document and figures. Sometime 4:30, then 4,5 or even the 12 hour notation is used.

p. 10, line 25: The +-1% signe is underlined. Please remove the underline.

p. 14, line 24 – 27: Please make the time consistent. They do not agree with the given times in the legend of Fig. 8B.

p. 15, line 19: Please make the notation of the Tucson radiosond consistent with legend in Fig. 7.

Fig. 3A: Please use the consisten units e.g. hPa instead of mb and h instead of hours (in Fig 3A and 3B)

Fig. 3B: What is the strong peak at about 14.6? In the text % is used but in the figure you use fraction for the right y-axes. Please make it consistent.

Fig. 8B: Please make the time in the label, legend and text consistent. Please give units?

Fig. A1: Please remove the title of the plot or make the symbols for standard deviation and mean value consistent. Please correct the x-axes label.

---

## Referee Comment (RC3) · D. Adams (Referee) · 9 May 2018

Review Ward et al. 2018 by David K. Adams

Recommendation: Minor Revision

The authors provide a detailed overview of an experiment carried out employing an ATOMMS instrument prototype between two mountaintops in southern Arizona. The article intends to demonstrate the unique characteristics of this technique of transmitting and receiving signals near the 22 and 183 GHz water vapor absorption lines

along with other frequencies, thereby overcoming the limitation of the present satellite RO systems which cannot independently predict temperature and water vapor content. The results of this experiment are placed in the context of employing ATOMMS in future possible satellite occultation systems. Overall, the paper is well written and the technique fairly well-explained. With respect to the results of the experiment, there is necessarily some speculation as to atmospheric conditions in line of site measurement given the lack of precise local data sources, but this is the nature of all experiments and is not a weakness of the manuscript. Below

I make some general comments and seek clarification of a few issues, in addition to a few suggestions

on sentence structure and/or grammar.

Minor Comments.

Line 26 You can probably be a bit more emphatic here. Water is the most important greenhouse gas, critical in the energy balance, responsible for storms etc... Line 29 Should be Sherwood et al., (2010) And you should probably include a few more "big picture" type references related to water vapor in the climate system.

Line 30 Water vapor observations must be unbiased and capture the full range of variability in clear and cloudy conditions across the globe... This sentence is a bit awkward, it could be written in a more concise manner or turn into two sentences.

Page 2.

Satellite systems typically do not have sufficient temporal or spatial resolution to capture many of the important processes related to the distribution of water vapor (such as deep convection in the Tropics).

And if the satellite systems do have this appropriate temporal and spatial resolution (e.g. GOES water vapor channels), they only provide column water vapor and not its vertical structure. You should include a bit more detail in this paragraph to give greater

force to your proposed system.

Line 19 Can you be specific as to what you mean here by insensitivity "(2) its insensitivity to water vapor in the colder regions of the troposphere and above."

Line 28 I think you should write "It can also profile ozone ..."

Line 2 Probably not necessary to include this "...we developed with funding from NSF,..."

Line 6 Clarify what you are referring to here "..and the forward modeled water vapor spectra,.."

Line 10 Don't use contractions in formal writing. "...simply do not work."

Line 16 Write "Sources of uncertainty ..."

Line 30 Write Refractivity and "the" extinction coefficient (or write coefficients)

The hydrostatic assumption would be very dubious during deep convective activity.

Line 11. This is a bit unclear. "The gas phase optical depth is due to water vapor and dry air absorption, which introduces temperature and pressure dependence, and any attenuation due to hydrometers." You are saying the gas phase optical depth is also dependent up the presence of non-gas constituents like hydrometeors?

Again, not sure if this is necessary to state. "With funding from NSF,..."

Page 6.

Line 14 "ATOMMS High Band signals" should be

Page 7.

Line 13. Which radar data are you referring to? You need to clarify this point.

Line 13 Write "By 16:30, the rain was considerably lighter"

Line 33. Can you back this statement up with any citations or some references. "This is likely the finest spectral resolution sampling of the 183 GHz line ever achieved in the field."

Page 8. Line 22 Should this be capitalized AM

Page 9.

Line 30. What size of error should we expect given the use of local pressure measurements at each of the sites? That is, across the line of site, there should be some small variability of pressure given updrafts and downdrafts.

Page 11 Line 10 Probably more common "signal-to-noise ratio"

Line 23 Write "comparison with independent, in-situ moisture..."

Line 1-3 Maybe you could be a little bit more specific here referring to the map " between the mountaintops on which instruments sit, while in-situ sensors are located on the ground at each end of the observation path and another in a valley below the observation path."

Line 17. Such behavior where moisture at the surface varies little while air aloft becomes significantly drier following summertime thunderstorms is common in this region (e.g., Fig. 4 in Kursinski et al. [2008]).

You can probably find a few more references that describe theromdynamic conditions after T-storms during the NAM

Line 21 "The nearby Tucson radiosonde indeed indicated that..." With all of the reference in the paper to this sounding, you should include it in the figures.

Page 18 Frequency of observations will always be an issue to some extent with the RO technique, particularly when the scales are of the time and space scales need for weather prediction.
* * *

---

## Author Comment (AC3) · 31 Jul 2018

**Responses to Comments from Anonymous Referee #1**

First, we wish to thank this reviewer for their helpful review. We first copy the comments of the reviewer **(in bold)** with our response below. In cases where the text is modified, we show both the original text and the revised text, with the changes shown in blue.

**General Comments:**

**1. This paper extends studies in preparation for a new satellite radio occultation technique to profile atmospheric humidity. In order to explore some aspects of the deployment of this technique between two LEO satellites, the paper describes implementation of the same technique with transmitter and receiver on two mountain tops.**

No response required.

**2. It is a well-conceived experiment and the ideas and results are interesting. The paper is generally well written. The results are presented concisely but adequately. The discussions of the results and of the future potential of this technology are careful and persuasive.**

No response required.

**3. There is some problem with Figure 2 – the annotation referred to in the text is missing.**

It looks like the wrong figure was uploaded as it is missing the annotations. Below is the figure 2 that should have been included with the paper. Figure 2 in the original document will be changed to the figure below. No change to caption, but it is repeated below the figure.

[Figure]

**Figure 2. Blue and red lines show observed changes in optical depth at 198.5 GHz and 24.4 GHz relative to reference period 1. The black line shows changes in optical depth at 198.5 GHz due to changes in liquid water after removing the contribution from changes in vapor pressure and temperature.**

**4. I recommend publication subject to minor revision to address the points below.**

No response required.

**Specific Comments:**

**5. p.1, l.24: "precision" and "absolute accuracy". The words "precision" and "precise" are used here and in other places, and it is not clear whether the usage is technical or just general. Also, nowadays, "uncertainty" is usually preferred for "accuracy". So in this case, does "precision" mean "random uncertainty" and does "absolute accuracy" mean "systematic uncertainty"? If so, I suggest to re-write this sentence to clarify this, and to review all other occurrences of "precise" "precision" and "accuracy" throughout.**

In the manuscript we use "precision" to mean "random uncertainty" and "accuracy" to mean "systematic uncertainty." We have made the following change.

Page 1, line 24. The word "precision" will be changed to "random uncertainty"

Original Sentence: Using an ATOMMS instrument prototype between two mountaintops, we have demonstrated its ability to penetrate through water vapor, clouds and rain up to optical depths of 17 (7 orders of magnitude reduction in signal power) and still isolate the vapor absorption line spectrum to retrieve water vapor with a precision better than 1%.

Revised Sentence: Using an ATOMMS instrument prototype between two mountaintops, we have demonstrated its ability to penetrate through water vapor, clouds and rain up to optical depths of 17 (7 orders of magnitude reduction in signal power) and still isolate the vapor absorption line spectrum to retrieve water vapor with a random uncertainty less than 1%.

**6. p.1, l.25: "constraining processes". This is too brief to be clear. It is well explained on p.2, l.1.**

The following sentences will be added at the end of the abstract:

ATOMMS' water vapor retrievals from orbit will not be biased by climatological or first guess constraints, and will be capable of capturing nearly the full range of variability through the atmosphere and around the globe, in both clear and cloudy conditions, and will therefore greatly improve our understanding and analysis of water vapor. This information can be used to improve weather and climate models through constraints on and refinement of processes affecting and affected by water vapor.

**7. p.1, l.28: "precise". See comment 5 above.**

This sentence has been eliminated based on the changes spurred by the comment from Reviewer 3, referring to line 26 of page 1.

**8. p.2, l.32: "precision". See comment 5 above**

We change the wording of two sentences in the last paragraph on page 2. The sentence beginning on page 2, line 26 contained the word "precisely" in line 27. Eliminate "precisely" by changing that sentence as follows

Original sentence: Profiling both the speed of light like GPS RO as well as the absorption of light, which GPS RO does not measure, enables ATOMMS to precisely profile temperature, pressure and water vapor simultaneously from near the surface to the mesopause (Kursinski et al., 2002).

Revised sentence: Profiling both the speed of light like GPS RO as well as the absorption of light, which GPS RO does not measure, enables ATOMMS to profile temperature, pressure and water vapor simultaneously from near the surface to the mesopause with little random or systematic uncertainties (Kursinski et al., 2002).

The sentence beginning on page 2, line 31 contained the word "precision." Eliminate the word precision by changing that sentence

Original Sentence: Kursinski et al. (2002) found that such a system could provide water vapor retrievals with a precision of $1 - 3\%$ from near the surface well into the mesosphere.

Revised Sentence: Kursinski et al. (2002) found that such a system could provide water vapor retrievals with a random uncertainty of $1 - 3\%$ from near the surface well into the mesosphere.

**9. p.4, l.1: "precise". See comment 5 above.**

We believe the word "precise" is appropriate as it is used on page 4, line 1. However, we do believe a wording change is warranted for "precise" on page 5, line 3. This instance was not specifically mentioned by the reviewer.

Original sentence: This ratio of ratios approach enables precise measurement of water vapor in the presence of clouds and rain as we demonstrate below.

Revised sentence: This ratio of ratios approach enables measurement of water vapor in the presence of clouds and rain with very small random and systematic uncertainty as we demonstrate below.

**10. p.4, l.3-4: ". . . attenuation . . . distributed along the path . . . In contrast, . . .". Although this problem is more acute for passive radiometry, it is also present for RO – the attenuation is also distributed along the path and it is not possible to say where exactly along the path it takes place, even in the full RO retrieval context. So "In contrast" is too strong.**

In order to address this concern, several wording changes are proposed within the paragraph that begins on page 4, line 1.

Original: ATOMMS functions as a precise, active spectrometer over the propagation path between the transmitter and receiver. Retrievals of water vapor from radiance measurements are inherently ambiguous because both the unknown signal source emission and attenuation, which are distributed along the path, must be solved for, creating an ill-posed problem (e.g., Rodgers, 2000). In contrast, the ATOMMS signal strength is known and the observed quantity is simply the attenuation along the path, which makes the retrievals much more direct and unambiguous. The active approach also enables precise and accurate retrievals under conditions of large path optical depths, which is not possible for passive retrievals.

Revised: ATOMMS functions as a precise, active spectrometer over the propagation path between the transmitter and receiver.  Retrievals of water vapor from radiance measurements are inherently ambiguous because both the signal source emission and attenuation along the path are unknown and must be solved for, creating an ill-posed problem (e.g., Rodgers, 2000).  In comparison to radiance retrievals, ATOMMS has the advantage that the transmitted signal strength is well known and the observed quantity is simply the attenuation along the path, which makes the retrievals much more direct and less ambiguous.  The active approach also enables retrievals with small random and systematic uncertainty under conditions of large path optical depths, which is not possible for passive retrievals.

**11. p.4, l.8: "in terms of amplitude rather than intensity". Does this explain the factor of 1 2 in eq.(1)? It may be worth pointing this out. Otherwise it looks like a rather unconventional definition of optical depth.**

Add the following sentence after the sentence ending on line 11 of page 4. The factor of ½ multiplying the optical depth comes about because intensity is proportional to amplitude squared.

**12. p.6, l.12-26. It would be helpful to say something here about another geometric difference: in this experiment, all the attenuation takes places at ~2600 m in height whereas, even in a single LEO-LEO measurement, it takes place over a range of altitudes characteristic of limb-viewing geometry.**

Four wording changes have been made to address this comment.

(a) Add the following sentence after the sentence ending on page 5, line 27.

The attenuation contributed at higher altitudes along the ray path is comparatively much smaller than the contribution near the ray path tangent altitude due to both the limb sounding geometry and the exponential decay in water vapor concentrations with altitude.

(b) Change the sentence that begins on page 5, line 27.

Original Sentence: We note that the Abel transform isolates the contributions of these layers.

New Sentence: We note that the Abel transform isolates the contribution from the lowest altitude portion of the signal path.

(c) Change the sentence that begins on page 5, line 28.

Original Sentence: For a vertical resolution of 100 m, the horizontal length of the lowest layer is approximately 70 km (Eq. 13, Kursinski et al., 2002).

New Sentence: For a vertical resolution of 100 m, the horizontal length of the path through the lowest layer is approximately 70 km (Eq. 13, Kursinski et al., 2002).

(d) Change the wording of the sentence that begins on page 6, line 12.

Original Sentence: "In this mountaintop demonstration, the atmospheric path from transmitter to receiver was only 5.4 km, such that the water vapor attenuation due to absorption by the weak 22 GHz line was too small to measure accurately."

New Wording (2 sentences): In this mountaintop demonstration, the atmospheric path from transmitter to receiver took place over a narrow altitude range from 2752 m to 2515 m above sea level and was only 5.4 km in length. Over this short path the water vapor attenuation due to absorption by the weak 22 GHz line was too small to measure accurately.

**13. p.7, l.2-3: "as indicated by the annotations in Fig.2". The intended annotations in Fig.2 appear to be missing. See comment 3 above. Similarly, "the First Reference period" (p.2, l.8) is not clear and presumably is intended to be an annotation on Fig.2.**

Replacing figure 2, based on general comment #3, eliminates this issue.

**14. p.7, l.21 and l.23: 4:30 –> 16:30?**

Our mistake. The time should be specified as 16:30 in both instances. Changes will be made.

**15. p.7, l.29: "as the calibration tone". Do you mean fCAL, which you call the "calibration signal" on p.4, l.20? It would be helpful to standardise references to fCAL, calibration signal, calibration frequency and calibration tone throughout.**

To avoid confusion and for consistency, we have decided to use the term "calibration signal" throughout. The calibration signal amplitude at a particular selected frequency (typically in the wing of the absorption line) is used to remove or reduce unwanted common mode amplitude variations before using the on line signal frequencies to estimate atmospheric absorption.

Here are all the instances of $f_{CAL}$, calibration frequency, and calibration tone that have been changed. The changes are shown below. If just a single word is changed, the change is indicated in blue.

[revised manuscript text omitted]

**16. p.8, l.1-5. The description of the processing is very compressed and it is not possible for the reader, from this alone, to understand how the processing is done. Can you give a reference to a more complete description?**

We propose to significantly change the wording under the heading "Signal Tuning and Detection," which begins on page 7, line 25. Major changes are shown in blue.

Original text.

The High Band portion of the ATOMMS ground-based prototype instrument simultaneously transmits and receives two continuous wave signals that are tunable from 181 to 206 GHz. For this experiment, one transmitter swept through the tunable frequency range generating a tuned tone that was received by a receiver sweeping through the same tuning sequence. The other tone was fixed at 198.5 GHz in order to function as the calibration tone. There were 122 tuning frequencies in the sweep, separated by 0.25 GHz, except for a gap between 191.5 and 193.5 GHz. This gap is due to the limited receiver response for Intermediate Frequencies (IF) less than one GHz and the first stage local oscillator (LO) being set to 192.5 GHz. This is likely the finest spectral resolution sampling of the 183 GHz line ever achieved in the field.

The dwell time for each frequency of the tuned transmitted tone was 100 ms. The timing of the transmitter-receiver tuning was synchronized using GPS receivers. Each received ATOMMS signal was filtered, down converted in frequency, digitized and recorded. The signal frequency in the final receiver stage ranged from 8 to 35 kHz for each of the 122 tuned frequencies. The frequency and power of the down converted signals were detected using a Fast Fourier Transform (FFT) and the signal amplitude was determined by taking the square root of the integrated signal power from each FFT. The integration time was 50 ms, which is half of the dwell time to allow time for each synthesizer tune to settle. Calibration tone amplitudes were computed using the same method.

One sweep of the frequencies took 12.2 seconds. The instrument cycled through the four combinations of the two transmitters and two receivers before repeating the tuning cycle to help isolate any transmitter or receiver issues. Thus, a full tuning cycle was completed every 48.8 s. The observations from the four different transmit-receive pairs were averaged together to yield new estimates for the ATOMMS signal ratio every 48.8 seconds (Eq. (2)). The resulting integration time for each particular tuned frequency was four times 50 ms or 200 ms."

Revised text.

The High Band portion of the ATOMMS ground-based prototype instrument simultaneously transmits and receives two continuous wave signals that are tunable from 181 to 206 GHz. For this mountaintop experiment, the frequency of the signal generated by one transmitter was swept through a tuning sequence that spanned the instrument's tunable frequency range. This signal was received by a narrowband heterodyne receiver whose second local oscillator was simultaneously swept through its matching tuning sequence. The frequency of the other signal was fixed at 198.5 GHz in order to function as the amplitude calibration signal for measuring differential absorption. There were 122 frequencies in the tuning sequence, separated by 0.25 GHz, except for a gap between 191.5 and 193.5 GHz due to the receiver's limited response for Intermediate Frequencies (IF) less than one GHz and the first stage local oscillator

(LO) being set to 192.5 GHz.

When executing the tuning sequence, the tuned transmitter tone dwelled at a particular frequency in the tuning sequence for 100 ms before moving to the next frequency in the sequence. The timing of the transmitter and receiver tuning sequences were synchronized using GPS receivers. At the receiver, each of the two received ATOMMS signals was filtered, down-converted in frequency, digitized and recorded. The frequency and power of the down-converted signals were determined using a Fast Fourier Transform (FFT), calculated over a 50 ms integration time. The reason that only half of the 100 ms tuning dwell time was used was to allow time for each synthesizer tune to settle. Each FFT-derived signal power estimate was then converted to an amplitude by taking the square root. The calibration signal amplitudes were computed using the same method.

One sweep through the frequency tuning sequence took 12.2 seconds. The instrument cycled through the four combinations of the two transmitters and two receivers before repeating the tuning cycle in order to help isolate any transmitter or receiver issues. Thus, a full tuning cycle was completed every 48.8 s. The observations from the four combinations of transmitter-receiver pairs were then averaged together such that new estimates for the ATOMMS signal amplitude ratios at all of the 122 tuning frequencies were generated every 48.8 seconds (Eq. (2)). As a result, the integration time used to estimate the signal amplitude and frequency for each of the 122 frequencies in the tuning sequence was four times 50 ms or 200 ms.

**17. p.8, l.15: "are identified in Fig.2". See comment 13 above**

Taken care of with the correctly annotated Figure 2.

**18. p.8, l.22 and elsewhere: "the am Atmospheric Model". This is not clear; is the name of this forward model the "Atmospheric Model", abbreviated as "am". Later it is referred to as "AM" sometimes with a version number. Please make all these references consistent.**

Several changes were made for clarification and consistency.

Page 8, line 22, original text. "…we used the *am* Atmospheric Model, version 7.2 [*Paine*, 2011] which was shown to fit the ATOMMS measurements to the 0.3% level in previous work with the ground-based ATOMMS prototype system [*Kursinski et al.*, 2012]."

Revised text. "… we used an atmospheric propagation tool known as the Atmospheric Model (am), version 7.2 (Paine, 2011), which we will refer to as *am7.2*. This model was shown to fit the ATOMMS measurements to the 0.3% level in previous work with the ground-based ATOMMS prototype system (Kursinski et al., 2012)."

Page 12, line 6, original text. "For the conditions of this particular experiment, based on the AM7.2 model, the sensitivity of the change in derived water vapor due to a temperature change relative to the reference period temperature was approximately -0.17 hPa/°C."

Revised text. "For the conditions of this particular experiment, based on forward calculations made with *am7.2*, the sensitivity of the change in derived water vapor due to a temperature change relative to the reference period temperature was approximately -0.17 hPa/°C."

Page 12, line 29, original text. "This half range represents a conservative estimate of the random uncertainty of the retrieved vapor pressure changes that includes both measurement and am model errors."

Revised text. "This half range represents a conservative estimate of the random uncertainty of the retrieved vapor pressure changes that includes both measurement and *am7.2 modeling errors*."

Caption of figure 5, current text. "forward calculated ATOMMS ratio using the am Model, version 7.2."

Revised text. "forward calculated ATOMMS ratio using *am7.2.*"

**19. p.9, l.10-11: "12 different solutions". At this point the 12 solutions have not been mentioned or explained. Please refer forwards to section 5 for this description.**

We have referred to Section 5 already. The current text is "The retrieved path-averaged vapor pressure between the instruments is shown in Fig. 3A. The figure shows 12 different solutions that were used to estimate the random uncertainty in the retrieval of vapor pressure as described in Section 5."

The following change has been made for clarity: "The figure shows 12 different solutions that were used to estimate the random uncertainty in the retrieval of vapor pressure. The methodology used to compute the 12 solutions is described in Section 5."

**19. p.10, l.4-20. This is a very ingenious solution to the problem – nice piece of work! The only question remaining at this point in the reader's mind concerns the inherent uncertainties in this method. You discuss this later in section 5, and so I suggest here you refer forward to this discussion.**

This is already done on page 10, line 10, where it is stated "The uncertainty associated with this temperature estimation is discussed in Section 5."

To be clearer that sentence will be moved to the end of the last paragraph of the section "Determining Temperature," which is page 10, line 20 in the original document.

**20. p.12, l.6 and l.26: "AM7.2" and "am". See comment 18 above.**

Changes specified in response to comment #18 above.

**21. p.18, l.7-13. In addition to these points, it is worth mentioning that GPS RO loses sensitivity to humidity, typically from the mid-troposphere upwards (at a height dependent on absolute humidity and hence temperature) because of the relatively low absorption coefficient of water vapour at GPS frequencies.**

We respectfully disagree with this comment. GPS has very little sensitivity to water vapor via absorption because its frequencies are so low in comparison to the 22 GHz water absorption line. The sensitivity that GPS does have to water vapor is via the propagation delay due to the refractivity of water vapor which is significant basically at temperatures above 240 K (because there is enough water vapor present to see this effect in individual profiles). However, one cannot isolate that water vapor signature directly from the GPS observations alone. One must add additional constraints on the dry part of the refractivity to isolate the wet part of the refractivity buried in the GPS refractivity estimates.

**22. p.18, l.1 and l.23: "achieves" –> "would achieve"? "offers" –>"would offer"?**

We like use of the word "achieve" on page 18, line 2. No change here.

On page 18, line 23, though, we like the reviewer's suggestion to change "offers" to "would achieve."

Revision to sentence shown: Given this present situation, ATOMMS' precise, all-weather retrieval capability, as demonstrated here,  would achieve a major advance in remote sensing of the atmosphere.

**23. p.19, l.4-6. It would be more conventional to write this sentence as a simple statement with reference "(Holger Vömel, personal communication)".**

Thank you.

Original wording. "However, when we discussed validating ATOMMS instruments to 1% with Holger Vömel, a chilled mirror hygrometer expert at NCAR, he indicated that no in-situ measurements can reliably achieve 1% accuracy out in the field."

Revised wording. "However, when we discussed validating ATOMMS instruments to 1% with a chilled mirror hygrometer expert at NCAR, we were told that no in-situ measurements can reliably achieve 1% accuracy out in the field (Holger Vömel, personal communication)."

**24. p.19-20. In addition to the arguments presented in this Appendix, I think there is an additional one. If the typical scales that need to be measured are ~13m, then any in situ observing system would**

**need to be so close to the axis of the ground-based remote sensing system that it would interfere with the measurement. Otherwise it is not measuring the same path.**

Thank you. This is a point worth mentioning. The following sentence has been added after the second sentence in the last paragraph that begins of page 19. "It would be difficult, if not impossible, to locate these sensors close enough to the signal propagation path without interfering with signal itself."

**25. Fig.2. See comments 3 and 13 above**

Figure 2 will be replaced with the correctly annotated figure.

**Editorial Points:**

**26. p.1, l.2. Title: "mm- wavelength" –> "mm-wavelength"?**

We agree. There should probably not be a space between mm- and wavelength. We will have to check with the editorial staff. Change initially will be made to original document.

**27. p.3, l.16. "Uncertainty" –> lower case?**

The word "Uncertainty" will be changed to lower case "uncertainty."

**28. p.11, l.20. "," after "seconds"?**

Comma will be added after the word "seconds" on page 11, line 20.

**29. p.16, l.30: "sought after" –> "sought-after"?**

Thanks. A hyphen should be used. "sought after" will be replaced by "sought-after"

---

## Author Comment (AC4) · 31 Jul 2018

**COMMENTS FROM DAVE ADAMS (REVIEWER 3).**

We wish to thank Dave Adams for his helpful review. We first copy the comments of the reviewer **(in bold)** with our response below. In cases where the text is modified, we show both the original text and the revised text, with the significant changes shown in blue.

**Minor Comments.**

**Line 26 You can probably be a bit more emphatic here. Water is the most important greenhouse gas, critical in the energy balance, responsible for storms etc... Line 29 Should be Sherwood et al., (2010) And you should probably include a few more "big picture" type references related to water vapor in the climate system.**

To address this comment, we will make substantial changes to the first paragraph in section 1.

Original text:

Water vapor is an important constituent in Earth's atmosphere and its distribution in space and time must be known to understand and predict weather and climate. Despite its importance, we do not have precise observations of its distribution in the atmosphere, its trend with time, or a good understanding of the factors controlling these (Sherwood et al., 2013). Water vapor is challenging to measure because of the wide range of concentrations and scales across which it varies. Water vapor observations must be unbiased and capture the full range of variability in clear and cloudy conditions across the globe in order to improve the understanding and analysis of water vapor, which is used to initialize weather prediction systems, to monitor trends and variations and to improve weather and climate models through constraints on and refinement of processes affecting and affected by water vapor (e.g., Bony et al., 2015).

Revised text:

Water vapor is an important constituent in Earth's atmosphere and its distribution in space and time must be known to understand and predict weather and climate. Water vapor is fundamental to the radiative balance of the Earth, both as the most important greenhouse gas and indirectly through clouds. Through its latent heat, water vapor is crucial to formation and evolution of severe weather, transport of energy both upward and poleward in the troposphere and transfer of energy between the surface and atmosphere. Furthermore, water vapor dominates tropospheric radiative cooling which drives convection (Sherwood et al., 2010). Uncertainty in modeled cloud feedback results in the factor of 3 spread in predictions in the surface temperature response to a doubling of atmospheric $CO_2$ concentrations and the cloud feedback depends critically on the strength of the water vapor feedback [Held and Soden, 2000]. Predicted amplification of extreme precipitation with warmer temperatures is tied directly to predicted increases in extreme water vapor concentrations and it may be underestimated (e.g., Allan and Soden, 2008).

The date on this reference item was incorrectly listed as 2013 and will be changed to 2010.

Sherwood, S. C., R. Roca, T. M. Weckwerth, and N. G. Andronova (2010), Tropospheric water vapor, convection, and climate, Rev. Geophys., 48, RG2001, doi:10.1029/2009RG000301

The following new references will be added to the reference list:

Held, Isaac M. and Soden, Brian J. (2000), Water vapor feedback and global warming, Annu Rev Energy Environ., 25, 441 – 475, doi: 10.1146/annurev.energy.25.1.441.

Allan, R.P., and B.J. Soden (2008), Atmospheric warming and the amplification of precipitation extremes. Science, 321, 1481-1484, doi: 10.1126/science.1160787.

**Line 30 Water vapor observations must be unbiased and capture the full range of variability in clear and cloudy conditions across the globe... This sentence is a bit awkward, it could be written in a more concise manner or turn into two sentences.**

This sentence has been removed based on the proposed change to the comment immediately above.

**Page 2.**

**Satellite systems typically do not have sufficient temporal or spatial resolution to capture many of the important processes related to the distribution of water vapor (such as deep convection in the Tropics). And if the satellite systems do have this appropriate temporal and spatial resolution (e.g. GOES water vapor channels), they only provide column water vapor and not its vertical structure. You should include a bit more detail in this paragraph to give greate force to your proposed system.**

To fully address this comment, we propose to make substantial changes and additions to Section 1 of the paper, Introduction/Motivation. We show the original text of the three paragraphs that begin on line 3 of page 2. After that, we show the proposed new text that will replace those paragraphs.

Original Text (starting on line 3 of page 2):

[revised manuscript text omitted]

The last two paragraphs of section 1 remain unchanged.

These changes require the following new items in our reference list.

Chen, J., A. D. del Genio, B. E. Carlson and M. G. Bosilovich, (2008), The Spatiotemporal Structure of Twentieth-Century Climate Variations in Observations and Reanalyses. Part I: Long-Term Trend. *J. Climate*, *21(11)*, 2611-2633.

Durran, D. R., and J. A. Weyn (2016), Thunderstorms Do Not Get Butterflies, *Bull. Amer. Met. Soc.,* February 2016, P. 237-244, doi.org/10.1175/BAMS-D-15-00070.1.

Esau, I. and S. Sorokina (2010), Climatology of the Arctic Planetary Boundary Layer,  in Atmospheric Turbulence, Meteorological Modeling and Aerodynamics, Editors: P. R. Lang and F. S. Lombargo, 2009 Nova Science Publishers, Inc., ISBN 978-1-60741-091-1.

Guan, B., and D. E. Waliser (2015), Detection of atmospheric rivers: Evaluation and application of an algorithm for global studies. *J. Geophys. Res. Atmos.*, **120**, 12 514–12 535, doi:10.1002/2015JD024257.

Hardy, K. R., Hajj, G. A. and Kursinski, E. R. (1994), Accuracies of atmospheric profiles obtained from GPS occultations. Int. J. Satell. Commun., 12: 463-473. doi:10.1002/sat.4600120508

Itterly, K. F., P. C. Taylor, J. B. Dodson, and A. B. Tawfik (2016), On the sensitivity of the diurnal cycle in the Amazon to convective intensity, J. Geophys. Res. Atmos., 121, 8186–8208, doi:10.1002/2016JD025039.

Kay, J. E., and A. Gettelman (2009), Cloud influence on and response to seasonal Arctic sea ice loss, *J. Geophys. Res*., **114**, D18204, doi:10.1029/2009JD011773.

Klingebiel, M., A. de Lozar, S. Molleker, R. Weigel, A. Roth, L. Schmidt, J. Meyer, A. Ehrlich, R. Neuber, M. Wendisch, and S. Borrmann (2015), Arctic low-level boundary layer clouds: in situ measurements and simulations of mono- and bimodal supercooled droplet size distributions at the top layer of liquid phase clouds, *Atmos. Chem. Phys*., **15**, 617–631, doi:10.5194/acp-15-617-2015.

Kuo, Y.-H., et al. (1998), A GPS/MET Sounding through an Intense Upper-Level Front, *Bull. Amer. Meteor. Soc.,* **79**, 617-626.

Kursinski, E. R., D. Ward, A. C. Otarola, A. L. Kursinski and C. McCormick (2016a), Reducing Climate and Weather Prediction Uncertainty via cm- and mm-Wavelength Satellite to Satellite Occultations, *White Paper Submitted to 2017 ESAS Decadal Survey In Applications\ of\ ATOMMS\ 072118.docx response to RFI#1, January 2016,* [http://surveygizmoresponseuploads.s3.amazonaws.com/fileuploads/15647/2289356/183-ea6d9954df8cfcbdb60a500c254348c4_KursinskiEmilR.pdf](http://surveygizmoresponseuploads.s3.amazonaws.com/fileuploads/15647/2289356/183-ea6d9954df8cfcbdb60a500c254348c4_KursinskiEmilR.pdf) .

Uttal, T., et al. (2002), Surface heat budget of the Arctic Ocean. *Bull. Amer. Meteor. Soc.,* **83,** 255–275.

Wang, J., A. Dai and C. Mears (2016), Global Water Vapor Trend from 1988 to 2011 and Its Diurnal Asymmetry Based on GPS, Radiosonde, and Microwave Satellite Measurements, *J. Clim.*, **29**, p. 5205-5222, DOI: 10.1175/JCLI-D-15-0485.1.

**Line 19 Can you be specific as to what you mean here by insensitivity "(2) its insensitivity to water vapor in the colder regions of the troposphere and above."**

We will add the following sentence to the end of the sentence in question. The new sentence begins on line 20 of page 2.

New Text: The insensitivity occurs when there is so little water vapor that the majority of the refractivity is dominated by the dry air component (e.g., Kursinski and Gebhardt, 2014).

**Line 28 I think you should write "It can also profile ozone ..."**

The statement about profiling ozone is provided in the very next sentence. No change here.

**Page 3**

**Line 2 Probably not necessary to include this "...we developed with funding from NSF,..."**

That part of the sentence will be removed. Instead, this information will be included in an acknowledgement section placed immediately before the references section.

Change to original sentence:

Using ground-based ATOMMS prototype instrumentation , we demonstrate the ability of ATOMMS to retrieve changes in the path-averaged water vapor between the instruments operating between two mountaintops in Southern Arizona to within 1%, during weather conditions that ranged from clear to cloudy to thunderstorms with heavy rain.

And add the following acknowledgement section:

**Acknowledgments**

We want to thank Jeff Kingsley for his support in making critical resources available at the University of Arizona's Steward Observatory needed to complete the ATOMMS instrumentation, and Chris Walker for sharing the Steward Observatory Radio Astronomy

Laboratory (SORAL) facilities with us during development of the prototype ATOMMS instrument.  We also want to thank Jim Grantham for providing access and modifications to the Mt Bigelow and Mt Lemmon facilities to support these observations. We thank David Adams and two anonymous reviewers whose constructive criticism improved the presentation of this paper considerably.  This work was supported by the National Science Foundation Major Research Instrumentation (MRI) Program grant 0723239 and the National Science Foundation, Division of Atmospheric and Geospace Sciences (GEO/AGS) grants 0946411 and 1313563.  In particular, we want to thank Jay Fein, program scientist and manager at NSF, who passed away in 2016.  Without Jay's insight and relentless effort and support, this research would never have been funded and taken place.

**Line 6 Clarify what you are referring to here "..and the forward modeled water vapor spectra,.."**

This line is in the introductory section. We do not believe that a detailed explanation belongs here. We will add a note that this is described in section 4.

Original text: The smaller than 1% discrepancies between the measured ATOMMS spectra and the forward modeled water vapor spectra …

New text: The smaller than 1% discrepancies between the measured ATOMMS spectra and the forward modeled water vapor spectra (described in Section 4) …

**Line 10 Don' t use contractions in formal writing. "...simply do not work."**

We will change don't on line 10 to do not.

**Line 16 Write "Sources of uncertainty ..."**

This typo (capitalized letter U) will be changed

Original text: Sources of Uncertainty

Revised text: Sources of uncertainty

**Line 30 Write Refractivity and "the" extinction coefficient (or write coefficients)**

The word "the" will be added in front of extinction coefficient in the sentence.

Original sentence: From these, occultation profiles of bending angle and absorption are derived and then used to derive radial profiles of refractivity and extinction coefficient using Abel Transforms [*Kursinski et al.,* 2002].

Revised sentence: From these, occultation profiles of bending angle and absorption are derived and then used to derive radial profiles of refractivity and the extinction coefficient using Abel Transforms (Kursinski et al., 2002).

**The hydrostatic assumption would be very dubious during deep convective activity**

As the manuscript notes, based on ATOMMS 100 to 200 m vertical resolution and equation 13 of Kursinski et al. (1997), the horizontal resolution of an orbiting ATOMMS system is approximately 100 km.  At the 100 km horizontal scale, hydrostatic equilibrium should be a good approximation.

We also note that NWP systems using non-hydrostatic models can assimilate the ATOMMS LEO observations of bending angle and path-integrated absorption, prior to the Abel transform and hydrostatic equilibrium steps in the retrieval process, to avoid the assumption of hydrostatic equilibrium. Non-hydrostatic NWP systems already do this with GPS RO bending angle profiles.

**Page 4**

**Line 11. This is a bit unclear. "The gas phase optical depth is due to water vapor and dry air absorption, which introduces temperature and pressure dependence, and any attenuation due to hydrometers." You are saying the gas phase optical depth is also dependent up the presence of non-gas constituents like hydrometeors?**

Yes, this does seem confusing. The sentence in question will be changed.

Original: The gas phase optical depth is due to water vapor and dry air absorption, which introduces temperature and pressure dependence, and any attenuation due to hydrometers.

Revised: The total optical depth is due to the gas phase optical depth plus the attenuation due to hydrometeors.  The gas phase optical depth includes water vapor and dry air absorption, which depend on temperature and pressure.  The hydrometeor attenuation also depends on temperature (Kursinski et al., 2009).

**Page 5**

**Again, not sure if this is necessary to state. "With funding from NSF,..."**

That part of the sentence will be removed. Instead, this information will be included in an acknowledgement section placed immediately before the references section. The new acknowledgement is shown above in response to a similar comment.

Change to original sentence:

We designed and built a ground-based, prototype ATOMMS instrument and then used it to demonstrate some key aspects of ATOMMS capabilities and

performance in several fixed geometries in southern Arizona with path lengths ranging from 800 m to 84 km.

**Page 6.**

**Line 14 "ATOMMS High Band signals" should be**

The original paper is inconsistent with references to the ATOMMS High Band signals.

We will search for all occurrences and change them to ATOMMS High-Band signals for consistency.

**Page 7.**

**Line 13. Which radar data are you referring to? You need to clarify this point.**

This is similar to a comment from reviewer 2 above. However, this reference to radar data precedes the one pointed out by reviewer 2. Thus, we will specify the radar here and include the reference in both places.

Original Sentence: The RADAR data and field observations indicated that rain was still falling over portions of the path between the two instruments.

Revised sentence: Radar data from the Tucson WSR-88D radar (Crum and Alberty, 1993) and field observations indicated that rain was still falling over portions of the path between the two instruments.

As mentioned in our reply to reviewer #2, the following reference will be added.

Crum, T.D. and R.L. Alberty, 1993: The WSR-88D and the WSR-88D Operational Support Facility. *Bull. Amer. Meteor. Soc.,* **74**, 1669–1688, https://doi.org/10.1175/1520-0477(1993)074<1669:TWATWO>2.0.CO;2

**Line 13 Write "By 16:30, the rain was considerably lighter"**

We will change the sentence in question by adding the comma after 16:30.

Original: By 16:30 the rain was considerably lighter.

Revised: By 16:30, the rain was considerably lighter.

**Line 33. Can you back this statement up with any citations or some references. "This is likely the finest spectral resolution sampling of the 183 GHz line ever achieved in the field."**

Revised text:

**Page 8.**

**Line 22 Should this be capitalized AM**

References to the am7.2 microwave propagation model have been standardized based on comment #18 from reviewer 1 as *"am7.2"*

Below is the revised text on page 8, line 22, where we first mention the model and define our terminology. All subsequent references to the model will use *"am7.2"* for consistency.

"… we used an atmospheric propagation tool known as the Atmospheric Model (am), version 7.2 (Paine, 2011), which we will refer to as *am7.2*.

**Page 9.**

**Line 30. What size of error should we expect given the use of local pressure measurements at each of the sites? That is, across the line of site, there should be some small variability of pressure given updrafts and downdrafts.**

**Background**: The figure below shows the in-situ pressure observations at both instrument locations and the one hour running mean that we used to determine the air temperature using a hydrostatic approximation. This figure is also included above in our response to reviewer #2 concerning **p. 10, line 19.** The figure shows that the unsmoothed pressure variations at each observation site were up to 2 hPa during the time of the convection as shown in the figure below. When we initially thought to use the hydrostatic pressure scale height to infer the average temperature over the path, these short term pressure variations mapped into unphysical temperature variations. As a result, we recognized immediately that these pressure variations must be non-hydrostatic. These 2 hPa variations are consistent with non-hydrostatic pressure variations of up to 2 hPa expected during convection.

[Figure]

That figure also shows that there were slower varying, highly correlated pressure variation of about 2 hPa over the duration of the experiment. As a result of these considerations, we ended up using a 60 minute running mean of pressure to estimate the average air temperature along the signal path.

**Response**: We interpret this question as asking, what is the impact of pressure variations along the path relative to the retrieval of water vapor. While variations in air pressure do impact the water vapor line shape, the resulting changes are quite small. The figure below, which we do not plan to include in the paper, shows the variations in the ATOMMS amplitude ratio over the 187.5 to 191.5 GHz range of signal frequencies that were used in our retrievals. The changes in the ATOMMS amplitude ratio that result from $\pm2$ hPa changes in pressure are comparable to changes due to $\pm 0.2\%$ changes in vapor pressure. The calculations for the figure assumed reference period one conditions, namely vapor pressure = 15 hPa, air temperature = 20° C, and air pressure = 743 hPa. Thus, the non-hydrostatic pressure variations during the convective period are insignificant relative to 1% variations in water vapor and therefore do not impact our conclusion that ATOMMS observations enabled water vapor retrievals to within 1%.

[Figure]

**Manuscript changes:** In response to this comment from Dave Adams about the impact of pressure variations on the retrievals, we intend to add a line in Figure 6 that represents the change in amplitude ratio that results from a change in pressure of +10 hPa relative to the reference conditions. The point is to show that even for pressure variations much larger than those that were observed, the impact on the water vapor retrieval is insignificant.

Below is the revised Fig. 6 and caption. This is followed by corresponding changes to the text on page 12 beginning with the sentence that starts on line 8, which is where we describe Fig. 6. Those changes are shown below the revised figure and caption.

[revised manuscript text omitted]

**Page 15**

**Line 17. Such behavior where moisture at the surface varies little while air aloft becomes significantly drier following summertime thunderstorms is common in this region (e.g., Fig. 4 in Kursinski et al. [2008]). You can probably find a few more references that describe theromdynamic conditions after T-storms during the NAM**

We will add the following sentence after the sentence that ends on line 17. The sentence ending on line 17 is repeated first, followed by the new sentence, which contains a few new citations.

Revised wording. Such behavior where moisture at the surface varies little while air aloft becomes significantly drier following summertime thunderstorms is common in this region (e.g., Fig. 4 in Kursinski et al., 2008). It is also common in the Amazon (e.g., Fig. 7 in Schiro et al., 2016) and may be associated with mid-level inflow of drier air into the precipitating region that results in evaporative cooling and descent of this air (e.g., Leary, 1980 and Houze, 2004).

This requires the following additions to the reference list

Schiro, K. A., J. D, Neelin, D. K. Adams and B. R. Lintner (2016), Deep Convection and Column Water Vapor over Tropical Land versus Tropical Ocean: A Comparison between the Amazon and the Tropical Western Pacific, *J. Atmos. Sci*., 73, p. 4043-4063, DOI: 10.1175/JAS-D-16-0119.1.

Leary, C. A. (1980), Temperature and humidity profiles in mesoscale unsaturated downdrafts, *J. Atmo. Sci*., 37, p. 1005-1012.

Houze, R. A., Jr. (2004), Mesoscale convective systems, *Rev. Geophys*., 42, RG4003, doi:10.1029/2004RG000150.

**Page 16**

**Line 21 "The nearby Tucson radiosonde indeed indicated that..." With all of the reference in the paper to this sounding, you should include it in the figures.**

Based on this recommendation, we performed a closer examination of the afternoon sonde profile of August 18, 2011, and we have found that we can infer a bit more than we had previously understood about what happened that afternoon.

Therefore the figure, below will be added to the paper. It shows specific humidity, $q$, and potential temperature, $\theta$, derived from the afternoon sonde. In the absence of sources and sinks, $q$ and $\theta$ are conserved variables that can provide additional insight into what happened that afternoon. The figure shows the 3,000 meters above the Tucson valley floor in order to see the boundary layer structures and the ATOMMS' height interval.

**Revisions:** The new figure, which will be Figure 9 in the paper, is shown immediately below with its proposed caption. We also propose to make changes to the text to describe the figure and its significance. The text changes are shown below the new figure.

**Figure 9.**

[Figure]

Figure 9: Vertical profiles of specific humidity and potential temperature minus 300 K calculated from the 00 UTC Tucson sonde. The local time of the sonde launch was approximately 16:30 on August 18. Theta label for the red line stands for potential temperature and PBL stands for planetary boundary layer.

To provide the reviewers with some context for the modified text, we first repeat the two paragraphs at the end of Section 6, starting on line 9 of page 15, with minor changes indicated with strikethrough marks for deletion and blue for text changes. Below that, the paragraph shown in entirely in blue is a new paragraph.

**Revised text:**

[revised manuscript text omitted]

**Page 18**

**Frequency of observations will always be an issue to some extent with the RO technique, particularly when the scales are of the time and space scales need for weather prediction.**

Below we respond to the reviewer's comment. We have proposed text changes to address these points in Section 1, which are in response to your second overall comment with respect to page 2 of the original document. We are not planning to make any additional changes to the text of the paper on page 18 as these have been addressed in Section 1.

The reviewer raises an important point that we have thought about since the conception of ATOMMS. ATOMMS measurements from low Earth orbit (LEO) will profile the atmosphere with very high, 100-200 m vertical resolution with a corresponding horizontal resolution of approximately 100 km as noted in the manuscript. As the results of this paper imply, ATOMMS profiling from LEO will work quite well in both clear and cloudy conditions which is critical for sampling convection. The vertical information that ATOMMS sensors in LEO will provide promises to provide information across the globe on atmospheric stability and particularly conditional instability that are critical for predicting the onset and evolution of atmospheric convection. We also note that ATOMMS' ~100 km horizontal resolution matches the 100 km scale that Durran and Weyn (2016) argue is the most important scale for forecasting thunderstorms.

Thus, ATOMMS LEO data promises to be very useful with regard to convection IF we can create sufficiently dense ATOMMS sampling densities.   This issue was discussed in Kursinski et al. (2016) who noted that a constellation of 60 very small satellites, carrying both ATOMMS and GNSS RO sensors, would produce approximately 25,000 ATOMMS occultations and 170,000 GNSS occultations each day, for a fraction of the cost of a NOAA polar orbiting weather satellite.  The orbits they noted would sample the entire globe every 6 hours to support the 6 hour update cycle of global weather prediction centers.  The average spacing between ATOMMS and GNSS occultations every 6 hours would be approximately 320 and 120 km respectively which is quite dense compared to present GNSS RO and radiosonde sampling.  To further put that into perspective from the standpoint of convection, such a system would provide approximately 300 ATOMMS profiles and 1,800 GNSS RO profiles respectively over the Amazon basin, each day.

Thus, an eventual large ATOMMS+GNSS RO constellation, which can be implemented relatively cost effectively, promises to be quite enlightening for improving our understanding and ability to predict atmospheric convection.

These points are made in the updated text in response to Dave Adams comment on page 2.

---

## Author Comment (AC5) · 1 Aug 2018

**Responses to Comments from Anonymous Referee #2**

We wish to thank this reviewer for their helpful review. We first copy the comments of the reviewer **(in bold)** with our response below. In cases where the text is modified, we show both the original text and the revised text, with the significant changes shown in blue.

**General Comments:**
**The paper is a very interesting and indeed necessary contribution to investigate the atmosphere based on occultation measurement technique by exploiting the microwave capacity. It describes in detail the ground based experiment by detecting water vapour with microwave signals under clear too very turbulent conditions as pre-study to a microwave occultation satellite mission. Besides a detailed results description an error and validation discussion was done. The discussion in the appendix about the difficulty or even impossability to in-situ validate such an experiment rounds up the paper. I recommend the paper for publication with minor revisions.**

**In this paper the low band signal is only used to determine whether there are small water particles present or not but it apparently is not used for the retrieval of water vapor, which is done only for the high band signal (Fig. 2). It is though claimed that to derive the water vapor content under stormy conditions when the high band becomes opaque, the low band is needed but it is not done for this experiment.**
**Did you get results on the 8 fixed frequencies at the 22 GHz region to investigate the heavy rain situation at 15:30, since for such heavy weather situation sufficiant attenuation occured?**

Unfortunately, we determined that the measurements taken at the 8 fixed frequencies near 22 GHz on this day could not be used to make accurate retrievals for water vapor. The main reason is that the path length (of only 5.4 km) is too short for accurate determination of water vapor using the weak 22 GHz line as the absorption is so small. While the liquid absorption was strong during the heavy rain period, the additional absorption due to water vapor is quite small across the 22 GHz band.

**Specific Comments:**
**p. 2, line 16—20: Please give a reference to the RO technique including the specifications/limits of this remote technique. Please give references for other studies on microwave occultation technique pre-studies too.**

To address this comment, we propose making some wording changes and adding additional references to the two consecutive paragraphs that begin on page 2, line 15.

Original text:

[revised manuscript text omitted]

**p. 5, line 1—4: Did you do investigations in terms of distance of frequency to the calibration frequency and the remaining error if unwanted sources (scintillation, ..) of errors show a frequency dependency?**

No formal (published) investigations have been done, though we have done internal simulations. The reference to Kursinski et al. (2016) that we suggest to add based on the next comment addresses this question for our ground-based retrievals.

**p. 5, line 7: If the demonstration of these key aspects are published please give a reference.**

In order to address this comment, we propose to add text and references to published work for our ground-based retrievals. The new text is rather long, but there is a lot to summarize. These references to past publications concerning ground-based retrievals probably should be contained in this paper.

We suggest to break the first paragraph in Section 3 after the second sentence and include a summary of our published findings from previous experiments using the ground-based ATOMMS system. The remaining sentences currently in the first paragraph will be moved to a new paragraph beginning after the suggested insertion given below.

Here is the revised beginning of Section 3. Again, only the first sentence is carried over.

New text to begin section 3:

With funding from NSF, wWe designed and built a ground-based, prototype ATOMMS instrument and then used it to demonstrate some key aspects of ATOMMS capabilities and performance in several fixed geometries in southern Arizona with path lengths ranging from 800 m to 84 km. The prototype ATOMMS High-Band system transmits and receives two simultaneous continuous wave (CW) signals tunable from 181 to 206 GHz. The prototype Low-Band system consists of eight CW transmitters and receivers at fixed frequencies from 18.5 to 25.5 GHz spaced approximately one GHz apart, 10 centered approximately on the 22 GHz water vapor absorption line. Below we summarize the content of previous published work based on field experiments with the ATOMMS ground-based prototype.

In terms of ATOMMS water vapor retrievals, Kursinski et al. (2012) demonstrated agreement at the 2% level between water vapor measurements derived along an 820 m path using the ATOMMS High-Band instrument and a nearby, capacitive-type hygrometer. High-Band mountaintop measurements yielded the first detection by ATOMMS of $H_2^{18}O$ via its 203 GHz absorption line (Kursinski et al., 2016). Such measurements in the upper troposphere will determine isotopic ratios to constrain the hydrological cycle (Kursinski et al., 2004).

In terms of spectroscopy, ATOMMS measured lineshape across the 4 GHz interval above the 183 GHz line center agreed with the HITRAN line shape with a standard deviation of 0.3% (Kursinski et al. 2012), some 8 times better than the previously best estimate of Payne et al. (2008). ATOMMS mountaintop measurements between 5 and 25 GHz above the line center revealed discrepancies with the HITRAN line shape (Kursinski et al., 2016) which may help explain inconsistencies in 183 GHz derived water vapor estimates (Brogniez et al., 2016) and may be associated with atmospheric turbulence (Calbet et al., 2018). The ATOMMS measurements also revealed the shape of the 183 GHz line as represented in the Liebe et al., (1993) model is incorrect (Kursinski et al., 2012). The Liebe model is popular, having been referenced more than 600 times in the literature, and is still being used.

Kursinski et al. (2012) combined ATOMMS High-Band measurements with precipitation radar measurements to derive cloud liquid water content (LWC) along the ATOMMS signal path.  Kursinski et al. (2016) demonstrated the ability to derive both cloud LWC and rainfall rates by combining the ATOMMS Low-Band and High-Band measurements.

Kursinski et al. (2016) derived the strength of atmospheric turbulence from scintillations of the ATOMMS signal amplitudes and demonstrated the ability to significantly reduce these turbulent amplitude variations via amplitude ratioing, in order to derive accurate water vapor estimates in turbulent conditions.

The last paragraph in the overview portion for Section 3 will be the end of the original paragraph:

On August 18, 2011, we collected approximately four hours of data with the instruments located on Mt. Lemmon Ridge (2752 m altitude) and Mt. Bigelow (2515 m altitude), separated by approximately 5.4 km. The observing geometry is shown in Fig. 1. The water vapor pressure derived from these ATOMMS measurements represents an average over the 5.4 km path which runs above a valley between the mountaintops on which the instruments sit.

This change also requires the following additional references to be added to the reference list:

Kursinski, E. R.,  D. Feng, D. Flittner, G. Hajj, B. Herman, F. Romberg, S. Syndergaard, D. Ward, and T. Yunck (2004), An Active Microwave Limb Sounder for Profiling Water Vapor, Ozone, Temperature, Geopotential, Clouds, Isotopes and Stratospheric Winds, in *Occultations for Probing Atmosphere and Climate* (OPAC-1), Springer-Verlag, Berlin, ISBN 978-3-540-22350-4, p. 173-188.

Calbet, X.,  N. Peinado-Galan, S. DeSouza-Machado, E.R. Kursinski, P. Oria, D. Ward, A. Otarola, P. Rípodas, and R Kivi (2018), Can turbulence within the field of view cause significant biases in radiative transfer modelling at the 183 GHz band? *Submitted to Atmos. Meas. Tech.*

Liebe, H.J., G.A. Hufford, and M.G. Cotton (1993), Propagation Modeling of Moist Air and Suspended Water/Ice Particles at Frequencies Below 1000 GHz. AGARD Conference Proc. 542, Atmospheric Propagation Effects through Natural and Man-Made Obscurants for Visible to MM-Wave Radiation, pp.3.1-3.10.

**p. 5, line12: Please determine here already on wich of these locations the transmitter and receivers are located.**

The following sentence will be added before the first sentence begins on page 5, line 13 of the originally submitted document. The paragraph that includes that sentence will be moved down in the introductory portion of section 3 as described in the previous response. It will begin after the sentence ending with "shown in Fig. 1.

The Mt. Lemmon instrument contained the 183 GHz transmitter and 22 GHz receiver and the Mt. Bigelow instrument contained the 22 GHz transmitter and 183 GHz receiver.

**p. 7, line 8: What is the reference period exactly. For a faster understanding it would helpt too if the reference period is marked in Fig. 2.**

See response to general comment #3 from anonymous reviewer #1. We originally submitted the wrong figure 2 without the annotations. Reference periods are marked with annotations in the correct figure.

**p. 8, line 15: Please mark the second reference period in the Fig.2 too.**

The second reference period is marked on the correct figure 2.

**p. 8, line 22: Please give a clear acronym for your Atmospheric Model, version 7.2 e.g. AM7.2 and make the notation consistent in the entire document.**

This was addressed in response to specific comment #18 from reviewer #1.

**p. 9, line 5: Please give the color code in the text too you are using in the figures for a faster understanding. It is not explained in the text what „raw" in the figure label means? Why is the blue line not always higher then the black line? To my understanding the blue lines contains all atmospheric effects and the black line only the liquid optical depth part.**

Figure 2 is introduced in the text beginning on page 6, line 32. We propose to make clarifying changes to the text at that point, rather than on page 9.

Original text: The observed variations in optical depth at 198.5 GHz and 24.4 GHz are shown in Fig. 2. 198.5 GHz was the frequency of the High Band calibration tone during this experiment. Also shown are the derived changes in liquid optical depth at 198.5 GHz, which was computed by subtracting the optical depth changes due to variations in the retrieved vapor 7 pressure and temperature from the total observed optical depth change.

Revised text: The measured changes in optical depth at 198.5 GHz (blue line, raw) and 24.4 GHz (red line, raw) are shown in Fig. 2. 198.5 GHz was the frequency of the High Band calibration tone during this experiment. Also shown are the derived changes in liquid optical depth at 198.5 GHz (black line) , which was computed by subtracting the optical depth changes due to variations in the retrieved vapor 7 pressure and temperature from the total observed optical depth change. The change in optical depth relative to reference period 1 will always be positive for liquid (rain and clouds), since there was no rain or clouds during the reference period. However, the change in optical depth due to changes in vapor pressure and temperature can be negative,

which means that the overall change in optical depth relative to the reference period can be less than the optical depth change due to liquid alone.

**p. 9, line 19: Please give a reference to this publications.**

Reference will be added. This paper is already in our reference list. We also noticed a reference to the wrong section at the beginning of the sentence, which will also be changed.

Original Sentence: In Section 5 we note that similar advection of dry air following summertime thunderstorms in this region have been observed in previously published work and show that our estimation of the minimum vapor pressure was consistent with the nearby radiosonde observations from Tucson.

New Sentence: In Section 6 we note that similar advection of dry air following summertime thunderstorms in this region have been observed in previously published work (Kursinski et al., 2008) and show that our estimation of the minimum vapor pressure was consistent with the nearby radiosonde observations from Tucson.

**p. 10, line 3: Why where the tempearature sensors so close to the ATOMMS instruments positioned? Why not in a seperate small tent, if protection due to heavy rainfall was needed, to avoide temperature bias due to lost heat by the ATOMMS instrument?**

In retrospect, this was a mistake on our part. If we were to repeat this experiment, we would definitely position in-situ sensors slightly away from the operating instruments. However, as you say, these instruments would still have to be protected from the heavy rain and winds in a tent and therefore, not as tightly tied to the environmental conditions as one would like.

**p. 10, line 19: Please use the color code of the graph when explaining the figures to gurantee no confusion with the graphs.**

The following change will be made to the text.

Original Sentence: Figure 4 shows the derived air temperature between the instruments that was used in the retrievals, as well as the nearby, in-situ thermometer observations.

New Sentence: Figure 4 shows the derived air temperature between the instruments that was used in the retrievals in black, as well as the nearby, in-situ thermometer observations, which are shown in red, green, and blue.

**p. 10, line 19: How do the aire pressure graphs look like including the one hour running mean of the air pressure?**

The differences between the one hour running mean of air pressure and the higher time resolution observations of pressure are almost imperceptible when plotted on the same graph. Yet, these small variations over short time scales (relative to each other) do create noise in the hydrostatic temperature estimates when not using the time smoothed pressure. We have not made any changes to the text based on this comment. Below is a figure that shows both the high time resolution measurements of air pressure and the one hour running mean. We do not plan to include the figure in the final document.

[Figure]

**p. 11, line 25: Did you estimate the residual error due to turbulences?**

The largest peaks in Fig. 3B are near 14.6 hours. As stated in our revised text, these peaks are due to momentary noise in the calibration signal, and therefore not due to turbulence. Outside of those peaks, the maximum residual to the fit is about 1.8% between 16 and 16.5 hours based on Figure 3B. As stated in the text, we think this is mostly due to residual turbulence, and thus is an estimate of the residual error due to turbulence.

Therefore, we propose to modify the paragraph beginning on page 11, line 23. The original text is shown in black with modifications shown in blue. Please also note the change to the caption for Fig. 3B, which is shown below in response to a comment specifically concerning Fig. 3B.

Turbulence-induced amplitude scintillations (Category 2) were quite significant during the periods of strong convection. These were reduced by almost an order of magnitude via

amplitude ratioing with the calibration signal (Kursinski et al., 2016). The strong peaks near 14.6 hours in Fig. 3B are caused by momentary noise in the calibration signal, which influences the frequency ratioing. Outside of this peak the largest fractional uncertainty is about 1.8% of the vapor pressure (green line). We attribute most of this to turbulent-induced scintillations that remain after the frequency ratioing. Thus, for the conditions of this field experiment, the upper bound for the random error in the vapor pressure retrieval due to turbulence is about 1.8% of the vapor pressure.

**p. 12, line 8: How did you get the value of -0.17 hPa/C? Could you give a short explanation to this?**

This was estimated by running forward simulations of the *am7.2* model for the approximate temperature and vapor pressure conditions observed during this experiment. This was partially addressed in one of the changes made to address comment 18 from Reviewer 1. We will add the additional underlined blue text to address this concern as well.

Page 12, line 6, original text. "For the conditions of this particular experiment, based on the AM7.2 model, the sensitivity of the change in derived water vapor due to a temperature change relative to the reference period temperature was approximately -0.17 hPa/°C."

Revised text. "For the conditions of this particular experiment, based on forward calculations made with *am7.2* for the range of temperature and vapor pressure conditions observed during the experiment, the sensitivity of the change in derived water vapor due to a temperature change relative to the reference period temperature was approximately -0.17 hPa/°C."

**p. 13, line 11: Please give a reference to the Mie cloud model you where applying?**

We will add the following citation at the end of the sentence … Mie cloud model (Bohren and Huffman, 1983).

And the following reference in our list of references.

Bohren, Craig F., and Donald R. Huffman. 1983. *Absorption and scattering of light by small particles*. New York: Wiley.

**p. 14, line 22: Do you have information/graphs on the wind speeds correlating with the shift in time too e.g. by using the radar data?**

Unfortunately we did not estimate wind speed. Our goal was simply to estimate wind direction in order to determine if the sign of the lags between the ATOMMS water vapor measurements and the *in situ* water vapor measurements on Mt. Bigelow made sense.

**p. 15, line 2: Do you have a reference on the Tucson WSR-88 RADAR?**

The Tucson radar type should have been specified as WSR-88D instead of WSR-88. We will add the following citation. However, reviewer #3 pointed out that we first reference radar data on page 7, so the citation will be placed there as well.

Original: "… Tucson WSR-88 RADAR"

Revised:  "…Tucson WSR-88D radar (Crum and Alberty, 1993)."

And reference

Crum, T.D. and R.L. Alberty, 1993: The WSR-88D and the WSR-88D Operational Support Facility. *Bull. Amer. Meteor. Soc.,* **74**, 1669–1688, https://doi.org/10.1175/1520-0477(1993)074<1669:TWATWO>2.0.CO;2

**p. 15, line 20: What was the horizontal distance of the radiosond to the actual microwave path, when passing through the mountain height levels of Mt. Bigelow and Mt. Lemmon? Was the radiosond actually passing the microwave path or was it very close or even far away from the microwave path?**

The original text incorrectly states that the launch point for the radiosonde is about 20 km south of the observation path. The location of the radiosonde is estimated to be about 20 km south of the observation path during the time it crosses the observation path ascending through the altitudes of Mt. Bigelow and Mt. Lemmon. The sentence will be changed as indicated below.

Original Sentence: The sonde launched between 16:30 and 16:45 from a location about 20 km south of the experiment and ascended through the Mt. Bigelow to Mt. Lemmon altitude interval between 16:35 and 16:50.

Revised Sentence: The sonde launched between 16:30 and 16:45 from a location about 28 km southwest of the experiment and ascended through the Mt. Bigelow to Mt. Lemmon altitude interval between 16:35 and 16:50 at a location approximately 20 km south of the observation path.

**p. 19, line 25. Please explaine shortly the other colored lines according to the altitude levels. It is not clear why e.g. the black and red lines show a less high ratio than the green and pink lines and they show even different linear dependencies within the length of the integration path.**

We have decided to make a few changes based on this comment. One of the changes is to make some slight changes to Figure A1.

First, though, here is the response to the question based on the original figure, which is the last figure in the original document and incorrectly labeled as Figure A2.

The results are based on observations from aircraft observations. The black and red lines, referenced by the reviewer, are at the highest altitudes (lowest pressure) lines on the plot, which

correspond with the driest part of the atmosphere. For these altitudes, there is a change in the slope of the ratio std(q)/mean(q) (<q-std>/<q-mean>) when the separation distance becomes smaller than about 10 km. This is likely because when the water vapor content is low, the observed ratio is more affected by instrumental effects. Under these dry conditions, we need separation distances larger than 10 km to be able to observe the effects of anisotropy in the absolute humidity field. Thus in the original figure, at separation distances between measurements longer than 10 km, we start to see the effects of anisotropy in the absolute humidity field for these highest altitude (lowest pressure) lines.

We have updated Figure A1 to show just the 447, 550 and 839 hPa pressure levels which are the only levels of relevance to us and the only levels shown and discussed in Otarola et al. (2011). This removes the odd behavior at the two lowest pressure levels that we believe is tied somehow to the performance of the aircraft hygrometer. We also cleaned up the figure so that the line that is interpolated to the ATOMMS horizontal distance separation does not extend beyond the y axis. The new figure and with an updated caption is shown below.

In carefully reproducing the figure, we also found that the original 13 m estimated length is actually 10 m. This requires that we change the document text references from 13 m to 10 m as listed below. The change to 10 is shown in blue.

Page 19, line 27: Given this power-law exponent and the requirement to keep uncertainties smaller than 1%, the path length required to achieve std($q$)/mean($q$) = 1% is approximately  10 m.

Page 19, line 31: Thus, in situ sensors, accurate to 1% each, would need to be placed every  10 m along a 5.4 km path to achieve an in situ-based path average consistent with the ATOMMS measurements to the 1% level.

[Figure]

**Commented [k1]:** Can you get rid of the blue background? I suspect AMT will not want that there.

**Figure A1: Ratio of the standard deviation of absolute humidity to the mean absolute humidity based on aircraft data taken at different altitudes, which is indicated by the air pressure along different flight paths. The red star on the dashed line constructed for the 550 hPa altitude observations corresponds with the value calculated from the three in-situ sensors operating during the experiment (ratio of 8% 5 for a 5 km path). The slope of the dashed line corresponds with a power law exponent of 0.35 for the dependence of std(q)/mean(q) with the length of the path, which is consistent with Kolomogorov turbulence. Extrapolation of this line to std(q)/mean(q) equal to 1% would indicate that in-situ observations are required every 10 m in order to validate the 1% accuracy of the ATOMMS retrievals. Adapted from Otarola et al. (2011).**

**Fig. 1: Is it possible to mark the position of the radiosond when passing through the altitude level of the two mountains?**

Unfortunately, this is not possible. Figure 1 is about 3 km by 5 km, which zooms on the instrument locations. The sonde was approximately 20 km to the south of the observation path.

**Typo Comments:**

**p. 2, line 9--10: I would recommend to be consistence with the entire paper to write everything in hPa than in mb.**

We will search for all remaining instances of 'mb' and change it to 'hPa' for consistency.

**p. 4, line 30: Please change hydrometers to hydrometeors.**

This typo is actually on line 13 of page 4. Change will be made.

**p. 7, line 21: Please make the notation for the time consistence in the entire document and figures. Sometime 4:30, then 4,5 or even the 12 hour notation is used.**

This was addressed in comment 14 from reviewer 1. We will use 24 hour notation, e.g., 16:30 for all such references to time.

**p. 10, line 25: The +-1% signe is underlined. Please remove the underline.**

The underline will be removed.

**p. 14, line 24 – 27: Please make the time consistent. They do not agree with the given times in the legend of Fig. 8B.**

The text providing the time ranges on page 14 will be changed to be consistent with the figure and a 24 hour clock time that will be used throughout.

**p. 15, line 19: Please make the notation of the Tucson radiosond consistent with legend in Fig. 7.**

The text on line 19 will be changed to match the text in the figure caption

Original text: 00Z Tucson radiosonde

Revised text: 00 UTC Tucson radiosonde

**Fig. 3A: Please use the consisten units e.g. hPa instead of mb and h instead of hours (in Fig 3A and 3B)**
The axis labels will be changed to always use hPa (instead of mb) and h (instead of hours). Below is the updated Fig. 3A. The updated Fig. 3B is included below to address the next comment regarding the right axis labeling of Fig. 3B.

[Figure]

**Fig. 3B: What is the strong peak at about 14.6? In the text % is used but in the figure you use fraction for the right y-axes. Please make it consistent.**

Below is the updated Fig. 3B in which the right axis has been labeled in percent. We added text to the caption for Fig 3B in two places for clarification. One change is to clarify that the right axis is expressed in percent. The other change is to address the comment related to page 11, line 25 above. The added text is shown in blue.

[Figure]

**Figure 3: A.** Retrieved vapor pressure for the 12 retrieval test cases described in the text. Each line is a different color. **B.** Blue line and left axis indicate the half range, which is one half of the maximum minus minimum vapor pressure from the 12 retrieval cases; green line and right axis is the half range divided by the absolute vapor pressure at each retrieval point expressed in percent. The strong peaks near 14.6 hours are due to momentary noise in the calibration signal.

**Fig. 8B: Please make the time in the label, legend and text consistent. Please give units?**
The text in the caption referring to the time will be changed to a 24 hour clock formation as shown in the figure legend. Original caption shown with changes highlighted in blue.

**Figure 8: A.** Vapor pressure derived from ATOMMS observations (blue) and measured with an in-situ sensor on Mt. Bigelow (red). Also shown in other colors are four time segments of the in-situ observations shifted in time (as described in text) to highlight correlation between the two vapor pressure data sets. The time shift for each colored line is indicated in panel (B). **B.** 5 Cross-correlation coefficients as a function of sample lags between the ATOMMS-derived vapor pressure and in-situ measurements of water vapor taken on Mt. Bigelow. The four lines correspond with the four time segments described in the text: green (14:03 – 14.47 hours), black (14:59 to 15:30 hours), cyan (16:00 to 16:25 hours), and magenta (16:45 to 17:23 hours).

**Fig. A1: Please remove the title of the plot or make the symbols for standard deviation and mean value consistent. Please correct the x-axes label.**

These issues have been fixed with our changes to figure A1 as described in our response to the comments related to page 19, line 25 above.

---

## Referee Report (RR1)

Second Review of Ward et al.  2018.
David K. Adams (dave.k.adams@gmail.com)

Recommendation:   The paper is essentially publishable as is.  Below, I have made some grammar and style corrections that should be made.

Abstract

Write "spectrum to retrieve water vapor with a random uncertainty **of** less than 1%."

Line 10
"it may be underestimated", this is a bit unclear.  Are you referring to predicted amplification of precipitation or to predicted increases in water vapor?

Line 12  Write "In recognition of the strengths and weaknesses of GPS RO and radiance measurements **as well as**  the need for better..."

Line 24 Write "from near the surface **to** well into the mesosphere."

Line 6  Write "At still higher cloud and rain opacities**,**  such as..."

Line 14 Write "Finally, in Section 7**,** ..."

Page 7  Line 1. Am I confused or should this be  " **less ambiguous**."

Page 8 and Page Line 25-30 to Line 5   Can you join these ideas into one paragraph?  It´s not very smooth reading.

Page 9  Line 28 and 34.  You have italicized  *Kursinski et al. (2009)*

 On Line you use "completed every **48.8 s.**" And then on line 11 you use **48.8 seconds.**  I guess you should probably use one or the other.

Line 13  Write "four times 50 ms**,** or 200 ms."

Line 12 Remove one of the periods,  " described in Section 5."
Line 15  Write "The **path-averaged** vapor pressure varied
Line 19,  Write "In Section 6**,** we ..."

Line 2. You capitalize "First Reference period" above, but not "second reference period"

Line 13 Maybe better showers instead of shower.

Line 18  Write  " ...the well-mixed, dry adiabatic,..."

Line 2 Write  " path without interfering with  **the** signal itself."

---

## Editor Decision (ED1)

Review comment of Anonymous Referee # 2 on the Authors Response:

Retrieval of Water Vapor using Ground-based Observations from a Prototype ATOMMS Active cm- and mm- Wavelength Occultation Instrument
Dale M. Ward et al.

The authors response did clearify all referees concerns and issues on the discussion paper. It improves and clearifies the understanding of the paper and is ready for publication if  carefully implemented into the paper structure.

---

## Author Response (AR2)

**Responses to Comments From Reviewer**

First, we wish to thank David Adams for his second helpful review. We first copy the comments of the reviewer *(in italics)* with our response below. In cases where the text is significantly modified, we show both the original text and the revised text, with the changes shown in blue.

*Second Review of Ward et al. 2018.*
*David K. Adams ([dave.k.adams@gmail.com](mailto:dave.k.adams@gmail.com))*

*Recommendation: The paper is essentially publishable as is. Below, I have made some grammar and style corrections that should be made.*

*Abstract*
*Write "spectrum to retrieve water vapor with a random uncertainty **of** less than 1%."*

We simply added the bold word **"of"** into the sentence.

Page 2 Line 10
*"it may be underestimated", this is a bit unclear. Are you referring to predicted amplification of precipitation or to predicted increases in water vapor?*

We are referring to the amplification of precipitation, which we clarify.

Original Sentence: Predicted amplification of extreme precipitation with warmer temperatures is tied directly to predicted increases in extreme water vapor concentrations and it may be underestimated (e.g., Allan and Soden, 2008).

Revised sentence: Predicted amplification of extreme precipitation with warmer temperatures is tied directly to predicted increases in extreme water vapor concentrations and future extreme precipitation may be underestimated (e.g., Allan and Soden, 2008).

*Page 3 Line 12*
*Write "In recognition of the strengths and weaknesses of GPS RO and radiance measurements **as well as** the need for better..."*

Change the word "and" in the sentence to "as well as" per comment.

*Page 3 Line 24*
*Write "from near the surface **to** well into the mesosphere."*

We simply added the bold word **"to"** into the sentence.

Page 6 Line 6
Write "At still higher cloud and rain opacities**,** such as..."

We simply added the bold comma **","** into the sentence.

*Page 6 Line 14*
*Write "Finally, in Section 7**,** ..."*

We simply added the bold comma **","** into the sentence.

*Page 7 Line 1.*
*Am I confused or should this be " **less ambiguous**."*

You are not confused. We changed **less unambiguous** to less ambiguous

*Page 8 and Page 9 Line 25-30 to Line 5*
*Can you join these ideas into one paragraph? It´s not very smooth reading.*

Significant wording changes were made for clarity and smoothness.

Original Text:
    In terms of spectroscopy, the ATOMMS measured line shape across the 4 GHz interval above the 183 GHz line center agreed with the HITRAN line shape with a standard deviation of 0.3% (Kursinski et al. 2012), some 8 times better than the previously best estimate of Payne et al. (2008).  ATOMMS mountaintop measurements between 5 and 25 GHz above the line center revealed discrepancies with the HITRAN line shape (Kursinski et al., 2016b) which may help explain inconsistencies in 183 GHz derived water vapor estimates (Brogniez et al., 2016) and may be associated with atmospheric turbulence (Calbet et al., 2018).  The ATOMMS measurements also revealed the shape of the 183 GHz line as represented in the Liebe et al., (1993) model is incorrect (Kursinski et al., 2012).  The Liebe model is popular, having been referenced more than 600 times in the literature, and is still being used.

    Kursinski et al. (2012) combined ATOMMS High-Band measurements with precipitation radar measurements to derive cloud liquid water content (LWC) along the ATOMMS signal path.  Kursinski et al. (2016b) demonstrated the ability to derive both cloud LWC and rainfall rates by combining the ATOMMS Low-Band and High-Band measurements.

    Kursinski et al. (2016b) derived the strength of atmospheric turbulence from scintillations of the ATOMMS signal amplitudes and demonstrated the ability to significantly reduce these turbulent amplitude variations via amplitude ratioing, in order to derive accurate water vapor estimates in turbulent conditions.

Revised Text:

Accurate knowledge of spectroscopy is key to interpreting the ATOMMS measurements. ATOMMS itself is perhaps the best 183 GHz spectrometer ever implemented. Its measurements of the line shape near the 183 GHz line center match that of the HITRAN model to within 0.3% (Kursinski et al. 2012) which agrees 8 times better than the best prior estimates of Payne et al. (2008). These same measurements revealed that the line shape of the popular Liebe et al., (1993) model is incorrect (Kursinski et al., 2012). Farther from line center, 5 to 25 GHz above line center, ATOMMS measurements revealed significant discrepancies with the HITRAN line shape (Kursinski et al., 2016b). These discrepancies may help explain inconsistencies between 183 GHz derived water vapor estimates discussed in Brogniez et al. (2016) that may be associated with atmospheric turbulence (Calbet et al., 2018).

In terms of sensing hydrometeors, Kursinski et al. (2012) derived cloud liquid water content (LWC) by combining ATOMMS High-Band measurements with precipitation radar measurements along the ATOMMS signal path. Kursinski et al. (2016b) further demonstrated the ability to derive both cloud LWC and rainfall rates by combining the ATOMMS Low-Band and High-Band measurements. ATOMMS also acts as a scintillometer to sense atmospheric turbulence. Kursinski et al. (2016b) derived the strength of atmospheric turbulence from scintillations of the ATOMMS signal amplitudes and further demonstrated how these turbulent amplitude variations can be reduced via amplitude ratioing, as needed to derive accurate water vapor estimates in turbulent conditions.

*Page 9 Line 28 and 34. You have italicized Kursinski et al. (2009)*

Italics were removed from the two citations in question.

*Page 12 On Line*
*you use "completed every **48.8 s."** And then on line 11 you use **48.8 seconds.** I guess you should probably use one or the other.*

Agree. All instances of seconds within the paper have been changed to s.

*Line 13 Write "four times 50 ms**,** or 200 ms."*

We simply added the bold comma **","** into the sentence.

*Page 13 Line 12*
*Remove one of the periods, " described in Section 5."*

Done.

*Line 15*
*Write "The **path-averaged** vapor pressure varied*

Bold hyphen above added between words path and averaged.

*Line 19,*
*Write "In Section 6**, we ..."*

We simply added the bold comma **","** into the sentence.

*Page 15*
*Line 2. You capitalize "First Reference period" above, but not "second reference period"*

The word reference is lower cases everywhere, except in the last sentence in section 4 of the paper. That sentence is written with a short-hand notation for the reference periods, which may be confusing. Therefore, we made the following change for clarification and consistency.

Orignal Sentence: Near the cloud peak, the Reference 1 water vapor solutions are greater than the Reference 2 solutions by only 0.03 hPa (0.2%), indicating the level of robustness of these vapor pressure retrievals.

Revised Sentence: Near the cloud peak, the water vapor solution based on the first reference period is greater than the water vapor solution based on the second reference period by only 0.03 hPa (0.2%), indicating the level of robustness of these vapor pressure retrievals.

*Page 19*
*Line 13 Maybe better showers instead of shower.*

We believe the original text is fine, since the word activity is associated with both shower and thunderstorm. No changes made.

Text in question: Although the winds were occasionally gusty, with variable direction due to shower and thunderstorm activity,

*Page 20*
*Line 18 Write " ...the well-mixed, dry adiabatic,..."*

Bold hyphen above added between words well and mixed.

*Page 25*
*Line 2 Write " path without interfering with **the** signal itself."*

The bold word "the" added into the sentence.

[revised manuscript text omitted]